# Exploring Criteria of Loss Reweighting to Enhance LLM Unlearning

Puning Yang [1]  Qizhou Wang [1]  Zhuo Huang [2]  Tongliang Liu [2]  Chengqi Zhang [3]  Bo Han [†1]

## Abstract

Loss reweighting has shown significant benefits for machine unlearning with large language models (LLMs). However, their exact functionalities are left unclear and the optimal strategy remains an open question, thus impeding the understanding and improvement of existing methodologies. In this paper, we identify two distinct goals of loss reweighting, namely, **Saturation** and **Importance**—the former indicates that those insufficiently optimized data should be emphasized, while the latter stresses some critical data that are most influential for loss minimization. To study their usefulness, we design specific reweighting strategies for each goal and evaluate their respective effects on unlearning. We conduct extensive empirical analyses on well-established benchmarks, and summarize some important observations as follows: **(i)** Saturation enhances efficacy more than importance-based reweighting, and their combination can yield additional improvements. **(ii)** Saturation typically allocates lower weights to data with lower likelihoods, whereas importance-based reweighting does the opposite. **(iii)** The efficacy of unlearning is also largely influenced by the smoothness and granularity of the weight distributions. Based on these findings, we propose **SatImp**, a simple reweighting method that combines the advantages of both saturation and importance. Empirical results on extensive datasets validate the efficacy of our method, potentially bridging existing research gaps and indicating directions for future research. Our code is available at https://github.com/tmlr-group/SatImp.

[1]TMLR Group, Department of Computer Science, Hong Kong Baptist University. [2]Sydney AI Centre, The University of Sydney. [3]Department of Data Science and Artificial Intelligence, The Hong Kong Polytechnic University. Correspondence to: Bo Han <bhanml@comp.hkbu.edu.hk>.

*Proceedings of the 42$^{nd}$ International Conference on Machine Learning*, Vancouver, Canada. PMLR 267, 2025. Copyright 2025 by the author(s).

## 1. Introduction

The remarkable capabilities of large language models (LLMs) are primarily attributed to the usage of extensive, web-scale datasets. However, these datasets also pose substantial risks as they often contain illegal, private, and potentially sensitive content (Karamolegkou et al., 2023; Patil et al., 2023). The associated LLMs may learn and later reproduce this sensitive information, raising ethical concerns and legal implications (Yu et al., 2023a; Yao et al., 2023; Barrett et al., 2023; Li et al., 2023a; Zhou et al., 2024; 2025). It has spurred recent research into LLM unlearning (Yao et al., 2023; Zhang et al., 2024; Fan et al., 2024; Maini et al., 2024), which seeks to remove the influence of harmful data post-training, thereby mitigating the risk associated with reproducing undesirable information in LLM outputs.

To implement LLM unlearning, existing methods typically built upon gradient ascent (GA) (Yao et al., 2023; Maini et al., 2024; Rafailov et al., 2024; Fan et al., 2025), which minimizes the log-likelihood of model responses targeted to be unlearned. Although GA-based methods are highly powerful to remove unwanted responses, they frequently result in over-forgetting (Liu et al., 2024d; Ye et al., 2025), which severely impair the model performance on other, non-targeted tasks, undermining its general functionality.

To overcome this issue, several studies (Rafailov et al., 2024; Zhang et al., 2024; Maini et al., 2024) explore loss reweighting to regulate their unlearning processes, either implicitly or explicitly. Generally speaking, loss reweighting, cf., Eqs. (3)-(4), operates on the premise that certain data points (or tokens) warrant heightened attention, with their loss values and gradients being adjusted upwards over others. However, despite the development of various reweighting strategies, the precise physical interpretations or specific functions of these strategies remain ambiguous. This uncertainty complicates the determination of which avenues to explore in the pursuit of optimal reweighting strategies.

This paper seeks to deepen our understanding of loss reweighting through a series of well-crafted experiments. We outline two primary categories of loss reweighting, namely, **Saturation** which advocates emphasizing examples that are not yet sufficiently unlearned, and **Importance** which focuses on the critical examples for achieving the unlearning goal. Specifically, the insufficiently unlearned

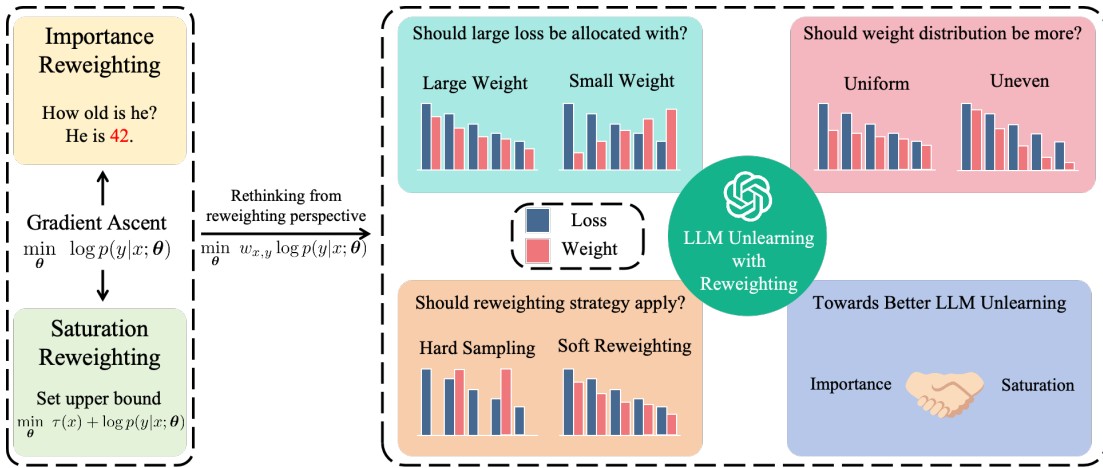

*Figure 1.* Overview of our paper. Beginning with a summary of existing LLM unlearning methods, we categorize them into importance-based and saturation-based reweighting, followed by a comprehensive analysis. Furthermore, we investigate the effects of specific reweighting operations and identify three core issues that significantly impact unlearning performance. Finally, we clarify the optimization direction for LLM unlearning, leading to a simple yet strong baseline: SatImp.

data are optimized slowly, hindering the convergence of unlearning. On the other hand, some critical data are important for unlearning, which could lead to fast convergence of loss function. As we can see, the above two strategies focus on two complementary types of data; however, their relationship and effectiveness remain unclear. To explore the optimal reweighting criteria, we implement specific loss reweighting methods for each of the aforementioned categories, and conduct extensive experiments on well-established benchmarks. We summarize our observations in two perspectives in the following.

**Comparisons between Saturation and Importance.** We observe that importance-based reweighting tends to allocate large weights to low-likelihood tokens, whereas saturation-based reweighting behaves in the opposite manner. Both methods aid in retention, but saturation-based methods exhibit superior overall efficacy. Additionally, we find that importance and saturation focus on different aspects of data, and their combinations can lead to further improvements.

**Impacts of Varying Weight Distributions.** Our findings indicate that an excessively sharp distribution of weights may lead to insufficient unlearning, whereas an overly uniform weight distribution will significantly degrade performance retention. Additionally, when comparing between token- and instance-wise reweighting (cf., Eqs. (3)-(4)), we find that token-wise reweighting often results in better performance, echoing previous studies (Wang et al., 2025b) and extends their implications on a wider scale.

Based on our aforementioned findings, we simplify the reweighting strategies related to importance and saturation (cf., Section 5), and further propose an enhanced version named **SatImp** that merges the strengths of both importance- and saturation-based methods. We conducted comprehen-

sive experiments across multiple well-established LLM unlearning benchmarks, and the results clearly demonstrate the superiority of our SatImp over its advanced counterparts.

## 2. Preliminaries

Considering an LLM parameterized by $\boldsymbol{\theta}$ that has been trained on a substantial web-scale corpora $\mathcal{D}_{\text{tr}}$, the task of LLM unlearning involves removing the parameterized knowledge related to harmful information. This content is typically associated with a specific subset of samples within $\mathcal{D}_{\text{tr}}$, known as the unlearn set $\mathcal{D}_{\text{u}}$. In contrast to supervised fine-tuning, which minimizes negative log-likelihood via gradient descent, LLM unlearning typically adopts gradient ascent (GA)-based methods that correspond to minimize the log-likelihood on the unlearn set, as described by

$$\min_{\boldsymbol{\theta}} \mathbb{E}_{(x,y) \sim \mathcal{D}_{\text{u}}} \log \ p(y|x; \boldsymbol{\theta}). \tag{1}$$

Eq. (1) decreases the model accuracy on harmful data, making the model effectively remove this harmful information.

However, the use of gradient ascent is known to potentially cause significant degradation in model performance on other, unrelated data. To mitigate this, regularization terms are often added to the GA unlearning objective. This term involves a retain set $\mathcal{D}_{\text{r}}$—data that are not included in $\mathcal{D}_{\text{u}}$—and leads to gradient difference (GD) as follows:

$$\min_{\boldsymbol{\theta}} \mathbb{E}_{(x,y) \sim \mathcal{D}_{\text{u}}} \log \ p(y|x; \boldsymbol{\theta}) - \lambda \mathbb{E}_{(x,y) \sim \mathcal{D}_{\text{r}}} \log \ p(y|x; \boldsymbol{\theta}), \tag{2}$$

where $\lambda$ is a trade-off hyper-parameter to be tuned.

**Loss Reweighting.** However, extreme values in the gradients of the GA component can dominate those of the regularization term (Wang et al., 2025a), typically causing

GD to still degrade overall performance. Numerous studies suggest modifying the original GA objective through reweighting, in accordance with the general formulation of

$$\mathbb{E}_{(x,y)\sim\mathcal{D}_{\mathrm{u}}} w_{x,y} \log\ p(y|x;\boldsymbol{\theta}), \tag{3}$$

where $w_{x,y}$ can be any function to be specified. Such an instance-wise reweighting scheme can be generalized as

$$\mathbb{E}_{(x,y)\sim\mathcal{D}_{\mathrm{u}}} \sum_{k}^{|y|} w_{x,y,k} \log\ p(y^k|y^{<k},x;\boldsymbol{\theta}), \tag{4}$$

where $p(y|x;\boldsymbol{\theta})$ is reformulated as $\prod_k p(y^k|y^{<k},x;\boldsymbol{\theta})$ following the standard probabilistic decomposition and the weights are assigned in a token-wise manner.

A representative example is **WGA** (Wang et al., 2025b), using a probability-based reweighting mechanism as

$$w_{x,y,k}^{\mathrm{wga}} = p(y^k|y^{<k},x;\boldsymbol{\theta})^\beta, \tag{5}$$

where $\beta$ is a temperature hyper-parameter that controls the strength of reweighting. Another wide-recognized unlearning objective, termed Negative Preference Optimization (**NPO**) (Zhang et al., 2024), also relies on loss reweighting to enhance the vanilla GA, albeit in an implicit manner. Its original objective is defined as

$$\mathbb{E}_{(x,y)\sim\mathcal{D}_{\mathrm{u}}}\Big[ -\frac{2}{\beta} \log\ \sigma(-\beta \log\ (\frac{p(y|x;\boldsymbol{\theta})}{p(y|x;\boldsymbol{\theta}_{\mathrm{ref}})}))\Big], \tag{6}$$

where $\sigma(\cdot)$ is the sigmoid function, $\beta$ is a temperature hyper-parameter, and $\boldsymbol{\theta}_{\mathrm{ref}}$ represents the reference model before unlearning. When analyzing its gradients (Zhang et al., 2024), it becomes apparent that it can be reformulated as a reweighting form of GA, which is given by

$$w_{x,y}^{\mathrm{npo}} = \frac{2p(y|x;\boldsymbol{\theta})^\beta}{p(y|x;\boldsymbol{\theta})^\beta + p(y|x;\boldsymbol{\theta}_{\mathrm{ref}})^\beta}. \tag{7}$$

Existing reweighting strategies have shown their promising performance, yet there is a possible room to further improve their designs. Moreover, the functionality of some reweighting mechanisms, such as NPO, remains poorly understood. This paper seeks to address these gaps by providing a comprehensive analysis of various reweighting strategies and proposing potential pathways for its future research.

## 3. Saturation and Importance

To examine and analyze different methods of loss reweighting, we categorize their mechanisms into two primary types: **importance-based reweighting** and **saturation-based reweighting**. In this section, we discuss their motivations, define each type, and explore their implementations.

### 3.1. Importance-Based Reweighting

Importance-based reweighting assumes that certain data points or tokens are inherently more important than others and, therefore, should be assigned greater weights. This approach is of our particular interest as certain tokens play a pivotal role in determining the whole outcome (Qi et al., 2025). For example, consider a QA pair from the TOFU benchmark (Maini et al., 2024) as follows:

> Q: Which universities did Hina Ameen attend to study geology?
>
> A: Hina Ameen completed her Bachelor's at the University of Karachi and proceeded to obtain her Master's and Ph.D. in Geology at the University of Cambridge.

Words highlighted in red are essential for conveying the overall meaning of the text. Emphasizing these key elements can largely distort its interpretation, thus highlighting the potential of importance-based reweighting.

**Implementation**. From the very beginning, it is not easy to automatically identify important tokens. Therefore, to facilitate our assess of importance-based reweighting, we have expanded the original TOFU benchmark by manually labeling each token to determine their importance (binary labeling of important/unimportant). Moreover, to increase the accuracy of our labeling, each data point has been reviewed by 4 members from our team. Tokens that received agreement from at least the half are considered to have accurate labels. For each QA pair $(x,y)$ within TOFU, we denote its associated set of important tokens as $\mathcal{T}_{x,y}$. Then, our importance-based reweighting involves adjusting the weight of each token in $\mathcal{T}_{x,y}$ to have greater emphasis, defined as

$$w_{x,y,k}^{\mathrm{imp}} = \begin{cases} 1-p, & \text{if } k \in \mathcal{T}_{x,y} \\ p, & \text{if } k \notin \mathcal{T}_{x,y} \end{cases}, \tag{8}$$

where $p > 0$ is hyper-parameter that controls the weight allocation. It typically should be set less than $0.5$ such that the weights assigned to important tokens are larger than that of others, aligning with our objective of emphasizing important tokens. Moreover, as observed, Eq. (8) employs a token-wise reweighting mechanism where the assigned weights remain fixed during unlearning.

### 3.2. Saturation-Based Reweighting

Considering common scenarios of excessive unlearning, which can largely impair model retention (Liu et al., 2024d; Wang et al., 2025a), saturation-based reweighting suggests for regulating the extent of unlearning. Specifically, data points or tokens that have not been fully unlearned should be assigned greater weights. Then, as these elements become sufficiently unlearned, their corresponding weights should decrease, thereby preventing excessive unlearning.

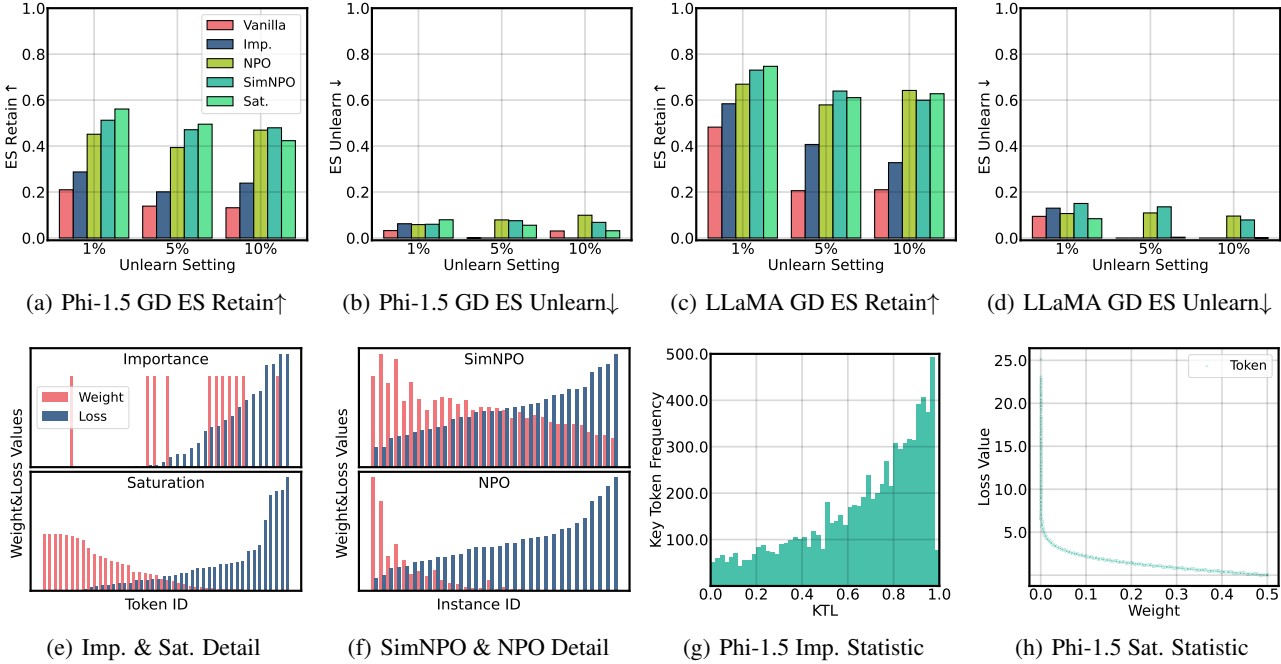

(a) Phi-1.5 GD ES Retain↑   (b) Phi-1.5 GD ES Unlearn↓   (c) LLaMA GD ES Retain↑   (d) LLaMA GD ES Unlearn↓

(e) Imp. & Sat. Detail   (f) SimNPO & NPO Detail   (g) Phi-1.5 Imp. Statistic   (h) Phi-1.5 Sat. Statistic

*Figure 2.* Comparisons between saturation- and importance-based reweighting. We depict TOFU settings (x-axis) versus ES scores (y-axis) on retain (ES Retain) and unlearn (ES Unlearn) data. ↑ / ↓ indicate larger / smaller values are preferable. While saturation-based reweighting outperforms importance-based reweighting on retention tasks, it slightly underperforms on unlearning tasks (a-d). This performance disparity stems from differences in the reweighting mechanisms: importance-based reweighting tends to assign higher weights to tokens with lower likelihoods (e, g), whereas saturation-based reweighting prioritizes tokens with higher likelihoods (e, f, h).

**Implementation.** To implement saturation-based reweighting, we need to measure the extent of leftover knowledge within the model. Following (Sablayrolles et al., 2019), which suggests a Bayes optimal strategy to quantify the extent of knowledge for membership inference (Shokri et al., 2017), we adopt the reweighting strategy as follows:

$$w^{\text{sat}}_{x,y,k} = \frac{p(y^k|y^{<k}, x; \boldsymbol{\theta})}{p(y^k|y^{<k}, x; \boldsymbol{\theta}) + \tau}, \quad (9)$$

with $\tau > 0$ being the hyper-parameter of calibration. Generally speaking, a higher value of $w^{\text{sat}}_{x,y,k}$ indicates that more knowledge is preserved, which aligns with the goal of importance-based reweighting. It is observed that Eq. (9) implements a dynamic token-wise reweighting mechanism, offering the same granularity as importance-wise reweighting in Eq. (8). Furthermore, previous studies, such as (Wang et al., 2025b), have demonstrated that token-wise reweighting is more powerful than its instance-wise counterpart. Therefore, we default to adopting the token-wise setup.

## 4. Empirical Analyses

This section presents empirical studies that examine the reweighting strategies previously discussed. We explore their methods of simplification, assess their efficacy, and analyze the varying impacts of their weight distributions.

**Configurations.** We conduct our experiments on the well-established TOFU benchmark (Maini et al., 2024), comprising synthetic authors profiles to test the efficacy of various unlearning methods in addressing privacy concerns. This benchmark includes three unlearning settings, aiming to remove 1%, 5%, and 10% of the total dataset. We employ the pre-trained models LLaMA-2-7B (Touvron et al., 2023) and Phi-1.5 (Li et al., 2023b). For performance assessment, we default to report the ES scores (Wang et al., 2025a), which have been shown to be more reliable than others, such as FQ and MU proposed by the TOFU benchmark. Generally, we prefer smaller ES scores for targeted data and larger values for non-targeted data, reflecting model ability of removal and retention, respectively. In our experiments, we default to setting the hyper-parameters of $p = 0.3$ and $\tau = 1$. Moreover, we explore other benchmarks such as WMDP (Li et al., 2024) and MUSE (Shi et al., 2024), and employ additional metrics to further solidify our analyses. Please refer to Appendix B.1 for their more details.

### 4.1. Comparsion Between Importance- and Saturation-based Reweighting

We begin by assessing the respective performance of the proposed reweighting methods. We present our results in Figures 2(a)-2(d), reporting the results of importance (Imp.)

*Table 1.* Enhancing unlearning via combining $w_{x,y,k}^{\text{imp}}$ and $w_{x,y,k}^{\text{sat}}$. ES scores for targeted (ES Un.) and untargeted data (ES Re.) are reported. ↑ / ↓ indicate larger / smaller values are preferable.

| Method | Phi-1.5 | | LLaMA-2-7B | |
|---|---|---|---|---|
| | ES Re. ↑ | ES Un. ↓ | ES Re. ↑ | ES Un. ↓ |
| $w_{x,y,k}^{\text{imp}}$ | 0.1655 | 0.0451 | 0.4082 | 0.0918 |
| $w_{x,y,k}^{\text{sat}}$ | 0.1277 | **0.0057** | 0.3863 | **0.0565** |
| Comb. | **0.1711** | 0.0328 | **0.4282** | 0.0649 |

following Eq. (8) and that of saturation (Sat.) following Eq. (9). Here, we focus on the 1% unlearning setup as a case study. As observed, both strategies show overall superior performance compared with the baseline of GA (Vanilla), with notable improvements in retention and comparative results of removal. Moreover, in terms of retention, saturation-based reweighting notably outperforms importance-based reweighting. Conversely, importance-based reweighting excels in achieving a more complete process in unlearning.

**Combination of Reweighting.** Moreover, given that importance and saturation focus distinct facets of unlearning, each enhancing its overall effectiveness, it is natural to ask whether their combination could yield additional improvements. To this end, we explore an intuitive combination of importance and saturation through their product of $w_{x,y,k}^{\text{sat}} \cdot w_{x,y,k}^{\text{imp}}$. We conduct experiments for this combined reweighting strategy and summarize the results in Table 1, the hyper-parameter is adjusted to $p = 0.4, \tau = 1$. As observed, the combination (Comb.) indeed leads to further improvements compared to using $w_{x,y,k}^{\text{sat}}$ or $w_{x,y,k}^{\text{imp}}$ alone. This observation underscores the potential of integrated reweighting, which will be further explored in Section 5.

**Simplification of Reweighting.** Although the concepts of importance and saturation are straightforward, their implementations are not that easy, particularly for importance-based reweighting that requires tedious manual labeling. It motivates us to explore further simplifications to make these proposed reweighting methods more practical. Here, we explore the relationship between the weight distributions given by saturation and importance, and the loss distributions (negative log-likelihood) across a random batch of samples. As an example, Figure 2(e) presents an example on 1% setup with Phi-1.5. Overall, we observe that importance-based reweighting tends to associate key annotations with low-likelihood tokens, whereas saturation-based reweighting generally allocates smaller weights to them.

To further solidify our observation, we analyze across the entire unlearning process and conduct a simple statistical analysis discussed as follows. Specifically, we first propose a Key-Token Labeling (KTL) index for importance-based reweighting. Considering a key token $k_i \in \mathcal{T}_{x,y}$, if the likelihood of this token ranks as the $M$-th largest among the

likelihoods in its sample, the KTL index can be defined as:

$$\mathcal{K}(k_i) = \frac{M}{N}. \quad (10)$$

Heuristically, if a key token has a lower likelihood, the value of $\mathcal{K}(\cdot)$ for this token is larger. We record all KTL values throughout the entire training process and analyze their occurrence frequencies across different intervals. The results are shown in Figure 2(g). As observed, the frequency of key tokens increases with lower likelihood values, supporting our assumption of a strong correlation between the key-token index and significant loss in each instance.

Second, for the saturation-based reweighting, we directly record the negative log-likelihood and weight value of each token throughout the complete training process, representing each token as a point in the loss-weight space. The result, shown in Figure 2(h), clearly highlights the strict inverse correlation between losses and weights, supporting our assumption again. More details are shown in Appendix C.1.

**Understanding Previous Reweighting.** By examining the relationship between likelihood distributions and weight distributions, we can analyze the behaviors of previous reweighting strategies. Here, we consider NPO and Sim-NPO (Fan et al., 2024), an improved version of NPO, as two examples. We observe that both NPO and SimNPO primarily focus on the saturation perspective of reweighting, with SimNPO leading to a smoother weight distribution. This finding contradicts the previous assumption suggested by (Wang et al., 2025b) that NPO concentrates on key tokens. Consequently, our $w_{x,y,k}^{\text{imp}}$ remains the only method capable of achieving importance-based reweighting. More validations about existing methods are presented in Appendix C.1.

**Summary.** Based on our analysis of the relationship between likelihood and weight distributions, we propose simplifying important- and saturation-based reweighting strategies. Specifically, we suggest allocating large weights to low-likelihood tokens for importance-based reweighting and small weights to low-likelihood tokens for saturation-based reweighting, which lead to Simple Importance (**SimImp**) and Simple Saturation (**SimSat**) in the following:

$$\begin{aligned} w_{x,y,k}^{\text{simsat}} &= p(y^k|y^{<k}, x; \boldsymbol{\theta})^{\beta} \\ w_{x,y,k}^{\text{simimp}} &= (1 - p(y^k|y^{<k}, x; \boldsymbol{\theta}))^{\beta}, \end{aligned} \quad (11)$$

where $\beta$ is a hyper-parameter to control the smoothness of weight distribution. Note that $w_{x,y,k}^{\text{simsat}}$ is in the same form as WGA. Therefore, we can view Eq. (11) as generalizing WGA to cover more characteristics of reweighting. Later in Section 4.2, we conduct experiments to show that SimImp and SimSat share the same properties as $w_{x,y,k}^{\text{sat}}$ and $w_{x,y,k}^{\text{imp}}$, and please refer to Appendix E for more results.

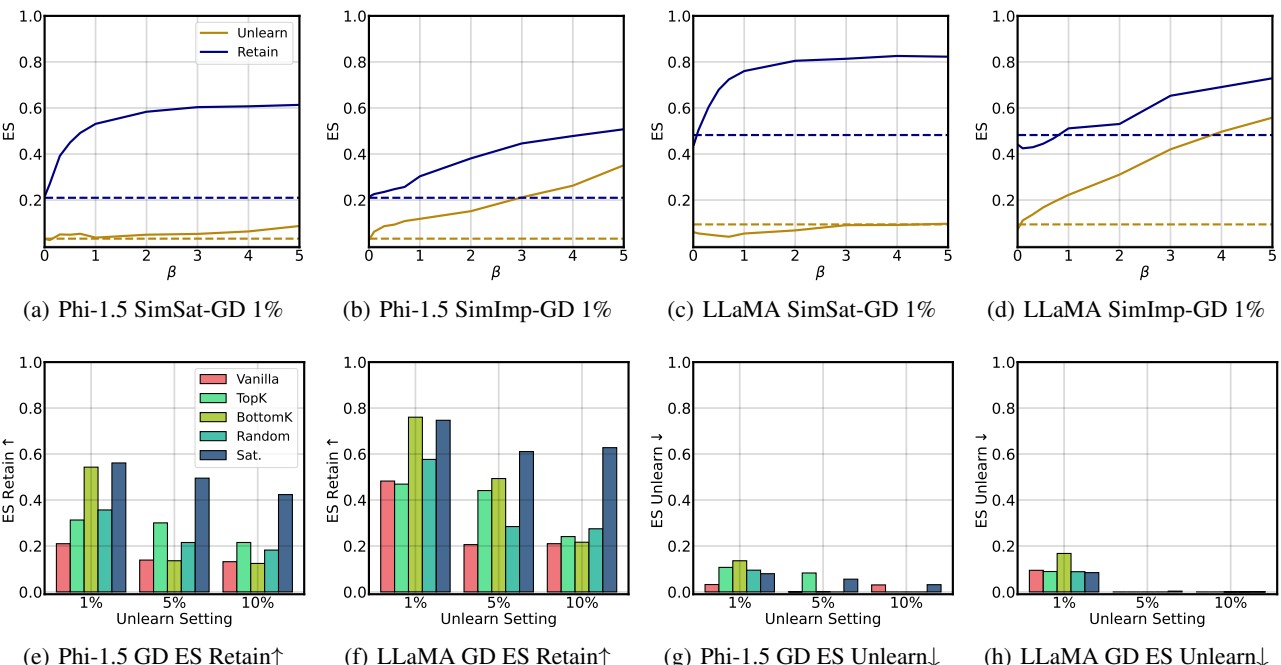

*Figure 3.* Correlations between performance and weight distribution. A comprehensive comparison between SimSat and SimImp indicates that each has its own performance characteristics, with SimSat achieving a better unlearn-retain trade-off. Model performance is sensitive to the smoothness of weight distribution (varying $\beta$). Besides, while hard-sampling is effective in enhancing LLM unlearning performance, it still underperforms saturation-based methods. Dashed line represents baseline (vanilla GD) performance.

### 4.2. Impacts of Weight Distributions

After showing the powers of importance- and saturation-based reweighting, we aim to further explore the effects of specific types of weight distributions. For example, we can investigate how varying shapes influence the overall efficacy of unlearning by varying $\beta$ in Eq. (11), which can control the shapes of our weight distribution. Moreover, we can extend Eq. (11) to include methods such as hard sampling and instance-wise reweighting, thereby delving further into the paths towards effective reweighting.

**Varying Smoothness.** We begin by exploring the impacts of the smoothness for the weight distribution. As indicated by Eq. (11), increasing $\beta$ causes the weights to concentrate more on data with extreme loss values, leading to more peaked distribution. Conversely, decreasing $\beta$ results in more uniform distribution of weights across data, allowing us to study how varying degrees of smoothness affect the learning efficacy. In Figures 3(a)-3(d), we illustrate the model performance with respect to both removal and retention. As observed, for both SimSat and SimImp, there are notable improvements in retention performance and degradation in removal performance as $\beta$ increases. Therefore, we recommend carefully selecting the appropriate values of $\beta$ to trade-off between removal and retention.

**Hard Sampling vs. Soft Reweighting.** Many previous works (Byrd & Lipton, 2019) have raised concerns that loss

---

**Algorithm 1** Hard Sampling

1: **Input:** token-wise likelihood $L = \{\ell_k\}$ for a data point $(x, y)$, sampling strategy, fraction $\beta$
2: **Output:** weight $w^{\text{TopK}}, w^{\text{BottomK}}, w^{\text{Random}}$;
3: Get sampling size $s \leftarrow \beta \times \texttt{len}(y)$;
4: Initialize $w = \{w_i = 0 \mid i = 1, 2, ..., y\}$;
5: Sort likelihoods $L \leftarrow \texttt{sortAscending}(L)$
6: **if** strategy **is** TopK **then**
7: $\quad w_k^{\text{TopK}} = 1, k \in L[0 : s]$;
8: **else if** strategy **is** BottomK **then**
9: $\quad w_k^{\text{BottomK}} = 1, k \in L[\text{len}(y) - s : \text{len}(y)]$;
10: **else if** strategy **is** Random **then**
11: $\quad w_k^{\text{Random}} = 1, k \in \texttt{randomSampling}(y, s)$;
12: **end if**

---

reweighting might not be reliable, especially for deep models with large capacity. To assess whether hard sampling can be more effective than reweighting, we propose Algorithm 1, which incorporates three specific sampling strategies of `TopK`, `BottomK`, and `Random` sampling. `TopK` echos the hard sampling version of $w^{\text{topK}}$, while `BottomK` corresponds to that of $w^{\text{bottomK}}$. Additionally, `Random` works as a simple baseline by randomly sampling tokens to ensure fair comparison. We present the results across various unlearning setups in Figures 3(e)-3(h). As observed, hard sampling can improve the overall efficacy of unlearning, yet it does not perform as well as the reweighting strategies

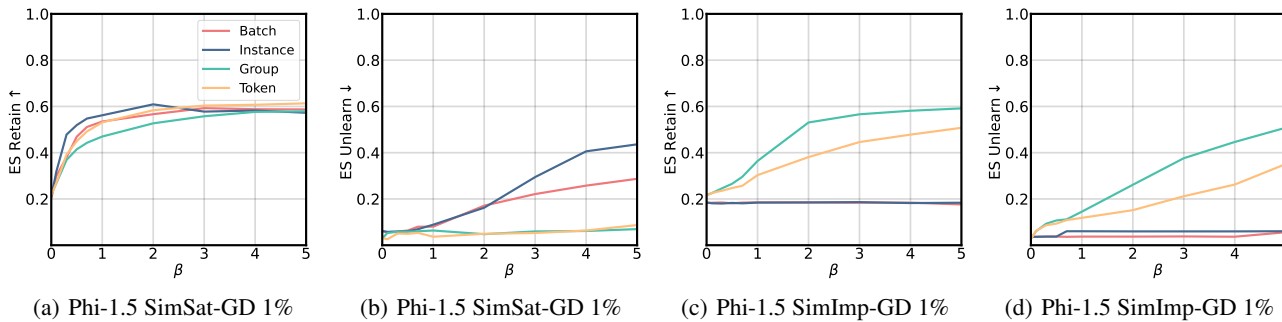

(a) Phi-1.5 SimSat-GD 1%   (b) Phi-1.5 SimSat-GD 1%   (c) Phi-1.5 SimImp-GD 1%   (d) Phi-1.5 SimImp-GD 1%

*Figure 4.* Comparisons of different weight granularity. Finer-grained reweighting strategies typically achieve a better unlearn-retain trade-off under the SimSat method. Meanwhile, under the SimImp method, fine-grained strategies lead to improvements on retention tasks, albeit at the cost of reduced unlearning performance. ES scores for targeted (ES Unlearn) and untargeted (ES Retain) are reported.

proposed in Eq. (11). Therefore, we empirically conclude that for LLM unlearning, soft reweighting proves to be more effective than hard sampling.

**Granularity of Reweighting.** Previous works (Wang et al., 2025b) have shown that token-wise reweighting typically surpasses instance-wise reweighting in terms of saturation. Here, we extend their research to cover importance-based reweighting and additional levels of granularity. We first extend the granularity to include token-wise, group-wise, instance-wise, and batch-wise units. Specifically, instance- and token-wise weights follow the formulations of Eq. (3) and Eq. (4); group-wise units first rank tokens in each data point in descending order based on weights, then averaging within groups and taking averaged values as the weights for each respective group; batch-level units averages instance-wise weights within each mini-batch, with the resulting average serving as the weight for the entire batch. We summarize our results in Figures 4(a)-4(d). We observe that SimSat with varying granularity can enhance retention as $\beta$ increases, while coarse-grained approaches, e.g., instance-wise and batch-wise reweighting, generally result in a notable decline in unlearning efficiency. Similarly, adjusting $\beta$ for SimImp also improves retention for fine-grained strategies but is accommpanied by a significant decline in the power of removal. Overall,finer-grained strategies result in better unlearning performance, and thus we still suggest the reweighting strategies following Eq. (11).

## 5. SatImp: An Improved Reweighting Method

Based on our comprehensive analyses of loss reweighting in Sections 3-4, we introduce an improved reweighting method named Saturation-Importance (**SatImp**). It is a soft-reweighting, token-wise approach that offers is flexible control over the smoothness. Specifically, the weight function of SatImp is defined as:

$$w_{x,y,k}^{\text{satimp}} = p(y^k|y^{<k}, x; \boldsymbol{\theta})^{\beta_1} \cdot (1 - p(y^k|y^{<k}, x; \boldsymbol{\theta}))^{\beta_2},$$
(12)

where $\beta_1, \beta_2$ are hyper-parameters to control the overall weight distribution. Compared to methods like importance

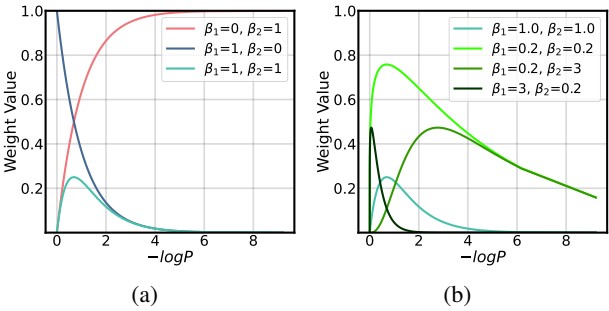

(a)          (b)

*Figure 5.* The correspondence of weight and loss values. (a) SatImp (Green) emphasize the middle-loss tokens in reweighting, which is different from previous importance-based (red) and saturation-based (blue) methods. Besides, (b) SatImp is more flexible in weight allocation, resulting in diverse reweighting strategies.

and saturation that focus on extreme loss values, SatImp emphasizes middle-loss tokens as shown in Figure 5(a): The values of $\beta_1$ and $\beta_2$ influence the scale of the weight distribution, with smaller $\beta_1$ and $\beta_2$ resulting in larger scale. Additionally, their relative magnitudes determine the tendency of the weight distribution, with larger $\beta_1$ favoring saturation and larger $\beta_2$ favoring importance.

**Experiments.** We show the superior performance of SatImp on the TOFU benchmark in Table 2, where we employ retain regularization following Eq. (2) to stabilize the unlearning process. The hyper-parameter for SatImp is set to $\beta_1 = 5$, $\beta_2 = 1$. More setting details for SatImp and counterparts are shown in appendix B.3. In addition to the ES score for retain (ES Re.) and unlearn (ES Un.) tasks, we also report the results of Forget Quality (FQ) and Model Utility (MU). We expect both FQ and MU to achieve high values. However, as discussed in existing literature (Wang et al., 2025b), FQ is not entirely suitable for LLM unlearning tasks. The FQ value only reflects the difference between the outputs of current model and the gold-standard model, while numerous studies indicate that even with a low FQ value, the model can still exhibit excellent unlearning performance (Shi et al., 2024; Fan et al., 2024).

*Table 2.* Comparison between unlearning objectives on TOFU with retain regularization to stabilize unlearning. ↑ / ↓ indicate larger / smaller values are preferable. The top two results are in **bold** font for each unlearning setup. The results with FQ are not highlighted, as it is a less meaningful metric that deviates the unlearning goals of LLMs.

| Method | Forget-1% | | | | Forget-5% | | | | Forget-10% | | | |
|---|---|---|---|---|---|---|---|---|---|---|---|---|
| | ES Re. ↑ | ES Un. ↓ | FQ ↑ | MU ↑ | ES Re. ↑ | ES Un. ↓ | FQ ↑ | MU ↑ | ES Re. ↑ | ES Un. ↓ | FQ ↑ | MU ↑ |
| Phi-1.5 | | | | | | | | | | | | |
| Original | 0.6857 | 0.6569 | 4.2e-5 | 0.5223 | 0.6912 | 0.7117 | 3.7e-14 | 0.5223 | 0.6857 | 0.7127 | 1.2e-17 | 0.5223 |
| GA | 0.2098 | **0.0322** | 0.7659 | 0.4334 | 0.1386 | **0.0001** | 7.5e-5 | 0.2660 | 0.1316 | **0.0305** | 2.6e-9 | 0.3912 |
| PO | 0.4951 | 0.2867 | 0.2656 | 0.4626 | 0.4908 | 0.4400 | 2.4e-10 | 0.4576 | **0.5426** | 0.4993 | 1.9e-15 | 0.4745 |
| DPO | 0.3897 | 0.1790 | 0.4046 | 0.4406 | 0.3057 | 0.2596 | 2.6e-7 | 0.4423 | 0.4521 | 0.4060 | 2.9e-14 | 0.5041 |
| NPO | 0.4514 | 0.0583 | 0.1650 | 0.4973 | 0.3936 | 0.0787 | 1.1e-5 | 0.5116 | 0.4693 | 0.0990 | 5.4e-8 | 0.4988 |
| SimNPO | 0.5557 | 0.1397 | 0.0143 | 0.5188 | 0.4983 | 0.1290 | 8.1e-8 | **0.5181** | 0.4697 | 0.1216 | 5.4e-8 | **0.5076** |
| WGA | **0.5835** | 0.0494 | 0.0971 | **0.5248** | **0.5219** | **0.0374** | 0.0396 | 0.5166 | 0.4655 | **0.0282** | 0.0017 | 0.5023 |
| SatImp | **0.6079** | **0.0464** | 0.0971 | **0.5248** | **0.5410** | 0.0427 | 1.8e-4 | **0.5214** | **0.4706** | 0.0407 | 1.2e-8 | **0.5107** |
| LLaMA2-7B | | | | | | | | | | | | |
| Original | 0.8529 | 0.8550 | 5.1e-7 | 0.6345 | 0.8529 | 0.8476 | 1.1e-14 | 0.6345 | 0.8529 | 0.8610 | 4.5e-18 | 0.6345 |
| GA | 0.4824 | 0.0941 | 0.4046 | 0.5651 | 0.2058 | **0.0000** | 2.9e-11 | 0.5321 | 0.2099 | **0.0000** | 2.6e-30 | 0.5197 |
| PO | 0.7163 | 0.4964 | 0.1650 | 0.5256 | 0.6727 | 0.6014 | 6.6e-12 | 0.5473 | 0.6347 | 0.6075 | 5.1e-17 | 0.5176 |
| DPO | 0.6227 | 0.3970 | 0.0971 | 0.5997 | 0.4936 | 0.3897 | 4.9e-10 | 0.5613 | **0.6645** | 0.6191 | 7.4e-15 | 0.6169 |
| NPO | 0.6692 | 0.1064 | 0.5786 | **0.6278** | 0.5791 | 0.1090 | 0.0521 | 0.6171 | 0.6420 | 0.0954 | 0.0162 | 0.6143 |
| SimNPO | **0.7835** | 0.2869 | 0.0030 | 0.6173 | **0.6761** | 0.2015 | 1.2e-4 | 0.6051 | 0.6010 | 0.1698 | 5.1e-6 | 0.6068 |
| WGA | 0.7603 | **0.0545** | 0.2659 | 0.6256 | 0.6649 | **0.0002** | 8.1e-8 | **0.6198** | 0.6471 | **0.0033** | 5.1e-17 | **0.6299** |
| SatImp | **0.8034** | **0.0579** | 0.2657 | **0.6315** | **0.6884** | 0.0027 | 2.2e-17 | **0.6212** | **0.6669** | 0.0148 | 2.5e-18 | **0.6315** |

*Table 3.* Comparison between unlearning objectives on WMDP with forget-only regularization. ↑ / ↓ indicate larger / smaller values are preferable. Top two results are in **bold** font.

| Method | Unlearn | | Retain |
|---|---|---|---|
| | WMDP-Bio ↓ | WMDP-Cyber ↓ | MMLU ↑ |
| Original | 0.6462 | 0.3924 | 0.5853 |
| GA | 0.2474 | **0.2431** | 0.2465 |
| NPO | 0.5260 | 0.4616 | **0.5607** |
| SimNPO | 0.3519 | 0.3562 | **0.3418** |
| WGA | **0.2467** | 0.2617 | 0.2963 |
| RMU | 0.2479 | 0.2963 | 0.3193 |
| SatImp | **0.2470** | **0.2456** | 0.3197 |

**Analysis.** Across the baseline methods, PO is the least effective one, with notable lower unlearning task completion rates (high ES Un. values) across diverse settings. While Vanilla GA often achieves strong unlearning performance, it comes at the cost of over-forgetting, reflected by its low ES Re. and MU values. In contrast, recent methods like NPO, SimNPO, and WGA strike a better balance, excelling in both unlearning effectiveness and retention.

For the proposed SatImp, we find that it demonstrates outstanding performance on TOFU. Specifically, in the retention task, SatImp outperforms all competitors, whether evaluated by the ES Re. or MU. In the unlearning task, ES-based evaluation indicates that SatImp ranks among the top in unlearning performance. When analyzing FQ, we find that our previous conclusions typically hold. Methods with high-FQ values, such as DPO and NPO, typically have a higher ES Un. values than SatImp. Jointly considering the results

of ES Un. and FQ, we find that SatImp largely facilitate the unlearning process on both LLaMA-2-7B and Phi-1.5, indicating that the above scenario in exceeding the gold standard occurs. Furthermore, we extend our exploration to the WMDP benchmark and are surprised to find that SatImp exhibits exceptional unlearning performance. The NPO method, which achieved a high FQ value on TOFU, turns out to be an example of ineffective unlearning on WMDP, further validating our analysis. Due to the space limit, we present more results and details in Appendix E.

## 6. Conclusion

In this paper, we systematically investigated the role of loss reweighting in LLM unlearning by identifying two distinct goals: Saturation and Importance. Through extensive experiments on established benchmarks, we found that saturation-based reweighting is generally more effective than importance-based strategies, and their combination provides additional improvements. Moreover, our investigation into specific reweighting operations revealed that the smoothness and granularity of weight distributions significantly influence unlearning performance. Based on these insights, we proposed SatImp, a simple yet effective reweighting method that integrates the strengths of both saturation and importance. Our work offers valuable insights into existing reweighting strategies, advancing the understanding of their mechanisms and providing analyses about specific reweighting operational details for the development of controllable and stable unlearning methodologies.

## Impact Statement

Unlearning mechanisms are essential for LLMs, as they enable the removal of sensitive data that could lead to privacy breaches or copyright violations, thereby enhancing the overall security of these models. By ensuring the ethical obligation to respect individual privacy, we reinforce responsible data usage and prevent the replication of sensitive information. Moreover, unlearning improves societal well-being by strengthening legal compliance and fostering public trust in AI technologies. It also enhances model controllability, allowing developers to manage knowledge boundaries effectively and mitigate risks associated with misuse, such as fraud or the spread of misinformation. In this paper, we benefit the research community by presenting a comprehensive study about the reweighting mechanism in LLM unlearning. While previous studies often lack a clear understanding of their own methodological positioning, we explicitly classify them into two distinct types: importance-based and saturation-based reweighting. Furthermore, while many works claim that controlling the unlearning process is challenging, our in-depth study of specific operations within the unlearning process clarifies these uncertainties. Based on these investigations, we contribute to the development of a more controllable and stable LLM unlearning framework. For the sake of reproducibility, we have meticulously documented the experimental configurations, hyper-parameter setups, and hardware specifications.

## Acknowledgements

PNY, QZW, and BH were supported by RGC Young Collaborative Research Grant No. C2005-24Y, NSFC General Program No. 62376235, Guangdong Basic and Applied Basic Research Foundation Nos. 2022A1515011652 and 2024A1515012399, HKBU Faculty Niche Research Areas No. RC-FNRA-IG/22-23/SCI/04, and HKBU CSD Departmental Incentive Scheme. ZH and TLL were partially supported by the following Australian Research Council projects: FT220100318, DP220102121, LP220100527, LP220200949, and IC190100031.

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

# A. Limitations

Although our work has conducted an in-depth investigation into the reweighting-based unlearning mechanism, we have yet to explore it from a more theoretical analytical perspective. Furthermore, while the proposed SATIMP method achieves a better unlearn-retain balance, it remains parameter-sensitive, similar to previous approaches. Addressing this issue will require further investigation into the inherent limitations of the reweighting mechanism itself, as well as the development of new mechanism to achieve a more robust and stable unlearning process.

# B. Benchmarks and Experiment Setup

## B.1. Benchmarks and Metrics

In this paper, we adopt three of the most popular benchmarks in LLM unlearning. Benchmark Details and corresponding metrics are introduced as follows.

### B.1.1. TOFU BENCHMARK

**TOFU Dataset (Maini et al., 2024).** The TOFU dataset comprises 4,000 question-answer pairs about 200 fictional authors, with each author assigned 20 QA pairs. Regarding specific unlearn tasks, TOFU establishes three settings: 1% (2 authors, 40 QApairs), 5% (10 authors, 200 QA pairs), and 10% (20 authors, 400 QA pairs). TOFU Authors provide fine-tuned models of Llama2-7B and Phi-1.5 on the full TOFU dataset. Additionally, the authors provide fine-tuned results of Llama2-7B and Phi-1.5 models on the Retain data to support subsequent performance evaluation.

**Forget Quality.** Forget Quality is a metric that measures the similarity between two distributions: the truth ratio of the retained model (pre-trained on the retain set) and the truth ratio of the unlearned model on the forget set. It is represented by calculating the p-value from a Kolmogorov-Smirnov (KS) test.

**Model Utility.** Model Utility is a comprehensive metric that considers the model's performance across different retain tasks. Specifically, it is expressed as the harmonic mean of nine numbers: the answer probability, truth ratio, and ROUGE recall scores for the retain, real authors, and world facts subsets.

**ES Score.** To quantify memory strength, recent literature (Wang et al., 2025a) proposes a more intuitive metric: Extraction Strength (ES), defined as the minimum prefix ratio required for restoring the suffix:

$$\mathrm{ES}(x, y, k) = 1 - \frac{1}{|y|} \min_{k} \{k | f(y^{<k}|x; \theta) = y^{>k}\} \tag{13}$$

where $k$ is the number of the answer tokens in the prefix, and $y$ is the number of answer tokens.

### B.1.2. WMDP BENCHMARK

**WMDP Dataset (Li et al., 2024).** The Weapons of Mass Destruction Proxy (WMDP) benchmark contains harmful contents about biology (1.27k lines), cyber-security (1.99k lines), and chemistry (408 lines) that could be used to create bioweapons or launch cyberattacks. Correspondingly, the retain set of WMDP comes from MMLU benchmark (Hendrycks et al., 2021a;b). Both WMDP and MMLU are multiple-choice QA datasets. The pre-trained model that provided by WMDP authors is Zephyr-7B-beta.

**Accuracy.** The evaluation metric for WMDP and MMLU benchmarks is the accuracy among these multiple-choice questions.

### B.1.3. MUSE BENCHMARK

**MUSE Dataset (Shi et al., 2024).** The Machine Unlearning Six-Way Evaluation (MUSE) for Language Models benchmark consists of the News and Books subsets, which extract data from BBC News and the Harry Potter series respectively. LLaMa-2-7B and ICLM-7B are pre-trained models in this benchmark. To more comprehensively evaluate the model's unlearning attributes, MUSE proposes six metrics: Verbatim Memorization, Knowledge Memorization, Privacy Leakage, Utility Preservation, Scalability, and Sustainability. In this paper, we utilize the first four metrics.

**Verbatim Memorization (VerbMem).** Considering a model $\theta$ with the first $k$ tokens from a sample $x$, the ROUGE-L value

between the output of $f(x^{[:k]}; \theta)$ and the true continuation $x^{[k+1:]}$ is defined as the VerbMem:

$$\text{VerbMem}(\theta, \mathcal{D}_{\text{u}}) = \frac{1}{|\mathcal{D}_{\text{u}}|} \sum_{(x) \in \mathcal{D}_{\text{u}}} \text{ROUGE}(f(x^{[:k]}; \theta), x^{[k+1:]}) \tag{14}$$

**Knowledge Memorization (KnowMem).** Different from VerbMem, KnowMem considers from a instance-wise perspective. Let $(q, a)$ represents a pair of question and answer in one sample $x$, the KnowMem is defined as:

$$\text{KnowMem}(\theta, \mathcal{D}_{\text{u}}) = \frac{1}{|\mathcal{D}_{\text{u}}|} \sum_{(q,a) \in \mathcal{D}_{\text{u}}} \text{ROUGE}(f(q; \theta), a) \tag{15}$$

**Privacy Leakage (PrivLeak).** It is essential that the unlearned model does not leak membership details that could reveal $\mathcal{D}_{\text{u}}$ is part of $\mathcal{D}_{\text{tr}}$. Therefore, to accurately measure the degree of leakage, PrivLeak is proposed by employing Min-K% Prob (Shi et al., 2023) and computing the standard AUC-ROC score (Murakonda et al., 2021; Ye et al., 2022) as follows:

$$\text{PrivLeak} := \frac{\text{AUC}(\theta_{\text{u}}; \mathcal{D}_{\text{u}}, \mathcal{D}_{\text{h}}) - \text{AUC}(\theta_{\text{r}}; \mathcal{D}_{\text{u}}, \mathcal{D}_{\text{h}})}{\text{AUC}(\theta_{\text{r}}; \mathcal{D}_{\text{u}}, \mathcal{D}_{\text{h}})}, \tag{16}$$

where $\theta_{\text{r}}$ is the retrained model learning on the retain dataset, $\theta_{\text{r}}$ is the unlearned model. $\mathcal{D}_{\text{h}}$ denotes a 'holdout' dataset which contains in-distribution samples but model has not been trained on. A well-performing unlearning algorithm should have a PrivLeak metric close to zero, whereas an over- or under-unlearning algorithm will result in a high positive or negative value.

**Utility Preservation (UtilPres).** Except for metrics on unlearn tasks, performance on other tasks are also concerned in MUSE. Specifically, UtilPres is represented as the evaluation on retain dataset with the KnowMem metric.

$$\text{UtilPres}(\theta, \mathcal{D}_{\text{r}}) = \frac{1}{|\mathcal{D}_{\text{r}}|} \sum_{(q,a) \in \mathcal{D}_{\text{r}}} \text{ROUGE}(f(q; \theta), a) \tag{17}$$

### B.2. Experiment Setup

**Environmental Configurations.** Our experiments are completed on 8 NVIDIA A100 GPUs, with Python 3.10.14 and PyTorch 2.1.0.

**TOFU experiment setup.** Following the common practice on TOFU benchmark, we utilize a linear warm-up learning rate during the first epoch, followed by a linearly decaying learning rate in the remaining epochs. Before the unlearning process, we fine-tune the LLaMA-2-7B model on the TOFU full set for 5 epochs with a batch size of 32 and a learning rete $1e - 5$ to obtain the original model. Phi-1.5 is also fine-tuned with a differnt learning rate $2e - 5$. For three unlearning settings: Forget01, Forget05, and Forget10, we train both LLaMA-2-7B and Phi-1.5 for 5 epochs with a batch size of 16. The learning rates for LLaMA-2-7B and Phi-1.5 are set to $1e - 5$ and $2e - 5$, respectively.

**WMDP experiment setup.** We utilize the official provided Zephyr-7B-beta model in the WMDP benchmark. The total training step is set to 125 steps for all unlearning objectives. The learning rate is set to $4e - 6$, which is also a linear warm-up learning rate during the first 25 step, followed by a linearly decaying learning rate in the remaining steps. The batch size is set to 16.

**MUSE experiment setup.** We utilize the official provided LLaMA-2-7B for the News task, and the ICLM-7B for the Books task. Each model is trained for 10 epochs with a constant learning rate $1e - 5$ and a batch size of 16. Results are selected from 10 checkpoints saved at each epoch.

### B.3. Methods Setup

In this paper, we mainly consider the following baselines: GA (Maini et al., 2024), PO (Rafailov et al., 2024), DPO (Rafailov et al., 2024), NPO (Zhang et al., 2024), SimNPO (Fan et al., 2024), WGA (Wang et al., 2025b), RMU (Li et al., 2024). In the main text, we have already introduced the GA, NPO, and WGA methods; here, we present additional approaches. Furthermore, we introduce the parameter settings for these baseline methods.

**GA.** GA has been introduced, here we introduce the parameter setting when the retain regularization (Eq. (2)) is utilized: $\lambda$ is set to 1 for GA.

**PO.** Preference Optimization (PO) is a method which trains model to respond "I don't know" to unlearn samples. Let $\mathcal{D}_{\text{idk}}$ denotes a an augmented forget dataset where the model's response to the question is 'I don't know.', the objective of PO is defined as:

$$\min_{\boldsymbol{\theta}} \mathbb{E}_{(x,y)\sim\mathcal{D}_{\text{idk}}} \log p(y|x;\boldsymbol{\theta}). \tag{18}$$

We set $\lambda = 1$ for PO when utilize the retain regularization (Eq. (2)).

**DPO.** Direct Preference Optimization (DPO) aims to align the unlearn model to a reference model $\boldsymbol{\theta}_{\text{ref}}$ with standard answers $y^e$, its objective can be defined as:

$$\mathbb{E}_{(x,y)\sim\mathcal{D}_{\text{u}}}\left[ - \frac{2}{\beta} \log \sigma(\beta \log \left(\frac{p(y^e|x;\boldsymbol{\theta})}{p(y^e|x;\boldsymbol{\theta}_{\text{ref}})}\right) - \beta \log \left(\frac{p(y|x;\boldsymbol{\theta})}{p(y|x;\boldsymbol{\theta}_{\text{ref}})}\right))\right] \tag{19}$$

For the parameter $\beta$, we search in a range [0.2, 0.5]. $\lambda$ is set to 1 for DPO when the retain regularization (Eq. (2)) is utilized.

**NPO.** NPO has been introduced, here we introduce the parameter setting: For the parameter $\beta$, we search in a range of [0.05, 0.2] due the recommendation in the NPO paper ($\beta = 0.1$). Besides, $\lambda$ is searched in a range of [0.2, 0.5] when the retain regularization (Eq. (2)) is utilized.

**SimNPO.** Simplify NPO (SimNPO) addresses NPO's ineffective gradient weight smoothing problem by introducing a constraint based on answer length:

$$\mathbb{E}_{(x,y)\sim\mathcal{D}_{\text{u}}}\left[ - \frac{2}{\beta} \log \sigma(-\frac{\beta}{|y|} \log (p(y|x;\boldsymbol{\theta}) - \gamma)\right] \tag{20}$$

when analyzing the gradients of SimNPO, it can be represented as the reweighting formulation of GA with a weight:

$$w_{x,y}^{\text{simnpo}} = \frac{2p(y|x;\boldsymbol{\theta})^{\frac{\beta}{|y|}}}{p(y|x;\boldsymbol{\theta})^{\frac{\beta}{|y|}} + 1} \cdot \frac{1}{|y|}. \tag{21}$$

According to the recommendation in the SimNPO paper ($\beta = 2.5, \lambda = 0.1375$), we search $\beta$ and $\lambda$ and in a range of [2.0, 3.0] and [0.2, 0.3], respectively.

**WGA.** WGA has been introduced, here we introduce the parameter setting: We have presented a mount of experiments in Sec. 4.1 with WGA. $\lambda$ is set to 1 for WGA when the retain regularization (Eq. (2)) is utilized. $\beta$ is set to 2 for WGA due to the consideration of balance between unlearning and retention.

**RMU.** Representation Misdirection for Unlearning (RMU) achieves unlearning by perturbing the model's representation of the target data with random noise. Let $\phi(x, y; \theta)$ represent the embedding features, the RMU objective can be defined as:

$$\mathbb{E}_{(x,y)\sim\mathcal{D}_{\text{u}}} \frac{1}{|y|-1} \sum_{k}^{|y|-1} ||\phi(y^k|y^{<k}, x;\boldsymbol{\theta}) - \beta \cdot u||_2^2,, \tag{22}$$

where $u$ is a random vector with elements sampled from [0,1) and $\beta$ is a scaling hyper-parameter. Recent literature (Wang et al., 2025b) has presented a comprehensive analysis of RMU. Considering its moderate performance on the TOFU benchmark, we only compare RMU on the WMDP benchmark, which is its strongest task. Hyper-parameters in RMU is set as the paper recommended ($\beta = 6.5$, the 7-th layer is choosen for the unlearning process).

**SatImp.** The proposed SatImp has tow hyper-parameters $\beta_1$, $\beta_2$. We present ablation study about them with Phi-1.5 (on TOFU 1%, 5%, 10% setting) and LLaMA-2-7B (on TOFU 1%) models. **Due to the space limit, we present the hyper-parameter ablation study in https://github.com/tmlr-group/SatImp.**

## C. More Details During Training

### C.1. Supplemental Analysis for Sec. 4.1

As mentioned in the main text, we provide more details about the correlation between weight and loss during training. Here we provide more details during training, first, we provide the allocation details as same as Figure 2(e). As shown in

Figure 6, we conduct that key annotations is high-correlated to tokens with lower likelihoods (higher negative log likelihood values). The weights of NPO and SimNPO are instance-wise, which tends to allocate large weights to tokens with higher likelihoods (Figure 7). Weight details of WGA and proposed saturation-based reweighting are also provided in Figure 8-9. WGA converges faster, as evidenced by more weights approaching zero (indicating completed unlearning) at the same training step.

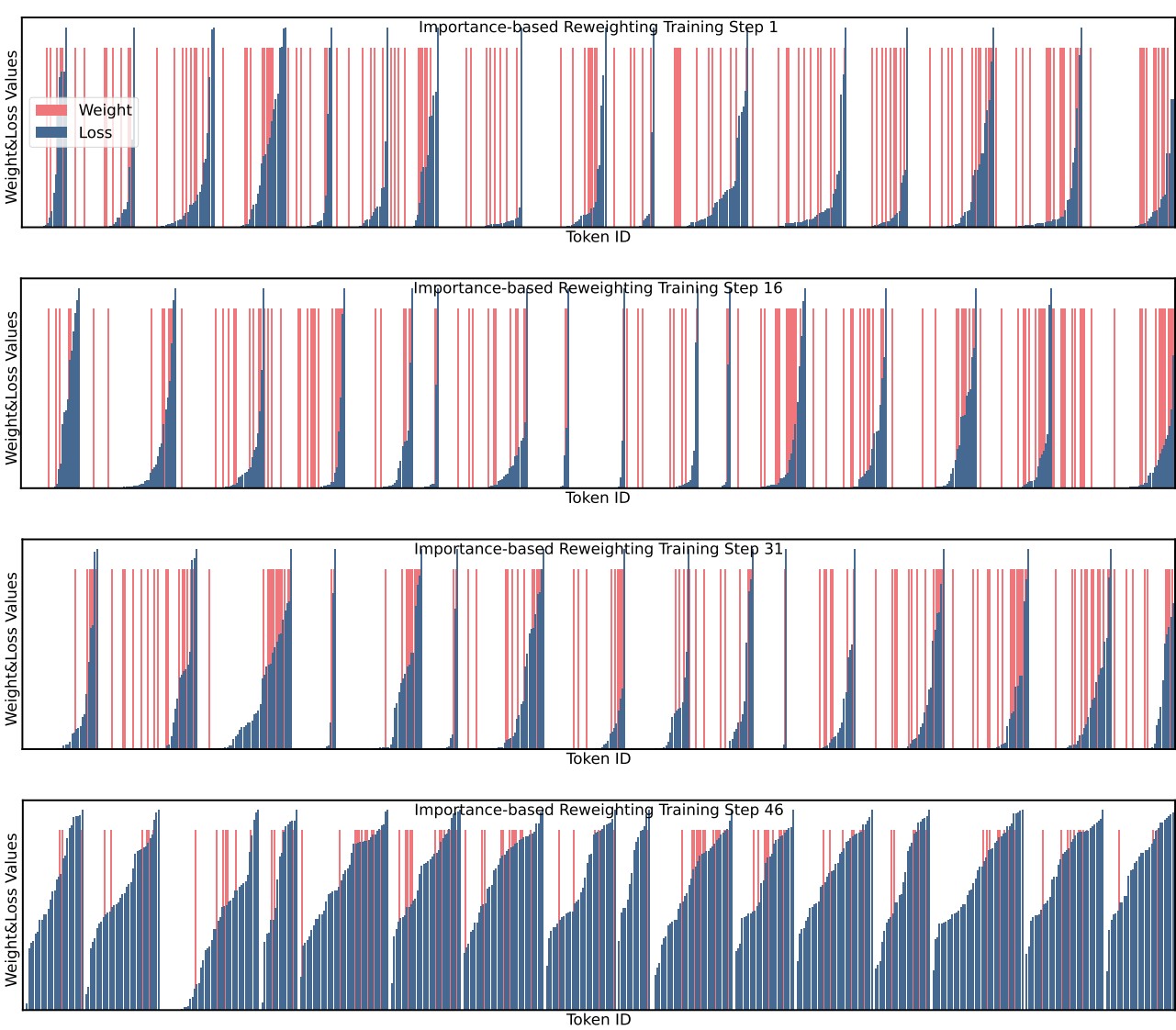

*Figure 6.* Loss-weight details for importance-based reweighting.

## C.2. VisE: Towards Smooth Training Process and Better Unlearning Model

In this section, we analyze the fluctuation in loss and gradient during training, resulting in a three-stage smooth training process. By evaluate the correlation between training metrics and model performance, we show that early-stopping strategies can be effectively utilized to achieve a better-performing model.

Despite the fact that the performance of sampling methods does not match that of weighting, we unexpectedly derived some interesting conclusions during training. As shown in Figure 10(a), 10(b), we observed that utilizing the BottomK strategy significantly mitigates fluctuations in both the gradient and loss terms during vanilla GD training. This results in a smoother training process and a clear emergence of a three-stage progression. Specifically, the sharp changes in the gradient serve

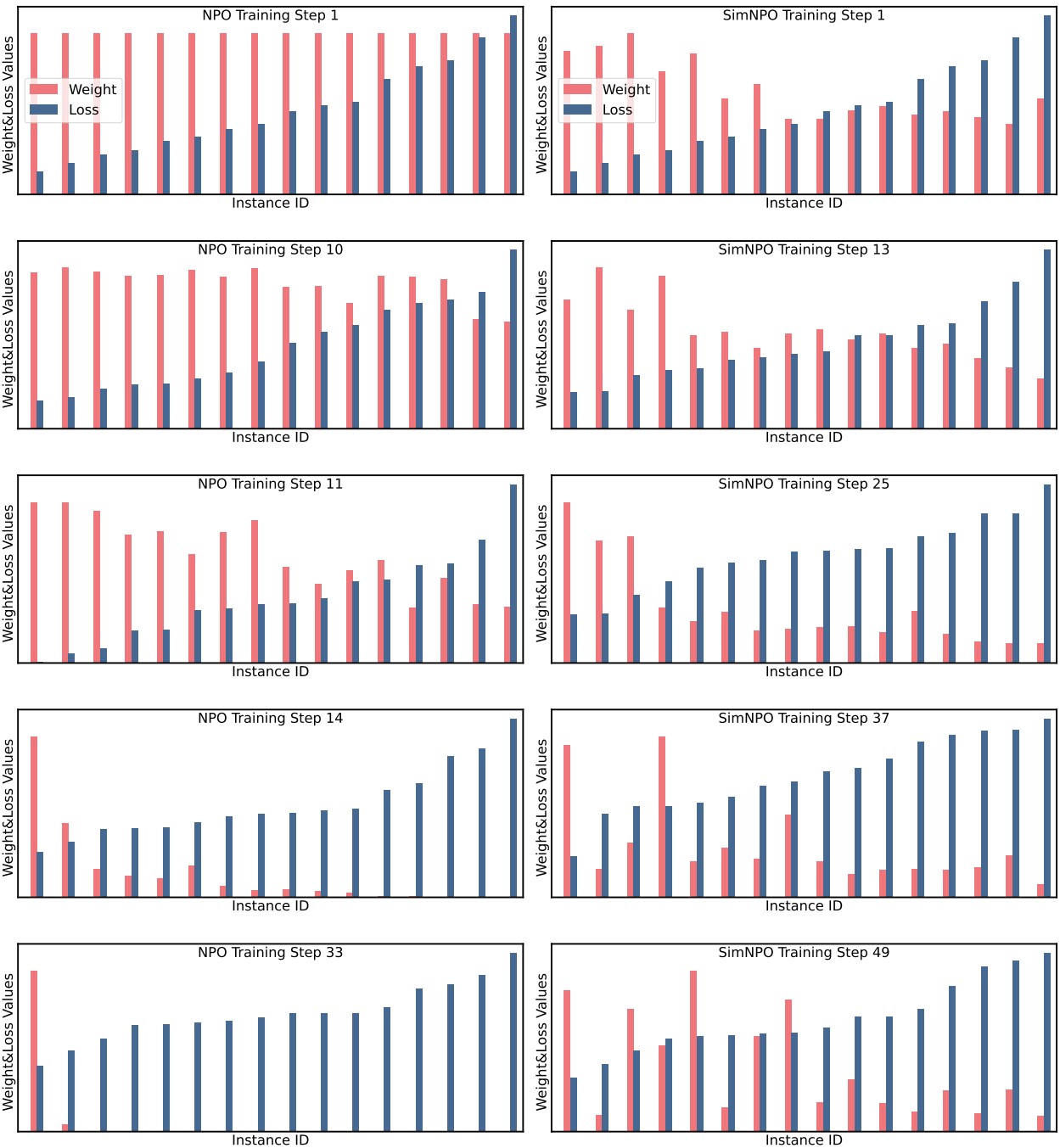

*Figure 7.* Loss-weight details for NPO and SimNPO methods.

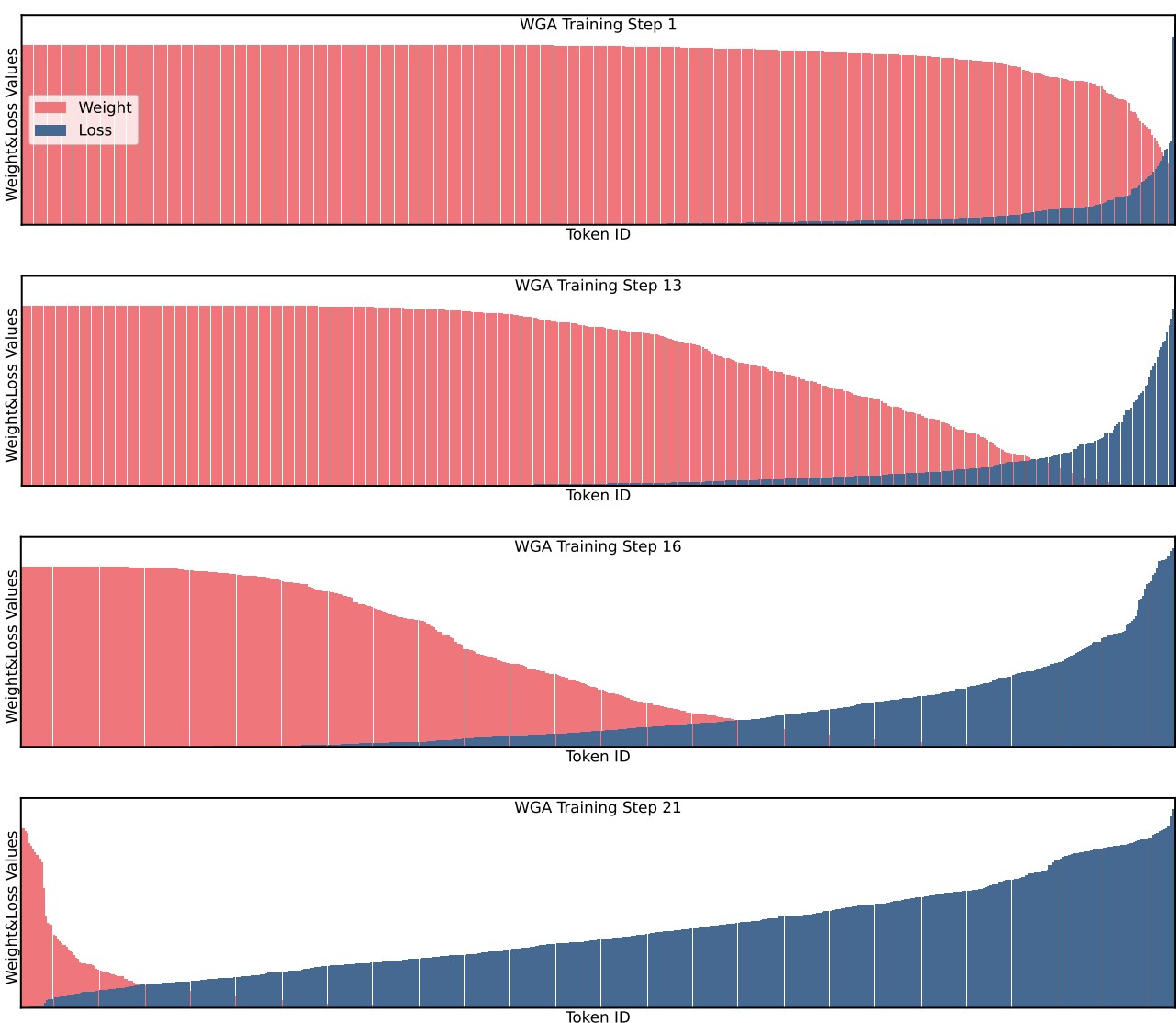

*Figure 8.* Loss-weight details for WGA.

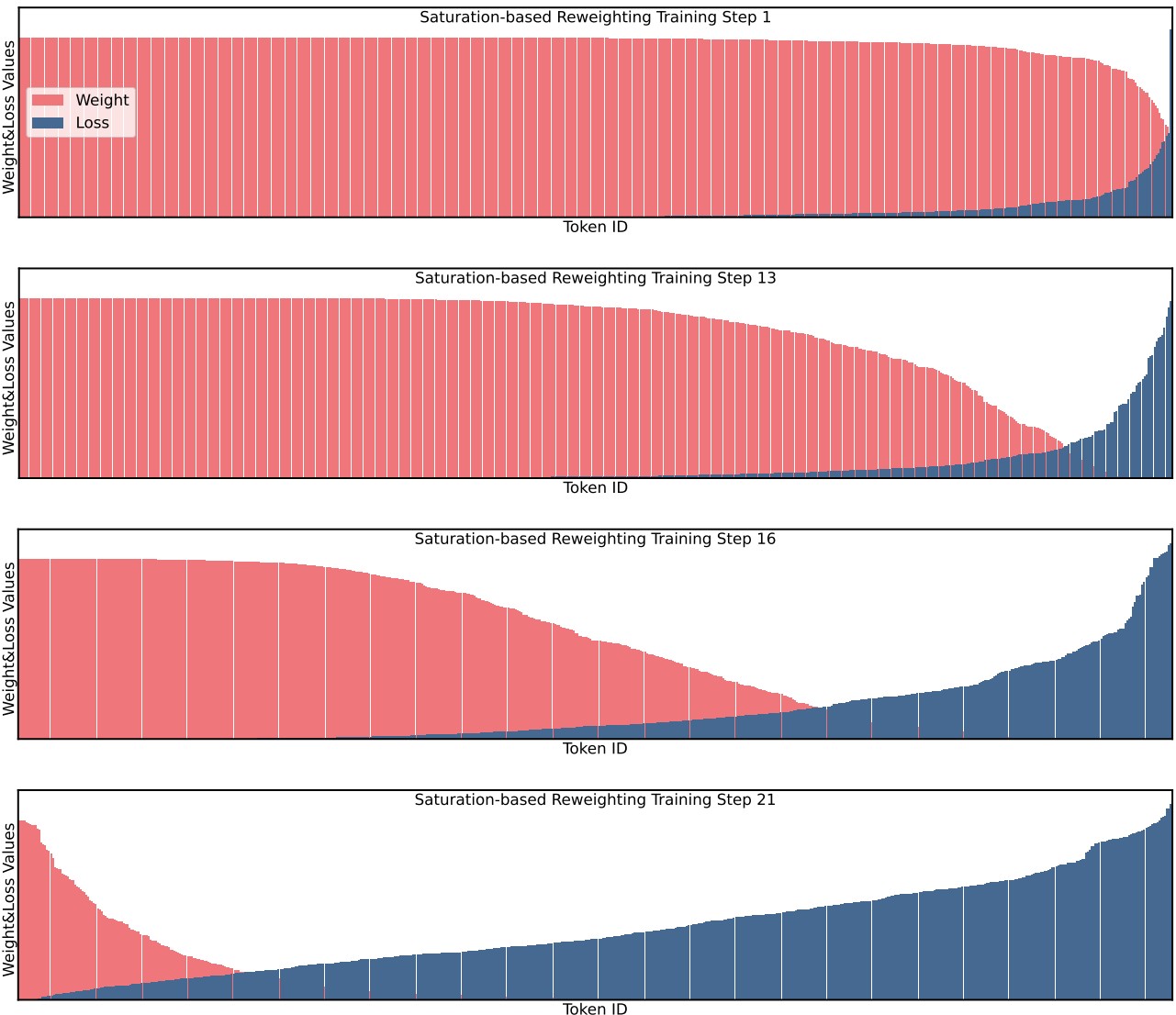

*Figure 9.* Loss-weight details for saturation-based reweighting.

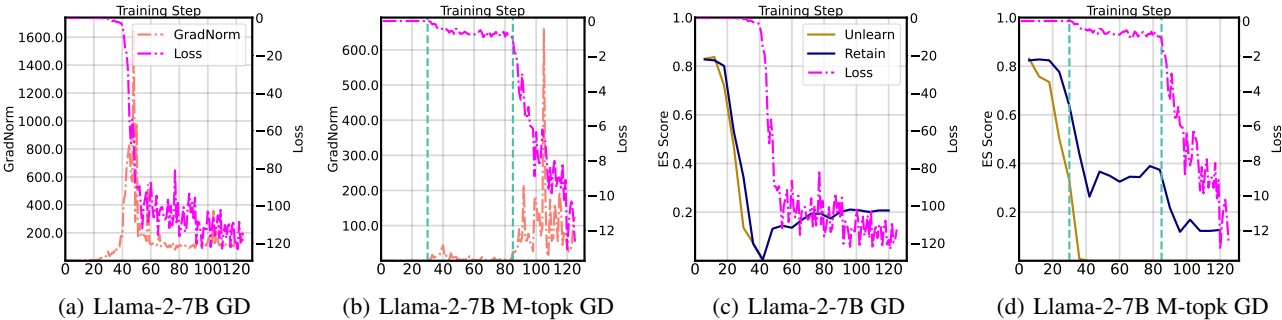

(a) Llama-2-7B GD      (b) Llama-2-7B M-topk GD      (c) Llama-2-7B GD      (d) Llama-2-7B M-topk GD

*Figure 10.* Enhancing LLM unleanring via early-stopping based on visualized training process. While hard-sampling methods are effective, they still underperform soft-weighting methods. Notably, the BottomK strategy enables a smoother training process with three distinct phases. Furthermore, the variations in training metrics are highly correlated with the model's performance, resulting in an early-stopping strategy based on the visualized training process, which can lead to models with better performance.

as boundaries, with the loss stabilizing in the first two stages and continuously declining in the third stage. Such a clear training process aids in predicting and assessing the statement of model.

Furthermore, we aim to investigate the correlation between model performance and these training metrics, which is illustrated in Figure 10(b), 10(d). Based on the performance in different stages, we name the three stages as: unlearning, stabilization, and collapse. In the unlearning stage, the model rapidly forgets the target data. In the stabilization stage, it mitigates the loss of retained information caused by prior unlearning. In the post-training phase, the model further unlearns, entering a collapse stage where it also unlearns the information that should been retained.

Based on aforementioned observations, we indicate that unlearning process should be early-stopped during the stabilization stage. The model has empirically completed the unlearning task during the stabilization stage, attempting to optimize performance on the retain task without compromising unlearning. In the subsequent collapse stage, although the training metrics show further loss increases—indicating intensified forgetting—existing metrics reveal that this forgetting mainly harms the model's utility.

# D. Related Work

## D.1. Machine Unlearning

Standard machine unlearning (Bourtoule et al., 2021) guarantees the "right to be forgotten" (Pardau, 2018; GDPR, 2016), enabling individuals to request the removal of their data from the machine learning service provider. The optimal solution for machine unlearning is retraining model from scratch after removing the unlearn data from training dataset (Liu et al., 2024b; Fan et al., 2025), which is often considered as the gold standard. However, cleaning such data can be very labor-intensive and inflexible, resulting in a large cost in computation and time. To address this issue, most research focus on approximate unlearning methods, including gradient ascent (Graves et al., 2021; Thudi et al., 2022) and various optimizations from different perspective: data selection (Izzo et al., 2021), feature modification (Golatkar et al., 2020; Liu et al., 2024b), loss design (Adolphs et al., 2022; Wang et al., 2023a; Di et al., 2024; Fan et al., 2023) and so on.

## D.2. LLM Unlearning

In the past few years, large language models (LLMs) (Touvron et al., 2023; Achiam et al., 2023; Jiang et al., 2023; Bao et al., 2024) have shown notable achievements across numerous downstream tasks. The efficacy of LLMs is largely dependent on the massive training process, which involves millions of training samples (Wang et al., 2022; Liu et al., 2023; 2024c). However, such vast training data often has flawed human annotations (Liu & Guo, 2020) or noisy (potentially harmful) web content (Wei et al., 2021; Zhu et al., 2023), making data curation particularly challenging. Such issues are also widespread in various traditional machine learning fields (Wu et al., 2022; Yu et al., 2023b; Huang et al., 2023; Wang et al., 2023b; 2024a). Consequently, the output of LLM often contains sensitive (Patil et al., 2023), private (Karamolegkou et al., 2023), prejudiced (Yu et al., 2023a; Motoki et al., 2024), or harmful (Barrett et al., 2023; Li et al., 2024) contents, motivating researchers to explore efficient unlearning on LLMs.

Existing LLM unlearning explorations can be primarily categorized into parameter-fixed (Thaker et al., 2024; Pawelczyk et al., 2023; Muresanu et al., 2024; Bhaila et al., 2024; Gao et al., 2024) and parameter-modified (Jang et al., 2022; Eldan & Russinovich, 2023; Chen & Yang, 2023; Jia et al., 2024; Wang et al., 2025c; Wuerkaixi et al., 2025). Parameter-fixed paradigm (Gu et al., 2024; Choi et al., 2024; Liu et al., 2024a; Mekala et al., 2024), highly related to Chain-of-Thought methods (Wei et al., 2022; Yu et al., 2024), utilizes different prompts to guide LLMs towards the unlearning objective without altering the model's parameters. Parameter-modified paradigm typically fine-tunes the target LLM by modifying the loss function, usually containing two objectives: maximizing the loss on forget samples and minimizing the loss on retain samples. This paper is relevant to the parameter-modified part, which can also be summarized as the vanilla Gradient Ascent (GA) (Yao et al., 2023), GA variants with different weighting strategies (Ilharco et al., 2022; Chen & Yang, 2023; Huang et al., 2024a; Rafailov et al., 2024; Dong et al., 2024; Ji et al., 2024; Zhang et al., 2024; Fan et al., 2024; Maini et al., 2024; Jia et al., 2024; Huang et al., 2024b; Ethayarajh et al., 2024; Wang et al., 2024b). While GA demonstrates strong forgetting capabilities, it often leads to over-forgetting, which diminishes model's utility on other tasks. GA-based variants mitigates this issue by setting gradient upper bounds to promote early convergence, yet results in under-forgetting (Fan et al., 2024; Wang et al., 2025a;b) that leaves undesirable content in the model.

# E. More Experiment Results

As mentioned in the main text, many experimental results should be presented in the Appendix. First, we present more results compared with counterparts on TOFU, WMDP, and MUSE benchmarks.

As shown in Table 4, we employ forget regularization from Eq. (1) to evaluate the performance of SatImp. Results on the Forget 1% setting indicate that SatImp maintains its excellent performance, just as it does in Table 2. However, experiments on Forget 5% and 10% settings indicate that SatImp encounters the over-forgetting as same as GA and WGA, which reveals a common disadvantage of token-wise reweighting methods. As shown in Figure 16(i)-16(v), token-wise strategies typically result in significant over-forgetting (ES Re. values approaching 0), while instance-wise strategies under the SimSat method exhibit slight mitigation. Thus, we demonstrate that SatImp remains effective, but as a token-wise strategy, its performance is inevitably influenced by the characteristics of this mechanism.

*Table 4.* Comparison between unlearning objectives on TOFU with forget-only regularization. $\uparrow$ / $\downarrow$ indicate larger / smaller values are preferable. The top two results are in **bold** font for each unlearning setup. The results with FQ are not highlighted, as it is a less meaningful metric that deviates the unlearning goals of LLMs.

| Method | Forget-1% | | | | Forget-5% | | | | Forget-10% | | | |
|---|---|---|---|---|---|---|---|---|---|---|---|---|
| | ES Re. $\uparrow$ | ES Un. $\downarrow$ | FQ $\uparrow$ | MU $\uparrow$ | ES Re. $\uparrow$ | ES Un. $\downarrow$ | FQ $\uparrow$ | MU $\uparrow$ | ES Re. $\uparrow$ | ES Un. $\downarrow$ | FQ $\uparrow$ | MU $\uparrow$ |
| Phi-1.5 | | | | | | | | | | | | |
| Original | 0.6857 | 0.6569 | 4.2e-5 | 0.5223 | 0.6912 | 0.7117 | 3.7e-14 | 0.5223 | 0.6857 | 0.7127 | 1.2e-17 | 0.5223 |
| GA | 0.1823 | **0.0369** | 0.7660 | 0.3793 | 0.0000 | **0.0000** | 1.8e-5 | 0.0000 | 0.0000 | **0.0000** | 1.1e-13 | 0.0000 |
| DPO | **0.3753** | 0.1745 | 0.5786 | 0.3942 | **0.1383** | 0.1279 | 0.0118 | 0.0628 | **0.2062** | 0.1944 | 7.7e-6 | **0.0791** |
| NPO | 0.2758 | 0.0571 | 0.1650 | **0.4357** | 0.0464 | 0.0369 | 0.0680 | **0.1823** | **0.0395** | 0.0355 | 0.0658 | **0.0754** |
| SimNPO | 0.2548 | 0.0593 | 0.2657 | 0.4303 | 0.0418 | 0.0315 | 0.0680 | **0.1274** | 0.0244 | 0.0241 | 0.0812 | 0.0460 |
| WGA | 0.1784 | **0.0369** | 0.0067 | 0.3894 | 0.0000 | **0.0000** | 1.5e-23 | 0.0000 | 0.0000 | **0.0000** | 1.7e-26 | 0.0000 |
| SatImp | **0.3110** | 0.0767 | 0.4046 | **0.4575** | 0.0000 | **0.0000** | 2.6e-14 | 0.0000 | 0.0000 | **0.0000** | 5.4e-18 | 0.0000 |
| LLaMA2-7B | | | | | | | | | | | | |
| Original | 0.8529 | 0.8550 | 5.1e-7 | 0.6345 | 0.8529 | 0.8476 | 1.1e-14 | 0.6345 | 0.8529 | 0.8610 | 4.5e-18 | 0.6345 |
| GA | 0.3957 | **0.0583** | 0.5786 | 0.5249 | 0.0000 | **0.0000** | 1.4e-12 | 5.9e-4 | 0.0000 | **0.0000** | 1.6e-11 | 0.0000 |
| DPO | **0.6188** | 0.3959 | 0.0971 | 0.5712 | **0.3749** | 0.3066 | 2.4e-8 | 0.0304 | **0.4477** | 0.4031 | 2.1e-13 | **0.2703** |
| NPO | 0.5382 | 0.0947 | 0.4049 | **0.5818** | **0.1067** | 0.0555 | 0.0680 | **0.3925** | 0.0790 | 0.0551 | 1.4e-6 | **0.2814** |
| SimNPO | 0.4758 | 0.0943 | 0.5631 | 0.5786 | 0.0852 | 0.0513 | 0.0021 | **0.3086** | **0.0810** | 0.0650 | 0.0002 | 0.1976 |
| WGA | 0.4801 | **0.0186** | 0.7659 | 0.5716 | 0.0000 | **0.0000** | 1.5e-26 | 0.0000 | 0.0000 | **0.0000** | 6.4e-40 | 0.0000 |
| SatImp | **0.6529** | **0.0583** | 0.5912 | **0.7660** | 0.0080 | 0.0057 | 4.9e-26 | 0.0000 | 0.0000 | **0.0000** | 1.9e-29 | 0.0000 |

Performances of multiple methods on MUSE benchmark are shown in Table 5. As mentioned before, PrivLeak is a metric for evaluating the degree of unlearning, thus, we prioritize checkpoint selection based on PrivLeak performance. Results reveals that SatImp exhibits excellent privacy protection and unlearning performance, except in the news scenario with retain regularization. This performance also highlights SatImp's exceptional ability to control the unlearning progress.

*Table 5.* Comparison between unlearning objectives on MUSE benchmark. ↑ / ↓ indicate larger / smaller values are preferable. → 0 indicate near 0 is preferable. Top results in PrivLeak and UtilPres are in **bold** font.

| Method | Books | | | | News | | | |
|---|---|---|---|---|---|---|---|---|
| | VerbMem ↓ | KnowMem ↓ | PrivLeak → 0 | UtilPres ↑ | VerbMem ↓ | KnowMem ↓ | PrivLeak → 0 | UtilPres ↑ |
| *Unlearning with forget-only regularization* | | | | | | | | |
| GA | 0.0 | 0.0 | 34.7 | 0.0 | 0.0 | 0.0 | 27.7 | 0.0 |
| NPO | 34.3 | 31.9 | -52.3 | **62.0** | 20.4 | 46.8 | -94.3 | **40.0** |
| SimNPO | 3.0 | 0.1 | -9.8 | 0.3 | 0.0 | 0.0 | 17.1 | 0.0 |
| WGA | 0.0 | 0.0 | 13.0 | 0.0 | 0.0 | 0.0 | 14.5 | 0.0 |
| SatImp | 0.0 | 0.0 | **7.2** | 0.0 | 0.0 | 0.0 | **13.6** | 0.0 |
| *Unlearning with retain regularization* | | | | | | | | |
| GA | 0.0 | 0.0 | 34.7 | 8.9 | 0.0 | 0.0 | **2.1** | 0.0 |
| SimNPO | 15.5 | 35.9 | -35.1 | **45.1** | 35.0 | 45.9 | -99.4 | 35.7 |
| WGA | 0.0 | 0.0 | -25.3 | 36.7 | 1.9 | 0.0 | 97.5 | 35.5 |
| SatImp | 0.0 | 0.0 | **-21.1** | 37.9 | 7.7 | 0.0 | 50.5 | **36.1** |

We further employ the retain regularization on WMDP benchmark. Notably, we do not specifically adjust hyperparameters for WMDP; all methods followed the settings used on TOFU. The results indicate that existing methods exhibit varying degrees of inefficient unlearning on WMDP. As a method thoroughly explored on WMDP, RMU achieves a better unlearn-retain trade-off. Although SatImp outperforms only WGA and SimNPO in unlearning, it demonstrates the best utility performance. This performance also highlights that, like other methods, SatImp is sensitive to hyper-parameters, warranting further investigation in future research.

*Table 6.* Comparison between unlearning objectives on WMDP with retain regularization. ↑ / ↓ indicate larger / smaller values are preferable. Top two results are in **bold** font.

| Method | Unlearn | | Retain |
|---|---|---|---|
| | WMDP-Bio ↓ | WMDP-Cyber ↓ | MMLU ↑ |
| Original | 0.6462 | 0.3924 | 0.5853 |
| GA | 0.2739 | **0.2657** | 0.4265 |
| NPO | 0.2647 | 0.3067 | 0.4434 |
| SimNPO | 0.2617 | 0.3163 | 0.4453 |
| WGA | **0.2590** | 0.2989 | 0.4806 |
| RMU | 0.3493 | 0.3578 | **0.5523** |
| SatImp | **0.2598** | **0.2815** | **0.5391** |

To facilitate easy reference to the investigate results in Section 4, we provide a simple index here.

Section 4.1, importance-based reweighting: Figure 11, 12, Table 7, 8.

Section 4.2, comparison between SimImp and SimSat: Figure 13, 14. Table 9, 10.

Section 4.2, granularity of reweighting: Figure 15, 16, Table 11, 12, 13, 14.

Section 4.2, hard sampling methods: Figure 17, 18. Table 15, 16.

Considering the total length of our paper, we present FQ and MU detailed results in our GitHub repository.

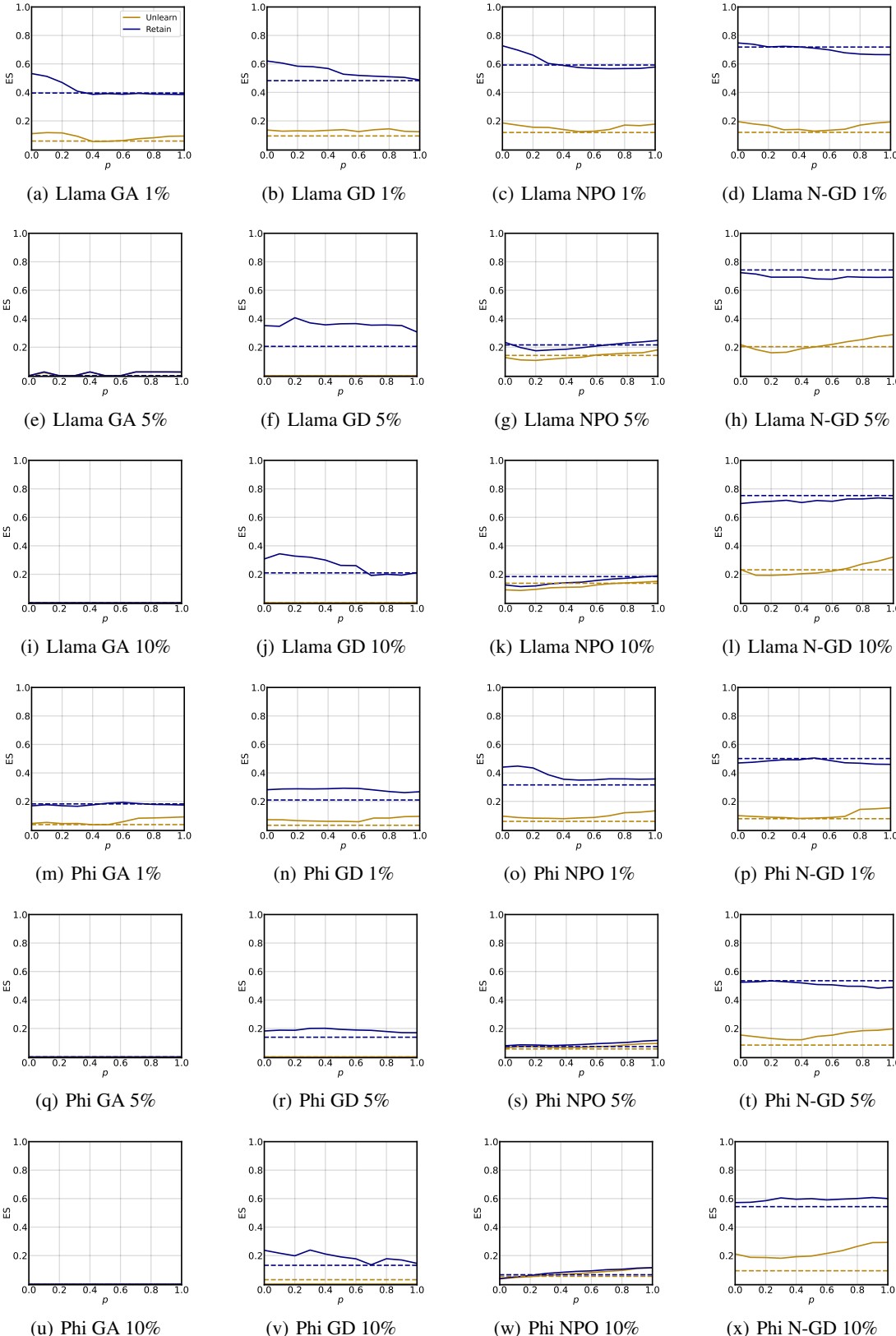

*Figure 11.* ES Scores with importance-based reweighting. We depict values of $p$ (**x-axis**) versus the ES scores (**y-axis**) on unlearn and retain data. We consider 2 LLMs (Phi-1.5 (Phi) and LLaMA-2-7B (Llama)) and 4 unlearning methods (GA, GD, NPO, and NPO-GD (N-GD)) under the 1%, 5%, and 10% TOFU setup. Dashed line represents baseline performance.

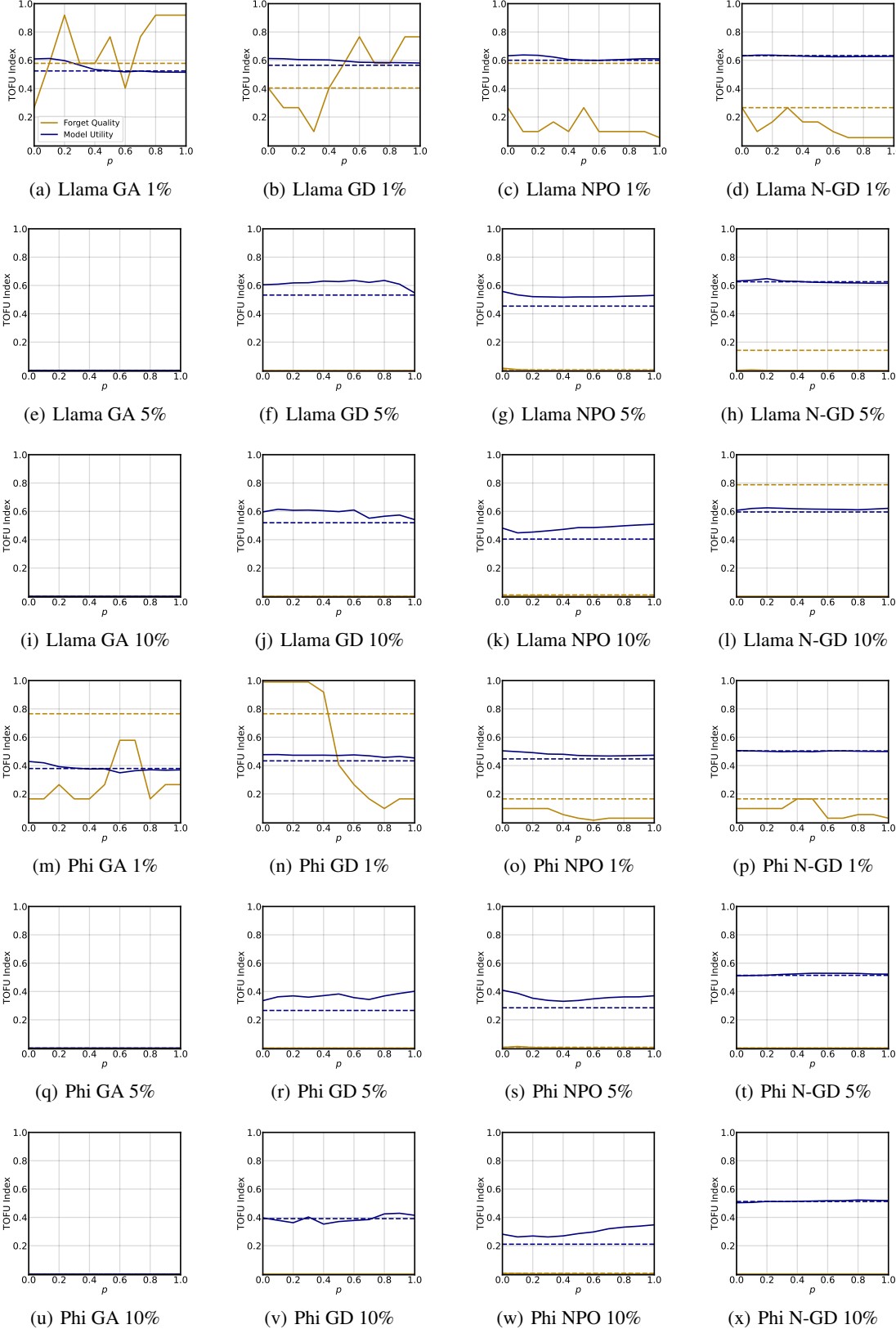

*Figure 12.* Forget Quality and Model Utility with importance-based reweighting. We depict values of $p$ (**x-axis**) versus the TOFU Index (**y-axis**) on unlearn and retain data. We consider 2 LLMs (Phi-1.5 (Phi) and LLaMA-2-7B (Llama)) and 4 unlearning methods (GA, GD, NPO, and NPO-GD (N-GD)) under the 1%, 5%, and 10% TOFU setup. Dashed line represents baseline performance.

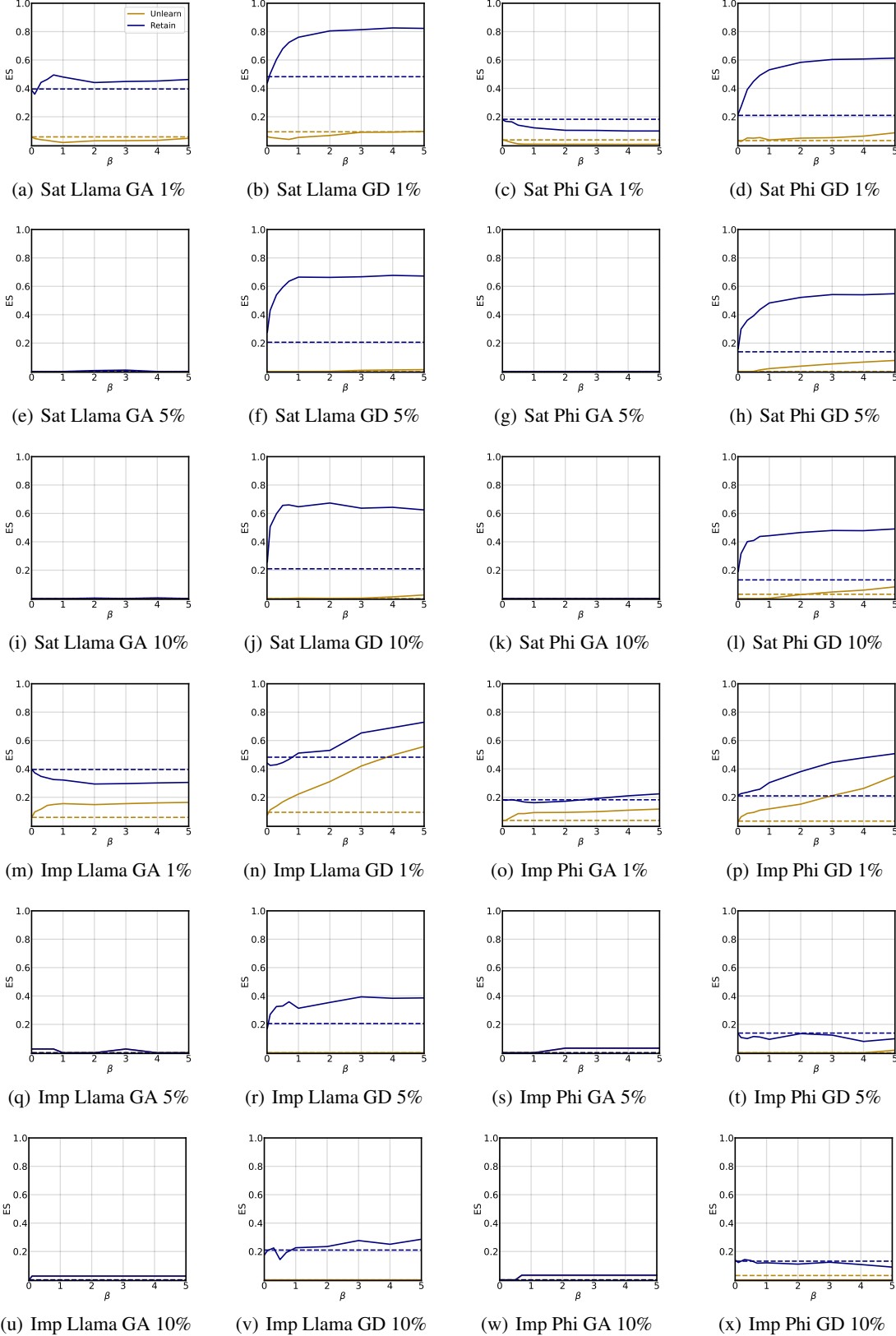

*Figure 13.* ES Scores with SimSat (Sat) and SimImp (Imp) paradigms. We depict values of $\beta$ (**x-axis**) versus the ES scores (**y-axis**) on unlearn and retain data. We consider 2 LLMs (Phi-1.5 (Phi) and LLaMA-2-7B (Llama)) and 2 unlearning methods (GA, GD) under the 1%, 5%, and 10% TOFU setup. Dashed line represents baseline performance.

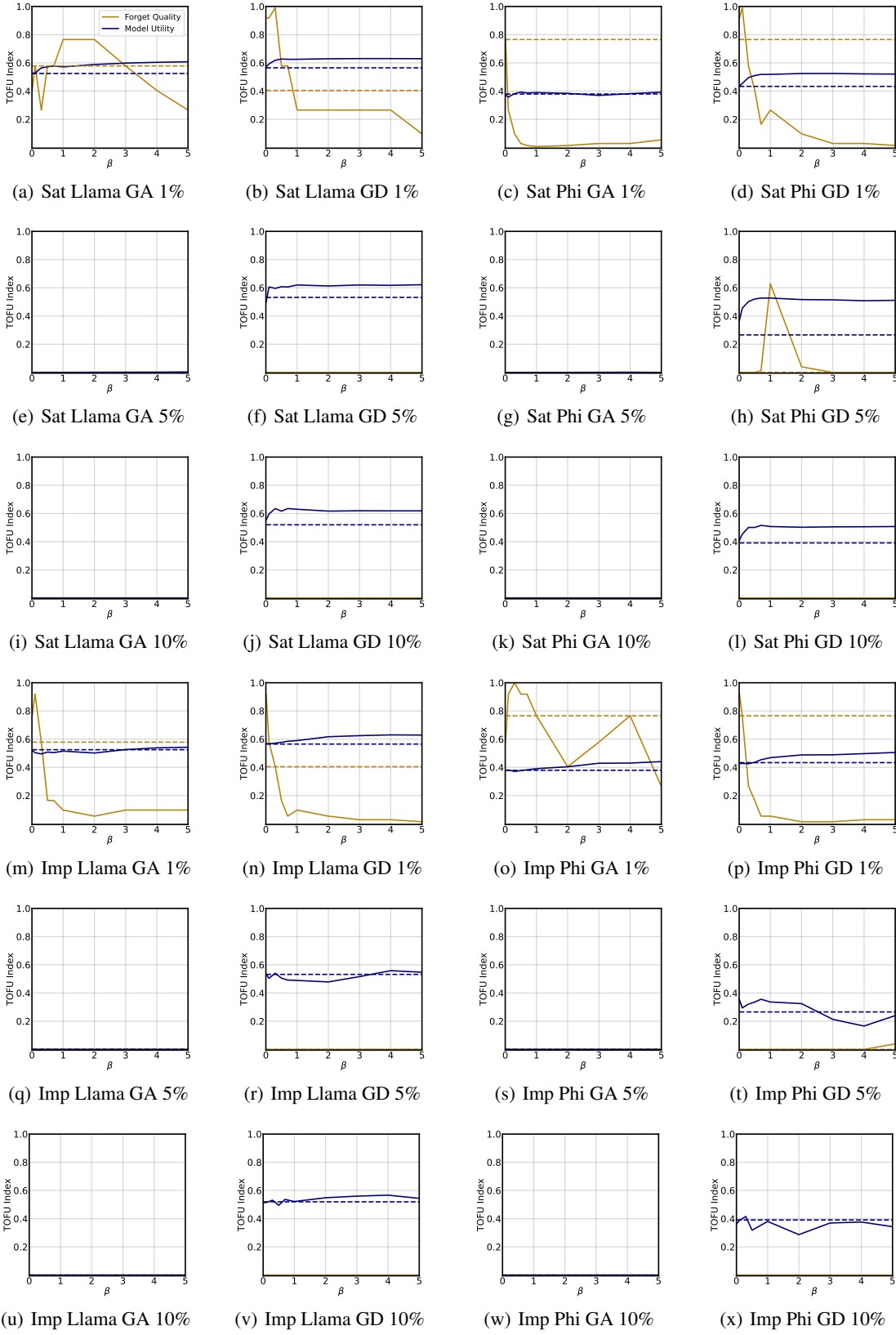

*Figure 14.* Forget Quality and Model Utility with SimSat (Sat) and SimImp (Imp) paradigms. We depict values of $\beta$ (**x-axis**) versus the TOFU-Index (**y-axis**) on unlearn and retain data. We consider 2 LLMs (Phi-1.5 (Phi) and LLaMA-2-7B (Llama)) and 2 unlearning methods (GA, GD) under the 1%, 5%, and 10% TOFU unlearning setup. Dashed line represents baseline performance.

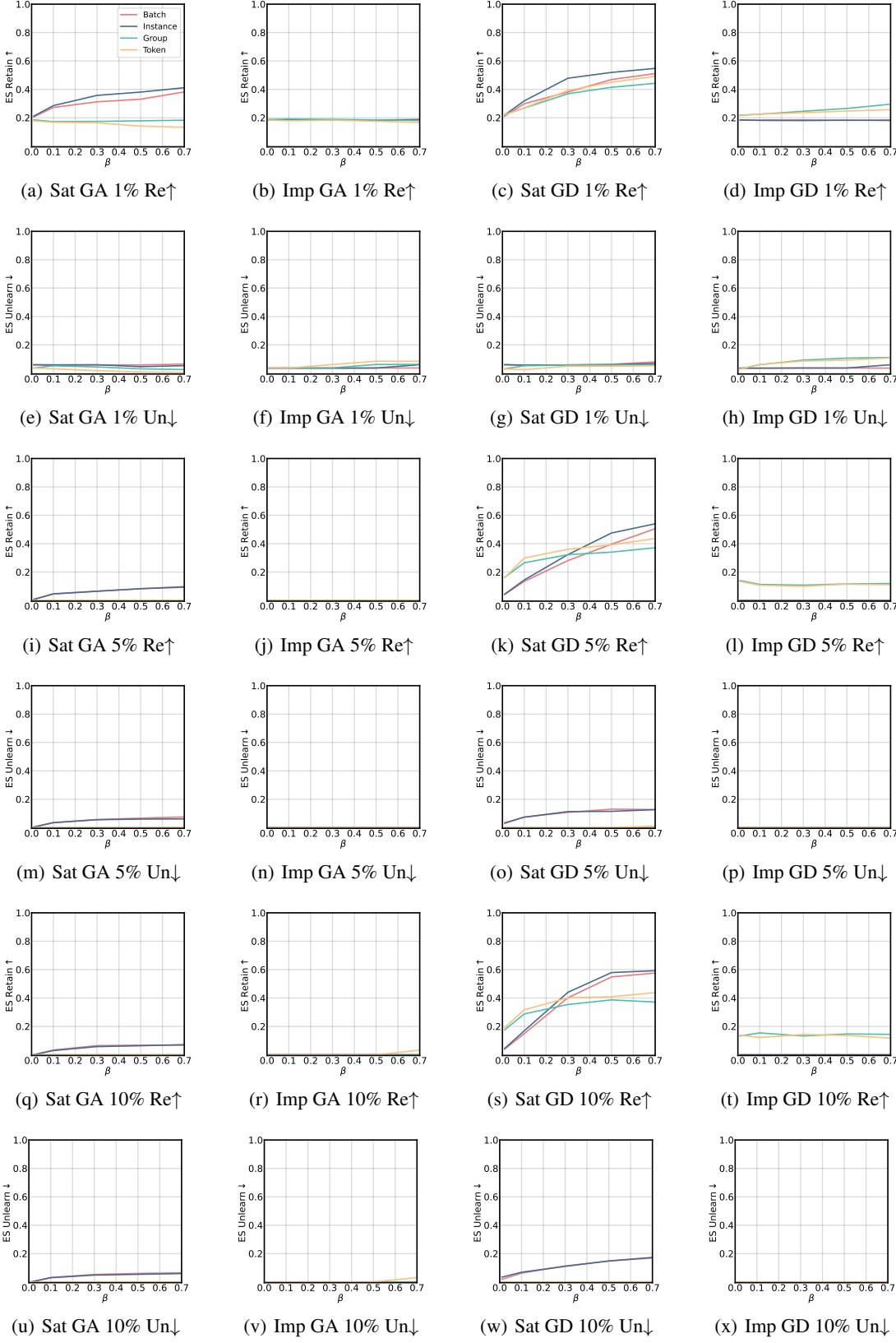

*Figure 15.* Impact of weight granularity on Phi-1.5 Model. We depict values of $\beta$ (**x-axis**) versus the ES scores (**y-axis**) on unlearn (Un↓) and retain (Re↑) data. SimSat (Sat) and SimImp (Imp) methods are investigated.

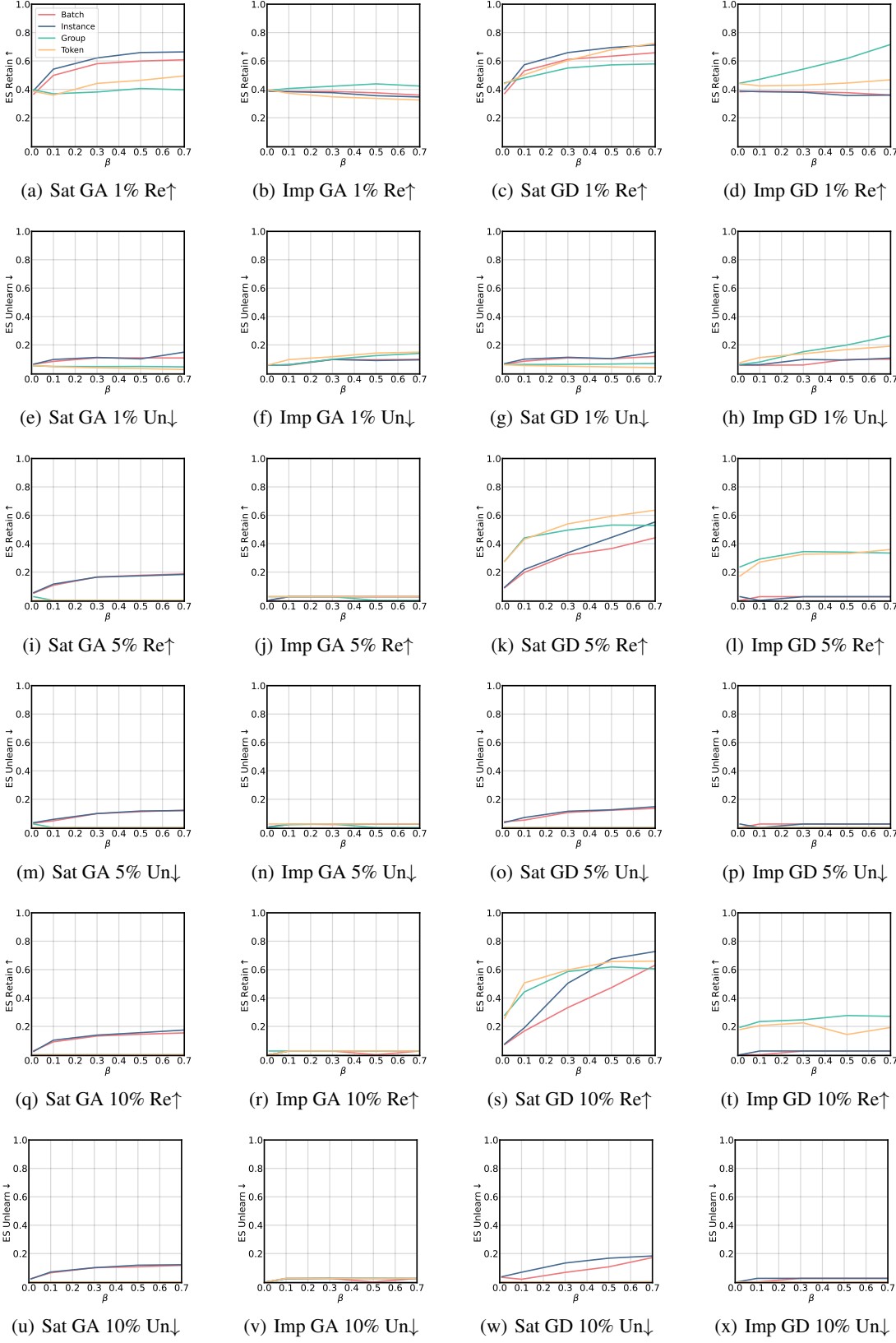

*Figure 16.* Impact of weight granularity on LLaMA-2-7B Model. We depict values of $\beta$ (**x-axis**) versus the ES scores (**y-axis**) on unlearn (Un↓) and retain (Re↑) data. SimSat (Sat) and SimImp (Imp) methods are investigated.

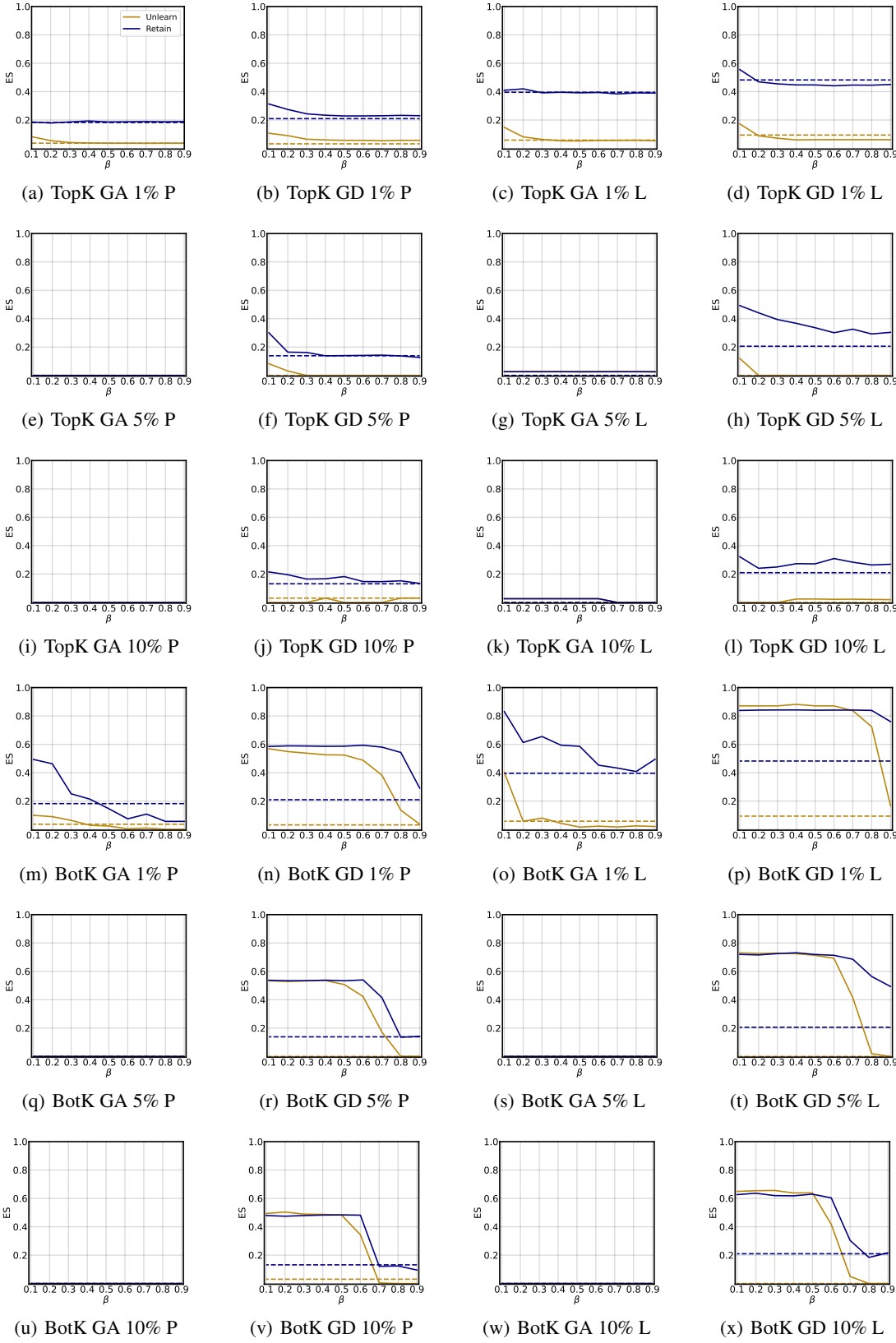

*Figure 17.* Hard sampling performance of TopK and BottomK (BotK) sampling. We depict values of $\beta$ (**x-axis**) versus the ES scores (**y-axis**) on unlearn (Un↓) and retain (Re↑) data. LLaMA-2-7B (L) and Phi-1.5 (P) are investigated. Dashed line represents baseline performance.

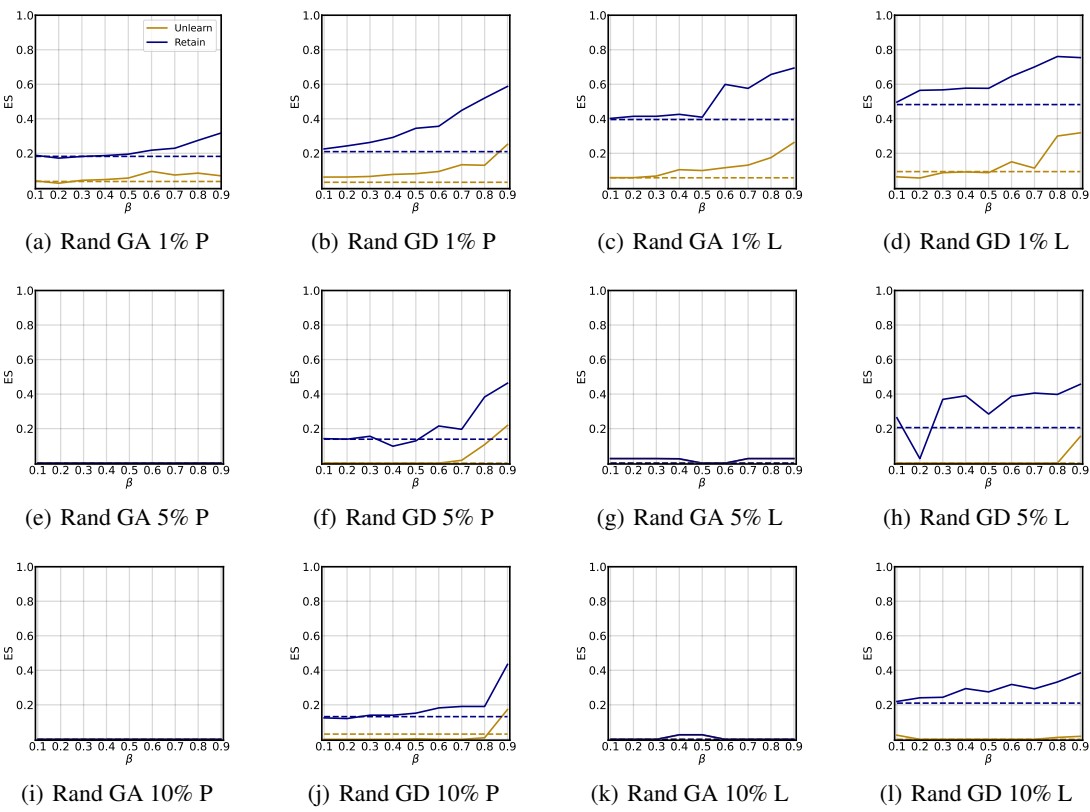

*Figure 18.* Hard sampling performance of random (Rand) sampling. We depict values of $\beta$ (**x-axis**) versus the ES scores (**y-axis**) on unlearn (Un↓) and retain (Re↑) data. LLaMA-2-7B (L) and Phi-1.5 (P) are investigated. Dashed line represents baseline performance.

*Table 7.* Importance-based reweighting (I) results on TOFU with LLaMA-2-7B (ES Score). We consider 4 unlearning methods (GA, GD, NPO, NPO-GD).

| Importance | | Forget-1% | | | | Forget-5% | | | | Forget-10% | | | |
| --- | --- | --- | --- | --- | --- | --- | --- | --- | --- | --- | --- | --- | --- |
| | | ES-exact | | ES-perturb | | ES-exact | | ES-perturb | | ES-exact | | ES-perturb | |
| Method | $1-p$ | retain ↑ | unlearn ↓ | retain ↑ | unlearn ↓ | retain ↑ | unlearn ↓ | retain ↑ | unlearn ↓ | retain ↑ | unlearn ↓ | retain ↑ | unlearn ↓ |
| | 0.00 | 0.3852 | 0.0935 | 0.2723 | 0.0841 | 0.0267 | 0.0234 | 0.0267 | 0.0234 | 0.0000 | 0.0000 | 0.0000 | 0.0000 |
| | 0.10 | 0.3867 | 0.0919 | 0.2694 | 0.0759 | 0.0267 | 0.0255 | 0.0267 | 0.0255 | 0.0000 | 0.0000 | 0.0000 | 0.0000 |
| | 0.20 | 0.3879 | 0.0815 | 0.2659 | 0.0719 | 0.0267 | 0.0255 | 0.0267 | 0.0255 | 0.0000 | 0.0000 | 0.0000 | 0.0000 |
| | 0.30 | 0.3933 | 0.0746 | 0.2655 | 0.0610 | 0.0267 | 0.0255 | 0.0267 | 0.0255 | 0.0000 | 0.0000 | 0.0000 | 0.0000 |
| | 0.40 | 0.3869 | 0.0619 | 0.2624 | 0.0561 | 0.0000 | 0.0000 | 0.0000 | 0.0000 | 0.0000 | 0.0000 | 0.0000 | 0.0000 |
| I-GA | 0.50 | 0.3912 | 0.0565 | 0.2625 | 0.0527 | 0.0000 | 0.0000 | 0.0000 | 0.0000 | 0.0000 | 0.0000 | 0.0000 | 0.0000 |
| | 0.60 | 0.3863 | 0.0550 | 0.2652 | 0.0517 | 0.0267 | 0.0255 | 0.0267 | 0.0255 | 0.0000 | 0.0000 | 0.0000 | 0.0000 |
| | 0.70 | 0.4082 | 0.0918 | 0.2756 | 0.0616 | 0.0000 | 0.0000 | 0.0000 | 0.0000 | 0.0000 | 0.0000 | 0.0000 | 0.0000 |
| | 0.80 | 0.4693 | 0.1154 | 0.2931 | 0.0613 | 0.0000 | 0.0000 | 0.0000 | 0.0000 | 0.0000 | 0.0000 | 0.0000 | 0.0000 |
| | 0.90 | 0.5125 | 0.1175 | 0.3093 | 0.0925 | 0.0267 | 0.0211 | 0.0267 | 0.0211 | 0.0000 | 0.0000 | 0.0000 | 0.0000 |
| | 1.00 | 0.5322 | 0.1103 | 0.3023 | 0.0784 | 0.0000 | 0.0000 | 0.0000 | 0.0000 | 0.0000 | 0.0000 | 0.0000 | 0.0000 |
| | 0.00 | 0.4872 | 0.1237 | 0.3328 | 0.1149 | 0.3068 | 0.0000 | 0.2434 | 0.0000 | 0.2118 | 0.0000 | 0.1807 | 0.0000 |
| | 0.10 | 0.5052 | 0.1269 | 0.3367 | 0.1167 | 0.3521 | 0.0000 | 0.2736 | 0.0000 | 0.1940 | 0.0000 | 0.1669 | 0.0000 |
| | 0.20 | 0.5097 | 0.1441 | 0.3355 | 0.0986 | 0.3561 | 0.0000 | 0.2732 | 0.0000 | 0.2003 | 0.0000 | 0.1734 | 0.0000 |
| | 0.30 | 0.5141 | 0.1376 | 0.3308 | 0.1124 | 0.3547 | 0.0000 | 0.2686 | 0.0000 | 0.1916 | 0.0000 | 0.1603 | 0.0000 |
| | 0.40 | 0.5192 | 0.1248 | 0.3345 | 0.1062 | 0.3656 | 0.0000 | 0.2660 | 0.0000 | 0.2609 | 0.0000 | 0.2119 | 0.0000 |
| I-GD | 0.50 | 0.5278 | 0.1383 | 0.3357 | 0.1129 | 0.3643 | 0.0000 | 0.2610 | 0.0000 | 0.2625 | 0.0000 | 0.2020 | 0.0000 |
| | 0.60 | 0.5678 | 0.1335 | 0.3470 | 0.1028 | 0.3570 | 0.0000 | 0.2630 | 0.0000 | 0.3000 | 0.0000 | 0.2333 | 0.0000 |
| | 0.70 | 0.5803 | 0.1282 | 0.3518 | 0.1063 | 0.3705 | 0.0000 | 0.2576 | 0.0000 | 0.3198 | 0.0000 | 0.2422 | 0.0000 |
| | 0.80 | 0.5840 | 0.1298 | 0.3455 | 0.0897 | 0.4069 | 0.0000 | 0.2703 | 0.0000 | 0.3278 | 0.0000 | 0.2444 | 0.0000 |
| | 0.90 | 0.6056 | 0.1277 | 0.3552 | 0.1089 | 0.3468 | 0.0000 | 0.2404 | 0.0000 | 0.3438 | 0.0000 | 0.2534 | 0.0000 |
| | 1.00 | 0.6205 | 0.1358 | 0.3563 | 0.0904 | 0.3516 | 0.0000 | 0.2509 | 0.0000 | 0.3077 | 0.0000 | 0.2407 | 0.0000 |
| | 0.00 | 0.5780 | 0.1773 | 0.4057 | 0.1578 | 0.2472 | 0.1811 | 0.2146 | 0.1547 | 0.1884 | 0.1508 | 0.1674 | 0.1363 |
| | 0.10 | 0.5684 | 0.1665 | 0.4119 | 0.1489 | 0.2374 | 0.1614 | 0.2075 | 0.1433 | 0.1821 | 0.1443 | 0.1599 | 0.1281 |
| | 0.20 | 0.5675 | 0.1706 | 0.4082 | 0.1529 | 0.2301 | 0.1582 | 0.1987 | 0.1399 | 0.1730 | 0.1400 | 0.1518 | 0.1219 |
| | 0.30 | 0.5664 | 0.1395 | 0.4068 | 0.1250 | 0.2198 | 0.1522 | 0.1941 | 0.1328 | 0.1666 | 0.1350 | 0.1503 | 0.1170 |
| | 0.40 | 0.5691 | 0.1274 | 0.4055 | 0.1263 | 0.2081 | 0.1458 | 0.1802 | 0.1258 | 0.1568 | 0.1254 | 0.1437 | 0.1103 |
| I-NPO | 0.50 | 0.5744 | 0.1246 | 0.4018 | 0.1147 | 0.1964 | 0.1293 | 0.1734 | 0.1202 | 0.1445 | 0.1107 | 0.1309 | 0.1020 |
| | 0.60 | 0.5892 | 0.1392 | 0.4001 | 0.1091 | 0.1857 | 0.1239 | 0.1639 | 0.1100 | 0.1404 | 0.1088 | 0.1265 | 0.1006 |
| | 0.70 | 0.6034 | 0.1539 | 0.3959 | 0.1239 | 0.1812 | 0.1162 | 0.1630 | 0.1029 | 0.1337 | 0.1056 | 0.1222 | 0.0950 |
| | 0.80 | 0.6615 | 0.1553 | 0.4043 | 0.1222 | 0.1750 | 0.1074 | 0.1634 | 0.0948 | 0.1185 | 0.0943 | 0.1145 | 0.0877 |
| | 0.90 | 0.6971 | 0.1698 | 0.4079 | 0.1270 | 0.1976 | 0.1107 | 0.1744 | 0.0990 | 0.1136 | 0.0865 | 0.1151 | 0.0825 |
| | 1.00 | 0.7274 | 0.1855 | 0.4107 | 0.1488 | 0.2336 | 0.1272 | 0.1915 | 0.1129 | 0.1250 | 0.0908 | 0.1133 | 0.0861 |
| | 0.00 | 0.6650 | 0.1926 | 0.4532 | 0.1703 | 0.6911 | 0.2885 | 0.4238 | 0.2147 | 0.7314 | 0.3214 | 0.4531 | 0.2240 |
| | 0.10 | 0.6656 | 0.1844 | 0.4531 | 0.1596 | 0.6903 | 0.2751 | 0.4266 | 0.2072 | 0.7361 | 0.2922 | 0.4501 | 0.2080 |
| | 0.20 | 0.6692 | 0.1703 | 0.4527 | 0.1544 | 0.6912 | 0.2540 | 0.4241 | 0.1828 | 0.7295 | 0.2732 | 0.4438 | 0.1919 |
| | 0.30 | 0.6782 | 0.1413 | 0.4457 | 0.1310 | 0.6949 | 0.2391 | 0.4211 | 0.1696 | 0.7289 | 0.2417 | 0.4384 | 0.1802 |
| | 0.40 | 0.6983 | 0.1343 | 0.4438 | 0.1344 | 0.6775 | 0.2199 | 0.4184 | 0.1664 | 0.7125 | 0.2227 | 0.4338 | 0.1709 |
| I-NPO-GD | 0.50 | 0.7107 | 0.1276 | 0.4390 | 0.1238 | 0.6794 | 0.2041 | 0.4063 | 0.1489 | 0.7180 | 0.2091 | 0.4281 | 0.1597 |
| | 0.60 | 0.7200 | 0.1401 | 0.4417 | 0.1048 | 0.6922 | 0.1897 | 0.4060 | 0.1338 | 0.7039 | 0.2039 | 0.4266 | 0.1620 |
| | 0.70 | 0.7236 | 0.1376 | 0.4372 | 0.1254 | 0.6920 | 0.1645 | 0.4131 | 0.1206 | 0.7196 | 0.1964 | 0.4241 | 0.1470 |
| | 0.80 | 0.7194 | 0.1675 | 0.4294 | 0.1342 | 0.6919 | 0.1610 | 0.4212 | 0.1272 | 0.7125 | 0.1925 | 0.4134 | 0.1450 |
| | 0.90 | 0.7383 | 0.1788 | 0.4310 | 0.1352 | 0.7137 | 0.1837 | 0.4324 | 0.1382 | 0.7064 | 0.1930 | 0.4177 | 0.1491 |
| | 1.00 | 0.7472 | 0.1949 | 0.4290 | 0.1569 | 0.7233 | 0.2178 | 0.4348 | 0.1561 | 0.6970 | 0.2327 | 0.4129 | 0.1663 |

*Table 8.* Importance-based reweighting (I) results on TOFU with Phi-1.5 (ES Score). We consider 4 unlearning methods (GA, GD, NPO, NPO-GD).

| Importance | | Forget-1% | | | | Forget-5% | | | | Forget-10% | | | |
|---|---|---|---|---|---|---|---|---|---|---|---|---|---|
| | | ES-exact | | ES-perturb | | ES-exact | | ES-perturb | | ES-exact | | ES-perturb | |
| Method | $1-p$ | retain ↑ | unlearn ↓ | retain ↑ | unlearn ↓ | retain ↑ | unlearn ↓ | retain ↑ | unlearn ↓ | retain ↑ | unlearn ↓ | retain ↑ | unlearn ↓ |
| | 0.00 | 0.1743 | 0.0912 | 0.1255 | 0.0644 | 0.0000 | 0.0000 | 0.0000 | 0.0000 | 0.0000 | 0.0000 | 0.0000 | 0.0000 |
| | 0.10 | 0.1772 | 0.0869 | 0.1269 | 0.0639 | 0.0000 | 0.0000 | 0.0000 | 0.0000 | 0.0000 | 0.0000 | 0.0000 | 0.0000 |
| | 0.20 | 0.1786 | 0.0839 | 0.1297 | 0.0664 | 0.0000 | 0.0000 | 0.0000 | 0.0000 | 0.0000 | 0.0000 | 0.0000 | 0.0000 |
| | 0.30 | 0.1849 | 0.0819 | 0.1253 | 0.0716 | 0.0000 | 0.0000 | 0.0000 | 0.0000 | 0.0000 | 0.0000 | 0.0000 | 0.0000 |
| | 0.40 | 0.1927 | 0.0576 | 0.1335 | 0.0704 | 0.0000 | 0.0000 | 0.0000 | 0.0000 | 0.0000 | 0.0000 | 0.0000 | 0.0000 |
| I-GA | 0.50 | 0.1877 | 0.0364 | 0.1255 | 0.0567 | 0.0000 | 0.0000 | 0.0000 | 0.0000 | 0.0000 | 0.0000 | 0.0000 | 0.0000 |
| | 0.60 | 0.1757 | 0.0362 | 0.1242 | 0.0508 | 0.0000 | 0.0000 | 0.0000 | 0.0000 | 0.0000 | 0.0000 | 0.0000 | 0.0000 |
| | 0.70 | 0.1655 | 0.0451 | 0.1216 | 0.0294 | 0.0000 | 0.0000 | 0.0000 | 0.0000 | 0.0000 | 0.0000 | 0.0000 | 0.0000 |
| | 0.80 | 0.1695 | 0.0440 | 0.1180 | 0.0263 | 0.0000 | 0.0000 | 0.0000 | 0.0000 | 0.0000 | 0.0000 | 0.0000 | 0.0000 |
| | 0.90 | 0.1765 | 0.0525 | 0.1186 | 0.0335 | 0.0000 | 0.0000 | 0.0000 | 0.0000 | 0.0000 | 0.0000 | 0.0000 | 0.0000 |
| | 1.00 | 0.1688 | 0.0450 | 0.1175 | 0.0365 | 0.0000 | 0.0000 | 0.0000 | 0.0000 | 0.0000 | 0.0000 | 0.0000 | 0.0000 |
| | 0.00 | 0.2677 | 0.0948 | 0.1601 | 0.0465 | 0.1701 | 0.0000 | 0.1261 | 0.0000 | 0.1452 | 0.0000 | 0.1192 | 0.0000 |
| | 0.10 | 0.2615 | 0.0928 | 0.1656 | 0.0464 | 0.1708 | 0.0000 | 0.1265 | 0.0000 | 0.1688 | 0.0000 | 0.1376 | 0.0000 |
| | 0.20 | 0.2691 | 0.0829 | 0.1682 | 0.0408 | 0.1787 | 0.0000 | 0.1338 | 0.0000 | 0.1778 | 0.0000 | 0.1355 | 0.0001 |
| | 0.30 | 0.2802 | 0.0830 | 0.1680 | 0.0394 | 0.1867 | 0.0000 | 0.1297 | 0.0000 | 0.1354 | 0.0000 | 0.1118 | 0.0000 |
| | 0.40 | 0.2909 | 0.0571 | 0.1736 | 0.0394 | 0.1891 | 0.0000 | 0.1358 | 0.0000 | 0.1773 | 0.0000 | 0.1261 | 0.0000 |
| I-GD | 0.50 | 0.2926 | 0.0599 | 0.1695 | 0.0385 | 0.1938 | 0.0000 | 0.1398 | 0.0000 | 0.1907 | 0.0000 | 0.1295 | 0.0000 |
| | 0.60 | 0.2889 | 0.0598 | 0.1729 | 0.0376 | 0.2015 | 0.0000 | 0.1422 | 0.0000 | 0.2109 | 0.0000 | 0.1414 | 0.0000 |
| | 0.70 | 0.2873 | 0.0620 | 0.1772 | 0.0393 | 0.2008 | 0.0000 | 0.1441 | 0.0000 | 0.2387 | 0.0000 | 0.1439 | 0.0000 |
| | 0.80 | 0.2883 | 0.0646 | 0.1715 | 0.0411 | 0.1873 | 0.0002 | 0.1358 | 0.0002 | 0.1984 | 0.0000 | 0.1313 | 0.0000 |
| | 0.90 | 0.2872 | 0.0708 | 0.1696 | 0.0430 | 0.1880 | 0.0002 | 0.1377 | 0.0002 | 0.2168 | 0.0000 | 0.1347 | 0.0000 |
| | 1.00 | 0.2818 | 0.0713 | 0.1630 | 0.0481 | 0.1824 | 0.0002 | 0.1307 | 0.0000 | 0.2370 | 0.0000 | 0.1441 | 0.0000 |
| | 0.00 | 0.3574 | 0.1337 | 0.2196 | 0.1233 | 0.1164 | 0.0947 | 0.1060 | 0.0804 | 0.1151 | 0.1138 | 0.1099 | 0.0962 |
| | 0.10 | 0.3556 | 0.1242 | 0.2115 | 0.1105 | 0.1113 | 0.0928 | 0.0998 | 0.0749 | 0.1119 | 0.1086 | 0.1097 | 0.0931 |
| | 0.20 | 0.3579 | 0.1203 | 0.2139 | 0.1176 | 0.1025 | 0.0860 | 0.0976 | 0.0724 | 0.1040 | 0.0956 | 0.1013 | 0.0843 |
| | 0.30 | 0.3582 | 0.0993 | 0.2115 | 0.1114 | 0.0975 | 0.0760 | 0.0918 | 0.0662 | 0.1008 | 0.0888 | 0.0995 | 0.0805 |
| | 0.40 | 0.3511 | 0.0872 | 0.2105 | 0.0987 | 0.0934 | 0.0728 | 0.0863 | 0.0639 | 0.0926 | 0.0805 | 0.0913 | 0.0775 |
| I-NPO | 0.50 | 0.3496 | 0.0837 | 0.2115 | 0.0877 | 0.0874 | 0.0678 | 0.0787 | 0.0618 | 0.0889 | 0.0729 | 0.0850 | 0.0736 |
| | 0.60 | 0.3557 | 0.0785 | 0.2080 | 0.0602 | 0.0834 | 0.0676 | 0.0726 | 0.0588 | 0.0817 | 0.0678 | 0.0797 | 0.0666 |
| | 0.70 | 0.3871 | 0.0812 | 0.2094 | 0.0590 | 0.0804 | 0.0670 | 0.0728 | 0.0611 | 0.0743 | 0.0608 | 0.0693 | 0.0559 |
| | 0.80 | 0.4351 | 0.0822 | 0.2104 | 0.0557 | 0.0840 | 0.0728 | 0.0733 | 0.0605 | 0.0595 | 0.0520 | 0.0558 | 0.0490 |
| | 0.90 | 0.4479 | 0.0868 | 0.2195 | 0.0631 | 0.0853 | 0.0747 | 0.0736 | 0.0672 | 0.0498 | 0.0459 | 0.0489 | 0.0416 |
| | 1.00 | 0.4406 | 0.0966 | 0.2149 | 0.0721 | 0.0788 | 0.0632 | 0.0647 | 0.0544 | 0.0389 | 0.0356 | 0.0390 | 0.0337 |
| | 0.00 | 0.4601 | 0.1546 | 0.2523 | 0.1266 | 0.4899 | 0.1973 | 0.2434 | 0.1419 | 0.6003 | 0.2922 | 0.2788 | 0.1653 |
| | 0.10 | 0.4612 | 0.1479 | 0.2531 | 0.1329 | 0.4830 | 0.1874 | 0.2445 | 0.1359 | 0.6077 | 0.2908 | 0.2776 | 0.1555 |
| | 0.20 | 0.4680 | 0.1439 | 0.2525 | 0.1191 | 0.4962 | 0.1846 | 0.2471 | 0.1323 | 0.6007 | 0.2645 | 0.2753 | 0.1469 |
| | 0.30 | 0.4713 | 0.0942 | 0.2538 | 0.0774 | 0.4968 | 0.1730 | 0.2470 | 0.1153 | 0.5962 | 0.2342 | 0.2709 | 0.1362 |
| | 0.40 | 0.4878 | 0.0867 | 0.2548 | 0.0644 | 0.5066 | 0.1531 | 0.2562 | 0.1098 | 0.5909 | 0.2152 | 0.2732 | 0.1274 |
| I-NPO-GD | 0.50 | 0.5048 | 0.0826 | 0.2571 | 0.0550 | 0.5085 | 0.1441 | 0.2558 | 0.1010 | 0.5997 | 0.1966 | 0.2623 | 0.1246 |
| | 0.60 | 0.4923 | 0.0801 | 0.2538 | 0.0544 | 0.5200 | 0.1210 | 0.2539 | 0.0939 | 0.5956 | 0.1920 | 0.2559 | 0.1255 |
| | 0.70 | 0.4922 | 0.0865 | 0.2437 | 0.0621 | 0.5278 | 0.1217 | 0.2551 | 0.0968 | 0.6051 | 0.1813 | 0.2594 | 0.1195 |
| | 0.80 | 0.4845 | 0.0893 | 0.2417 | 0.0653 | 0.5342 | 0.1296 | 0.2605 | 0.0946 | 0.5854 | 0.1862 | 0.2719 | 0.1264 |
| | 0.90 | 0.4750 | 0.0944 | 0.2367 | 0.0711 | 0.5276 | 0.1420 | 0.2662 | 0.0995 | 0.5741 | 0.1876 | 0.2685 | 0.1295 |
| | 1.00 | 0.4696 | 0.0998 | 0.2322 | 0.0727 | 0.5247 | 0.1539 | 0.2612 | 0.1023 | 0.5720 | 0.2111 | 0.2648 | 0.1361 |

*Table 9.* SimSat (Sat) and SimImp (Imp) results on TOFU with LLaMA-2-7B (ES Score). We consider 2 unlearning methods (GA, GD).

| Sec. 4.1 | | Forget-1% | | | | Forget-5% | | | | Forget-10% | | | |
|---|---|---|---|---|---|---|---|---|---|---|---|---|---|
| | | ES-exact | | ES-perturb | | ES-exact | | ES-perturb | | ES-exact | | ES-perturb | |
| Method | $\beta$ | retain ↑ | unlearn ↓ | retain ↑ | unlearn ↓ | retain ↑ | unlearn ↓ | retain ↑ | unlearn ↓ | retain ↑ | unlearn ↓ | retain ↑ | unlearn ↓ |
| Sat-GA | 0.01 | 0.3852 | 0.0559 | 0.2632 | 0.0537 | 0.0000 | 0.0000 | 0.0000 | 0.0000 | 0.0000 | 0.0000 | 0.0000 | 0.0000 |
| | 0.10 | 0.3592 | 0.0466 | 0.2473 | 0.0404 | 0.0000 | 0.0000 | 0.0000 | 0.0000 | 0.0000 | 0.0000 | 0.0000 | 0.0000 |
| | 0.30 | 0.4419 | 0.0384 | 0.2906 | 0.0354 | 0.0000 | 0.0000 | 0.0000 | 0.0000 | 0.0000 | 0.0000 | 0.0000 | 0.0000 |
| | 0.50 | 0.4636 | 0.0335 | 0.3095 | 0.0327 | 0.0000 | 0.0000 | 0.0000 | 0.0000 | 0.0000 | 0.0000 | 0.0000 | 0.0000 |
| | 0.70 | 0.4947 | 0.0266 | 0.3314 | 0.0261 | 0.0000 | 0.0000 | 0.0000 | 0.0000 | 0.0000 | 0.0000 | 0.0000 | 0.0000 |
| | 1.00 | 0.4801 | 0.0186 | 0.3202 | 0.0189 | 0.0007 | 0.0065 | 0.0000 | 0.0000 | 0.0018 | 0.0035 | 0.0000 | 0.0000 |
| | 2.00 | 0.4411 | 0.0307 | 0.3104 | 0.0308 | 0.0001 | 0.0097 | 0.0005 | 0.0096 | 0.0015 | 0.0003 | 0.0013 | 0.0005 |
| | 3.00 | 0.4487 | 0.0314 | 0.2965 | 0.0326 | 0.0000 | 0.0000 | 0.0000 | 0.0000 | 0.0037 | 0.0044 | 0.0039 | 0.0040 |
| | 4.00 | 0.4519 | 0.0338 | 0.3011 | 0.0354 | 0.0000 | 0.0000 | 0.0000 | 0.0000 | 0.0002 | 0.0006 | 0.0003 | 0.0003 |
| | 5.00 | 0.4626 | 0.0484 | 0.3062 | 0.0405 | 0.0005 | 0.0000 | 0.0004 | 0.0000 | 0.0000 | 0.0000 | 0.0000 | 0.0000 |
| Sat-GD | 0.01 | 0.4396 | 0.0603 | 0.2921 | 0.0599 | 0.2766 | 0.0002 | 0.2242 | 0.0002 | 0.2578 | 0.0000 | 0.1995 | 0.0000 |
| | 0.10 | 0.5026 | 0.0549 | 0.3169 | 0.0570 | 0.4307 | 0.0000 | 0.3271 | 0.0000 | 0.5062 | 0.0000 | 0.3434 | 0.0000 |
| | 0.30 | 0.6037 | 0.0498 | 0.3716 | 0.0433 | 0.5396 | 0.0000 | 0.3633 | 0.0000 | 0.5977 | 0.0000 | 0.3794 | 0.0000 |
| | 0.50 | 0.6793 | 0.0449 | 0.4081 | 0.0404 | 0.5933 | 0.0002 | 0.3774 | 0.0001 | 0.6567 | 0.0000 | 0.4036 | 0.0000 |
| | 0.70 | 0.7242 | 0.0405 | 0.4315 | 0.0378 | 0.6360 | 0.0000 | 0.3880 | 0.0000 | 0.6600 | 0.0012 | 0.4130 | 0.0015 |
| | 1.00 | 0.7603 | 0.0545 | 0.4558 | 0.0496 | 0.6649 | 0.0002 | 0.4052 | 0.0000 | 0.6471 | 0.0033 | 0.4110 | 0.0036 |
| | 2.00 | 0.8049 | 0.0681 | 0.4808 | 0.0537 | 0.6630 | 0.0010 | 0.4087 | 0.0014 | 0.6732 | 0.0018 | 0.4129 | 0.0028 |
| | 3.00 | 0.8137 | 0.0909 | 0.4908 | 0.0541 | 0.6673 | 0.0083 | 0.4243 | 0.0088 | 0.6369 | 0.0029 | 0.4119 | 0.0030 |
| | 4.00 | 0.8259 | 0.0919 | 0.4891 | 0.0625 | 0.6771 | 0.0109 | 0.4156 | 0.0128 | 0.6431 | 0.0120 | 0.4061 | 0.0126 |
| | 5.00 | 0.8227 | 0.0970 | 0.4946 | 0.0662 | 0.6724 | 0.0131 | 0.4155 | 0.0134 | 0.6247 | 0.0253 | 0.3900 | 0.0252 |
| Imp-GA | 0.01 | 0.3955 | 0.0605 | 0.2639 | 0.0605 | 0.0267 | 0.0255 | 0.0267 | 0.0255 | 0.0000 | 0.0000 | 0.0000 | 0.0000 |
| | 0.10 | 0.3722 | 0.0966 | 0.2537 | 0.0734 | 0.0267 | 0.0255 | 0.0267 | 0.0255 | 0.0267 | 0.0252 | 0.0267 | 0.0252 |
| | 0.30 | 0.3474 | 0.1166 | 0.2385 | 0.0852 | 0.0267 | 0.0255 | 0.0267 | 0.0255 | 0.0267 | 0.0252 | 0.0267 | 0.0252 |
| | 0.50 | 0.3361 | 0.1426 | 0.2392 | 0.1379 | 0.0267 | 0.0255 | 0.0267 | 0.0255 | 0.0267 | 0.0252 | 0.0267 | 0.0252 |
| | 0.70 | 0.3249 | 0.1493 | 0.2353 | 0.1459 | 0.0267 | 0.0255 | 0.0267 | 0.0255 | 0.0267 | 0.0252 | 0.0267 | 0.0252 |
| | 1.00 | 0.3217 | 0.1553 | 0.2350 | 0.1313 | 0.0000 | 0.0000 | 0.0000 | 0.0000 | 0.0267 | 0.0252 | 0.0267 | 0.0252 |
| | 2.00 | 0.2934 | 0.1485 | 0.2269 | 0.1206 | 0.0000 | 0.0000 | 0.0000 | 0.0000 | 0.0267 | 0.0252 | 0.0267 | 0.0252 |
| | 3.00 | 0.2964 | 0.1552 | 0.2248 | 0.1249 | 0.0267 | 0.0255 | 0.0267 | 0.0255 | 0.0267 | 0.0252 | 0.0267 | 0.0252 |
| | 4.00 | 0.3011 | 0.1606 | 0.2319 | 0.1328 | 0.0000 | 0.0000 | 0.0000 | 0.0000 | 0.0267 | 0.0252 | 0.0267 | 0.0252 |
| | 5.00 | 0.3047 | 0.1639 | 0.2351 | 0.1383 | 0.0000 | 0.0000 | 0.0000 | 0.0000 | 0.0267 | 0.0252 | 0.0267 | 0.0252 |
| Imp-GD | 0.01 | 0.4401 | 0.0752 | 0.2843 | 0.0611 | 0.1723 | 0.0000 | 0.1528 | 0.0000 | 0.1775 | 0.0000 | 0.1478 | 0.0000 |
| | 0.10 | 0.4247 | 0.1112 | 0.2741 | 0.1132 | 0.2703 | 0.0000 | 0.2179 | 0.0000 | 0.2060 | 0.0000 | 0.1791 | 0.0000 |
| | 0.30 | 0.4295 | 0.1377 | 0.2908 | 0.1323 | 0.3257 | 0.0000 | 0.2423 | 0.0000 | 0.2248 | 0.0000 | 0.1929 | 0.0000 |
| | 0.50 | 0.4445 | 0.1676 | 0.2974 | 0.1444 | 0.3294 | 0.0000 | 0.2448 | 0.0000 | 0.1428 | 0.0000 | 0.1357 | 0.0000 |
| | 0.70 | 0.4671 | 0.1905 | 0.3159 | 0.1481 | 0.3592 | 0.0000 | 0.2671 | 0.0000 | 0.1926 | 0.0000 | 0.1735 | 0.0000 |
| | 1.00 | 0.5113 | 0.2224 | 0.3396 | 0.1997 | 0.3135 | 0.0000 | 0.2098 | 0.0000 | 0.2262 | 0.0000 | 0.1889 | 0.0000 |
| | 2.00 | 0.5304 | 0.3104 | 0.3631 | 0.2312 | 0.3546 | 0.0000 | 0.2568 | 0.0000 | 0.2345 | 0.0000 | 0.2021 | 0.0000 |
| | 3.00 | 0.6531 | 0.4202 | 0.4221 | 0.2727 | 0.3938 | 0.0000 | 0.2823 | 0.0000 | 0.2768 | 0.0000 | 0.2166 | 0.0000 |
| | 4.00 | 0.6909 | 0.4967 | 0.4404 | 0.3049 | 0.3839 | 0.0000 | 0.2855 | 0.0000 | 0.2508 | 0.0000 | 0.1986 | 0.0000 |
| | 5.00 | 0.7290 | 0.5579 | 0.4473 | 0.3408 | 0.3864 | 0.0000 | 0.2788 | 0.0000 | 0.2859 | 0.0000 | 0.2389 | 0.0000 |

*Table 10.* SimSat (Sat) and SimImp (Imp) results on TOFU with Phi-1.5 (ES Score). We consider 2 unlearning methods (GA, GD).

| Section 4.1 | | Forget-1% | | | | Forget-5% | | | | Forget-10% | | | |
|---|---|---|---|---|---|---|---|---|---|---|---|---|---|
| | | ES-exact | | ES-perturb | | ES-exact | | ES-perturb | | ES-exact | | ES-perturb | |
| Method | $\beta$ | retain ↑ | unlearn ↓ | retain ↑ | unlearn ↓ | retain ↑ | unlearn ↓ | retain ↑ | unlearn ↓ | retain ↑ | unlearn ↓ | retain ↑ | unlearn ↓ |
| Sat-GA | 0.01 | 0.1784 | 0.0369 | 0.1208 | 0.0577 | 0.0000 | 0.0000 | 0.0000 | 0.0000 | 0.0000 | 0.0000 | 0.0000 | 0.0000 |
| | 0.10 | 0.1680 | 0.0317 | 0.1149 | 0.0271 | 0.0000 | 0.0000 | 0.0000 | 0.0000 | 0.0000 | 0.0000 | 0.0000 | 0.0000 |
| | 0.30 | 0.1641 | 0.0188 | 0.1194 | 0.0120 | 0.0000 | 0.0000 | 0.0000 | 0.0000 | 0.0000 | 0.0000 | 0.0000 | 0.0000 |
| | 0.50 | 0.1408 | 0.0089 | 0.1108 | 0.0096 | 0.0000 | 0.0000 | 0.0000 | 0.0000 | 0.0000 | 0.0000 | 0.0000 | 0.0000 |
| | 0.70 | 0.1334 | 0.0069 | 0.1083 | 0.0056 | 0.0000 | 0.0000 | 0.0000 | 0.0000 | 0.0000 | 0.0000 | 0.0000 | 0.0000 |
| | 1.00 | 0.1222 | 0.0069 | 0.1040 | 0.0056 | 0.0000 | 0.0000 | 0.0000 | 0.0000 | 0.0000 | 0.0000 | 0.0000 | 0.0000 |
| | 2.00 | 0.1047 | 0.0062 | 0.0865 | 0.0061 | 0.0000 | 0.0000 | 0.0000 | 0.0000 | 0.0000 | 0.0000 | 0.0000 | 0.0000 |
| | 3.00 | 0.1040 | 0.0062 | 0.0832 | 0.0062 | 0.0000 | 0.0000 | 0.0000 | 0.0000 | 0.0000 | 0.0000 | 0.0000 | 0.0000 |
| | 4.00 | 0.0999 | 0.0055 | 0.0822 | 0.0062 | 0.0000 | 0.0000 | 0.0000 | 0.0000 | 0.0000 | 0.0000 | 0.0000 | 0.0000 |
| | 5.00 | 0.0998 | 0.0062 | 0.0848 | 0.0062 | 0.0000 | 0.0000 | 0.0000 | 0.0000 | 0.0000 | 0.0000 | 0.0000 | 0.0000 |
| Sat-GD | 0.01 | 0.2200 | 0.0311 | 0.1326 | 0.0425 | 0.1601 | 0.0001 | 0.1308 | 0.0001 | 0.1911 | 0.0000 | 0.1412 | 0.0000 |
| | 0.10 | 0.2693 | 0.0257 | 0.1540 | 0.0381 | 0.2988 | 0.0000 | 0.1824 | 0.0000 | 0.3169 | 0.0000 | 0.1933 | 0.0000 |
| | 0.30 | 0.3920 | 0.0507 | 0.1864 | 0.0282 | 0.3607 | 0.0000 | 0.2184 | 0.0000 | 0.4013 | 0.0000 | 0.2078 | 0.0000 |
| | 0.50 | 0.4499 | 0.0492 | 0.2117 | 0.0255 | 0.3924 | 0.0010 | 0.2256 | 0.0011 | 0.4089 | 0.0001 | 0.2173 | 0.0001 |
| | 0.70 | 0.4917 | 0.0535 | 0.2242 | 0.0301 | 0.4362 | 0.0111 | 0.2334 | 0.0129 | 0.4378 | 0.0000 | 0.2300 | 0.0000 |
| | 1.00 | 0.5309 | 0.0367 | 0.2419 | 0.0333 | 0.4824 | 0.0202 | 0.2401 | 0.0204 | 0.4429 | 0.0020 | 0.2389 | 0.0017 |
| | 2.00 | 0.5835 | 0.0494 | 0.2681 | 0.0463 | 0.5219 | 0.0374 | 0.2604 | 0.0396 | 0.4655 | 0.0282 | 0.2382 | 0.0305 |
| | 3.00 | 0.6036 | 0.0528 | 0.2771 | 0.0551 | 0.5419 | 0.0532 | 0.2631 | 0.0532 | 0.4798 | 0.0462 | 0.2400 | 0.0508 |
| | 4.00 | 0.6071 | 0.0636 | 0.2795 | 0.0607 | 0.5406 | 0.0663 | 0.2657 | 0.0691 | 0.4783 | 0.0594 | 0.2423 | 0.0646 |
| | 5.00 | 0.6134 | 0.0872 | 0.2791 | 0.0629 | 0.5479 | 0.0776 | 0.2599 | 0.0851 | 0.4907 | 0.0825 | 0.2455 | 0.0754 |
| Imp-GA | 0.01 | 0.1835 | 0.0374 | 0.1245 | 0.0577 | 0.0000 | 0.0000 | 0.0000 | 0.0000 | 0.0000 | 0.0000 | 0.0000 | 0.0000 |
| | 0.10 | 0.1788 | 0.0374 | 0.1239 | 0.0692 | 0.0000 | 0.0000 | 0.0000 | 0.0000 | 0.0000 | 0.0000 | 0.0000 | 0.0000 |
| | 0.30 | 0.1831 | 0.0618 | 0.1301 | 0.0718 | 0.0000 | 0.0000 | 0.0000 | 0.0000 | 0.0000 | 0.0000 | 0.0000 | 0.0000 |
| | 0.50 | 0.1756 | 0.0850 | 0.1256 | 0.0746 | 0.0000 | 0.0000 | 0.0000 | 0.0000 | 0.0000 | 0.0000 | 0.0000 | 0.0000 |
| | 0.70 | 0.1666 | 0.0850 | 0.1220 | 0.0753 | 0.0000 | 0.0000 | 0.0000 | 0.0000 | 0.0329 | 0.0305 | 0.0329 | 0.0305 |
| | 1.00 | 0.1617 | 0.0922 | 0.1222 | 0.0961 | 0.0000 | 0.0000 | 0.0000 | 0.0000 | 0.0329 | 0.0305 | 0.0329 | 0.0305 |
| | 2.00 | 0.1720 | 0.0935 | 0.1307 | 0.0790 | 0.0329 | 0.0307 | 0.0329 | 0.0307 | 0.0329 | 0.0305 | 0.0329 | 0.0305 |
| | 3.00 | 0.1917 | 0.0986 | 0.1442 | 0.0818 | 0.0329 | 0.0307 | 0.0329 | 0.0307 | 0.0329 | 0.0305 | 0.0329 | 0.0305 |
| | 4.00 | 0.2103 | 0.1089 | 0.1533 | 0.0886 | 0.0329 | 0.0307 | 0.0329 | 0.0307 | 0.0329 | 0.0305 | 0.0329 | 0.0305 |
| | 5.00 | 0.2242 | 0.1170 | 0.1591 | 0.1114 | 0.0329 | 0.0307 | 0.0329 | 0.0307 | 0.0329 | 0.0305 | 0.0329 | 0.0305 |
| Imp-GD | 0.01 | 0.2148 | 0.0295 | 0.1348 | 0.0441 | 0.1341 | 0.0001 | 0.1120 | 0.0001 | 0.1381 | 0.0000 | 0.1146 | 0.0000 |
| | 0.10 | 0.2258 | 0.0621 | 0.1375 | 0.0448 | 0.1074 | 0.0000 | 0.0916 | 0.0000 | 0.1219 | 0.0000 | 0.1105 | 0.0000 |
| | 0.30 | 0.2354 | 0.0865 | 0.1451 | 0.0550 | 0.1000 | 0.0000 | 0.0895 | 0.0000 | 0.1432 | 0.0000 | 0.1246 | 0.0000 |
| | 0.50 | 0.2473 | 0.0933 | 0.1527 | 0.0685 | 0.1147 | 0.0000 | 0.0934 | 0.0000 | 0.1362 | 0.0000 | 0.1258 | 0.0000 |
| | 0.70 | 0.2568 | 0.1086 | 0.1635 | 0.0771 | 0.1108 | 0.0000 | 0.0835 | 0.0000 | 0.1175 | 0.0000 | 0.0984 | 0.0000 |
| | 1.00 | 0.3030 | 0.1183 | 0.1760 | 0.0841 | 0.0946 | 0.0000 | 0.0799 | 0.0000 | 0.1201 | 0.0000 | 0.1028 | 0.0000 |
| | 2.00 | 0.3809 | 0.1516 | 0.2022 | 0.1170 | 0.1348 | 0.0000 | 0.1097 | 0.0000 | 0.1108 | 0.0000 | 0.0964 | 0.0000 |
| | 3.00 | 0.4459 | 0.2117 | 0.2219 | 0.1428 | 0.1243 | 0.0000 | 0.0950 | 0.0000 | 0.1242 | 0.0000 | 0.0977 | 0.0000 |
| | 4.00 | 0.4781 | 0.2623 | 0.2301 | 0.1844 | 0.0796 | 0.0000 | 0.0542 | 0.0000 | 0.1078 | 0.0000 | 0.0880 | 0.0000 |
| | 5.00 | 0.5075 | 0.3509 | 0.2382 | 0.1973 | 0.0988 | 0.0178 | 0.0685 | 0.0122 | 0.0899 | 0.0000 | 0.0835 | 0.0000 |

*Table 11.* Impact of weight granularity on TOFU with LLaMA-2-7B (SimSat). ES score is reported. We consider Batch-wise (B), Instance-wise (I), Group-wise (G), and Token-wise (T) reweighting with 2 unlearning methods (GA, GD).

| Section 4.1 | | Forget-1% | | | | Forget-5% | | | | Forget-10% | | | |
|---|---|---|---|---|---|---|---|---|---|---|---|---|---|
| | | ES-exact | | ES-perturb | | ES-exact | | ES-perturb | | ES-exact | | ES-perturb | |
| Method | $p$ | retain ↑ | unlearn ↓ | retain ↑ | unlearn ↓ | retain ↑ | unlearn ↓ | retain ↑ | unlearn ↓ | retain ↑ | unlearn ↓ | retain ↑ | unlearn ↓ |
| B-GA | 0.01 | 0.3668 | 0.0641 | 0.2520 | 0.0579 | 0.0515 | 0.0310 | 0.0521 | 0.0296 | 0.0289 | 0.0232 | 0.0291 | 0.0234 |
| | 0.10 | 0.4984 | 0.0838 | 0.3288 | 0.0748 | 0.1071 | 0.0481 | 0.1029 | 0.0487 | 0.0912 | 0.0643 | 0.0898 | 0.0665 |
| | 0.30 | 0.5807 | 0.1090 | 0.3864 | 0.0935 | 0.1655 | 0.0989 | 0.1453 | 0.0871 | 0.1322 | 0.1004 | 0.1211 | 0.0943 |
| | 0.50 | 0.5994 | 0.1086 | 0.4073 | 0.1052 | 0.1775 | 0.1122 | 0.1565 | 0.1013 | 0.1447 | 0.1069 | 0.1311 | 0.1040 |
| | 0.70 | 0.6083 | 0.1080 | 0.4216 | 0.1099 | 0.1872 | 0.1235 | 0.1641 | 0.1078 | 0.1541 | 0.1169 | 0.1375 | 0.1080 |
| I-GA | 0.01 | 0.3926 | 0.0645 | 0.2647 | 0.0603 | 0.0553 | 0.0352 | 0.0557 | 0.0342 | 0.0240 | 0.0215 | 0.0246 | 0.0210 |
| | 0.10 | 0.5423 | 0.0974 | 0.3648 | 0.0894 | 0.1155 | 0.0584 | 0.1107 | 0.0602 | 0.1028 | 0.0700 | 0.0999 | 0.0703 |
| | 0.30 | 0.6214 | 0.1119 | 0.4167 | 0.0998 | 0.1643 | 0.0993 | 0.1478 | 0.0866 | 0.1388 | 0.1005 | 0.1260 | 0.0943 |
| | 0.50 | 0.6593 | 0.1016 | 0.4307 | 0.1003 | 0.1736 | 0.1172 | 0.1561 | 0.0947 | 0.1559 | 0.1182 | 0.1398 | 0.1099 |
| | 0.70 | 0.6638 | 0.1492 | 0.4367 | 0.1329 | 0.1834 | 0.1198 | 0.1652 | 0.1024 | 0.1745 | 0.1212 | 0.1537 | 0.1127 |
| G-GA | 0.01 | 0.3963 | 0.0532 | 0.2609 | 0.0503 | 0.0267 | 0.0255 | 0.0267 | 0.0255 | 0.0000 | 0.0000 | 0.0000 | 0.0000 |
| | 0.10 | 0.3688 | 0.0483 | 0.2452 | 0.0476 | 0.0000 | 0.0000 | 0.0000 | 0.0000 | 0.0000 | 0.0000 | 0.0000 | 0.0000 |
| | 0.30 | 0.3814 | 0.0473 | 0.2643 | 0.0455 | 0.0000 | 0.0000 | 0.0000 | 0.0000 | 0.0000 | 0.0000 | 0.0000 | 0.0000 |
| | 0.50 | 0.4060 | 0.0483 | 0.2717 | 0.0455 | 0.0000 | 0.0000 | 0.0000 | 0.0000 | 0.0000 | 0.0000 | 0.0000 | 0.0000 |
| | 0.70 | 0.3971 | 0.0450 | 0.2728 | 0.0409 | 0.0000 | 0.0000 | 0.0000 | 0.0000 | 0.0000 | 0.0000 | 0.0000 | 0.0000 |
| T-GA | 0.01 | 0.3852 | 0.0559 | 0.2632 | 0.0537 | 0.0000 | 0.0000 | 0.0000 | 0.0000 | 0.0000 | 0.0000 | 0.0000 | 0.0000 |
| | 0.10 | 0.3592 | 0.0466 | 0.2473 | 0.0404 | 0.0000 | 0.0000 | 0.0000 | 0.0000 | 0.0000 | 0.0000 | 0.0000 | 0.0000 |
| | 0.30 | 0.4419 | 0.0384 | 0.2906 | 0.0354 | 0.0000 | 0.0000 | 0.0000 | 0.0000 | 0.0000 | 0.0000 | 0.0000 | 0.0000 |
| | 0.50 | 0.4636 | 0.0335 | 0.3095 | 0.0327 | 0.0000 | 0.0000 | 0.0000 | 0.0000 | 0.0000 | 0.0000 | 0.0000 | 0.0000 |
| | 0.70 | 0.4947 | 0.0266 | 0.3314 | 0.0261 | 0.0000 | 0.0000 | 0.0000 | 0.0000 | 0.0000 | 0.0000 | 0.0000 | 0.0000 |
| B-GD | 0.01 | 0.3711 | 0.0641 | 0.2606 | 0.0707 | 0.0896 | 0.0424 | 0.0883 | 0.0407 | 0.0723 | 0.0325 | 0.0741 | 0.0319 |
| | 0.10 | 0.5303 | 0.0854 | 0.3442 | 0.0826 | 0.1971 | 0.0527 | 0.1735 | 0.0465 | 0.1671 | 0.0190 | 0.1541 | 0.0202 |
| | 0.30 | 0.6111 | 0.1086 | 0.3974 | 0.0915 | 0.3205 | 0.1068 | 0.2618 | 0.0954 | 0.3326 | 0.0673 | 0.2528 | 0.0595 |
| | 0.50 | 0.6339 | 0.1024 | 0.4189 | 0.1095 | 0.3656 | 0.1225 | 0.2855 | 0.1051 | 0.4733 | 0.1066 | 0.3347 | 0.0858 |
| | 0.70 | 0.6580 | 0.1191 | 0.4303 | 0.1099 | 0.4416 | 0.1354 | 0.3145 | 0.1163 | 0.6314 | 0.1718 | 0.4100 | 0.1399 |
| I-GD | 0.01 | 0.4011 | 0.0682 | 0.2690 | 0.0598 | 0.0927 | 0.0364 | 0.0921 | 0.0361 | 0.0760 | 0.0370 | 0.0806 | 0.0372 |
| | 0.10 | 0.5731 | 0.0997 | 0.3788 | 0.0906 | 0.2178 | 0.0711 | 0.1899 | 0.0703 | 0.1898 | 0.0684 | 0.1681 | 0.0654 |
| | 0.30 | 0.6589 | 0.1129 | 0.4342 | 0.1019 | 0.3365 | 0.1154 | 0.2702 | 0.1024 | 0.5043 | 0.1334 | 0.3424 | 0.1163 |
| | 0.50 | 0.6948 | 0.1034 | 0.4477 | 0.1003 | 0.4443 | 0.1257 | 0.3131 | 0.1119 | 0.6762 | 0.1674 | 0.4253 | 0.1336 |
| | 0.70 | 0.7124 | 0.1488 | 0.4503 | 0.1289 | 0.5529 | 0.1476 | 0.3550 | 0.1185 | 0.7276 | 0.1831 | 0.4565 | 0.1457 |
| G-GD | 0.01 | 0.4454 | 0.0618 | 0.2882 | 0.0592 | 0.2761 | 0.0000 | 0.2200 | 0.0000 | 0.2784 | 0.0000 | 0.2110 | 0.0000 |
| | 0.10 | 0.4805 | 0.0624 | 0.3065 | 0.0616 | 0.4407 | 0.0000 | 0.3096 | 0.0000 | 0.4424 | 0.0000 | 0.3085 | 0.0000 |
| | 0.30 | 0.5505 | 0.0624 | 0.3463 | 0.0767 | 0.4961 | 0.0000 | 0.3408 | 0.0000 | 0.5865 | 0.0000 | 0.3753 | 0.0000 |
| | 0.50 | 0.5724 | 0.0653 | 0.3694 | 0.0644 | 0.5313 | 0.0000 | 0.3430 | 0.0000 | 0.6188 | 0.0004 | 0.3642 | 0.0004 |
| | 0.70 | 0.5792 | 0.0689 | 0.3744 | 0.0658 | 0.5293 | 0.0001 | 0.3516 | 0.0001 | 0.6052 | 0.0007 | 0.3986 | 0.0008 |
| T-GD | 0.01 | 0.4396 | 0.0603 | 0.2921 | 0.0599 | 0.2766 | 0.0002 | 0.2242 | 0.0002 | 0.2578 | 0.0000 | 0.1995 | 0.0000 |
| | 0.10 | 0.5026 | 0.0549 | 0.3169 | 0.0570 | 0.4307 | 0.0000 | 0.3271 | 0.0000 | 0.5062 | 0.0000 | 0.3434 | 0.0000 |
| | 0.30 | 0.6037 | 0.0498 | 0.3716 | 0.0433 | 0.5396 | 0.0000 | 0.3633 | 0.0000 | 0.5977 | 0.0000 | 0.3794 | 0.0000 |
| | 0.50 | 0.6793 | 0.0449 | 0.4081 | 0.0404 | 0.5933 | 0.0002 | 0.3774 | 0.0001 | 0.6567 | 0.0000 | 0.4036 | 0.0000 |
| | 0.70 | 0.7242 | 0.0405 | 0.4315 | 0.0378 | 0.6360 | 0.0000 | 0.3880 | 0.0000 | 0.6600 | 0.0012 | 0.4130 | 0.0015 |

*Table 12.* Impact of weight granularity on TOFU with Phi-1.5 (SimSat). ES score is reported. We consider Batch-wise (B), Instance-wise (I), Group-wise (G), and Token-wise (T) reweighting with 2 unlearning methods (GA, GD).

| Section 4.1 | | Forget-1% | | | | Forget-5% | | | | Forget-10% | | | |
|---|---|---|---|---|---|---|---|---|---|---|---|---|---|
| | | ES-exact | | ES-perturb | | ES-exact | | ES-perturb | | ES-exact | | ES-perturb | |
| Method | $\beta$ | retain ↑ | unlearn ↓ | retain ↑ | unlearn ↓ | retain ↑ | unlearn ↓ | retain ↑ | unlearn ↓ | retain ↑ | unlearn ↓ | retain ↑ | unlearn ↓ |
| B-GA | 0.01 | 0.2057 | 0.0589 | 0.1330 | 0.0591 | 0.0083 | 0.0047 | 0.0077 | 0.0046 | 0.0001 | 0.0001 | 0.0001 | 0.0001 |
| | 0.10 | 0.2725 | 0.0556 | 0.1850 | 0.0528 | 0.0477 | 0.0363 | 0.0428 | 0.0379 | 0.0340 | 0.0316 | 0.0332 | 0.0313 |
| | 0.30 | 0.3122 | 0.0589 | 0.1973 | 0.0563 | 0.0661 | 0.0570 | 0.0625 | 0.0524 | 0.0646 | 0.0528 | 0.0600 | 0.0500 |
| | 0.50 | 0.3306 | 0.0585 | 0.2048 | 0.0533 | 0.0830 | 0.0670 | 0.0719 | 0.0640 | 0.0683 | 0.0600 | 0.0632 | 0.0582 |
| | 0.70 | 0.3820 | 0.0650 | 0.2146 | 0.0630 | 0.0982 | 0.0751 | 0.0791 | 0.0699 | 0.0669 | 0.0642 | 0.0635 | 0.0621 |
| I-GA | 0.01 | 0.2118 | 0.0609 | 0.1412 | 0.0341 | 0.0070 | 0.0041 | 0.0067 | 0.0050 | 0.0008 | 0.0003 | 0.0009 | 0.0003 |
| | 0.10 | 0.2856 | 0.0596 | 0.1878 | 0.0492 | 0.0458 | 0.0346 | 0.0440 | 0.0348 | 0.0299 | 0.0297 | 0.0288 | 0.0286 |
| | 0.30 | 0.3575 | 0.0603 | 0.2060 | 0.0470 | 0.0645 | 0.0556 | 0.0627 | 0.0509 | 0.0577 | 0.0483 | 0.0514 | 0.0465 |
| | 0.50 | 0.3808 | 0.0467 | 0.2096 | 0.0483 | 0.0828 | 0.0601 | 0.0755 | 0.0532 | 0.0637 | 0.0538 | 0.0596 | 0.0536 |
| | 0.70 | 0.4115 | 0.0540 | 0.2183 | 0.0515 | 0.0937 | 0.0615 | 0.0857 | 0.0569 | 0.0716 | 0.0592 | 0.0662 | 0.0556 |
| G-GA | 0.01 | 0.1853 | 0.0369 | 0.1242 | 0.0577 | 0.0000 | 0.0000 | 0.0000 | 0.0000 | 0.0000 | 0.0000 | 0.0000 | 0.0000 |
| | 0.10 | 0.1724 | 0.0529 | 0.1150 | 0.0271 | 0.0000 | 0.0000 | 0.0000 | 0.0000 | 0.0000 | 0.0000 | 0.0000 | 0.0000 |
| | 0.30 | 0.1740 | 0.0450 | 0.1233 | 0.0259 | 0.0000 | 0.0000 | 0.0000 | 0.0000 | 0.0000 | 0.0000 | 0.0000 | 0.0000 |
| | 0.50 | 0.1784 | 0.0312 | 0.1286 | 0.0246 | 0.0000 | 0.0000 | 0.0000 | 0.0000 | 0.0000 | 0.0000 | 0.0000 | 0.0000 |
| | 0.70 | 0.1828 | 0.0263 | 0.1315 | 0.0195 | 0.0000 | 0.0000 | 0.0000 | 0.0000 | 0.0000 | 0.0000 | 0.0000 | 0.0000 |
| T-GA | 0.01 | 0.1784 | 0.0369 | 0.1208 | 0.0577 | 0.0000 | 0.0000 | 0.0000 | 0.0000 | 0.0000 | 0.0000 | 0.0000 | 0.0000 |
| | 0.10 | 0.1680 | 0.0317 | 0.1149 | 0.0271 | 0.0000 | 0.0000 | 0.0000 | 0.0000 | 0.0000 | 0.0000 | 0.0000 | 0.0000 |
| | 0.30 | 0.1641 | 0.0188 | 0.1194 | 0.0120 | 0.0000 | 0.0000 | 0.0000 | 0.0000 | 0.0000 | 0.0000 | 0.0000 | 0.0000 |
| | 0.50 | 0.1408 | 0.0089 | 0.1108 | 0.0096 | 0.0000 | 0.0000 | 0.0000 | 0.0000 | 0.0000 | 0.0000 | 0.0000 | 0.0000 |
| | 0.70 | 0.1334 | 0.0069 | 0.1083 | 0.0056 | 0.0000 | 0.0000 | 0.0000 | 0.0000 | 0.0000 | 0.0000 | 0.0000 | 0.0000 |
| B-GD | 0.01 | 0.2109 | 0.0604 | 0.1384 | 0.0341 | 0.0423 | 0.0296 | 0.0430 | 0.0287 | 0.0395 | 0.0185 | 0.0394 | 0.0186 |
| | 0.10 | 0.2992 | 0.0564 | 0.1858 | 0.0507 | 0.1344 | 0.0743 | 0.1151 | 0.0708 | 0.1491 | 0.0620 | 0.1212 | 0.0616 |
| | 0.30 | 0.3806 | 0.0589 | 0.2121 | 0.0559 | 0.2811 | 0.1074 | 0.1932 | 0.0935 | 0.4015 | 0.1127 | 0.2394 | 0.0933 |
| | 0.50 | 0.4692 | 0.0637 | 0.2330 | 0.0571 | 0.3965 | 0.1300 | 0.2237 | 0.0980 | 0.5486 | 0.1477 | 0.2634 | 0.1054 |
| | 0.70 | 0.5112 | 0.0800 | 0.2609 | 0.0598 | 0.5052 | 0.1272 | 0.2417 | 0.0910 | 0.5760 | 0.1685 | 0.2680 | 0.1186 |
| I-GD | 0.01 | 0.2190 | 0.0609 | 0.1393 | 0.0341 | 0.0445 | 0.0336 | 0.0448 | 0.0334 | 0.0445 | 0.0336 | 0.0456 | 0.0345 |
| | 0.10 | 0.3190 | 0.0579 | 0.1993 | 0.0509 | 0.1444 | 0.0737 | 0.1174 | 0.0678 | 0.1689 | 0.0688 | 0.1390 | 0.0691 |
| | 0.30 | 0.4784 | 0.0590 | 0.2349 | 0.0492 | 0.3228 | 0.1131 | 0.2191 | 0.0868 | 0.4401 | 0.1100 | 0.2521 | 0.0991 |
| | 0.50 | 0.5193 | 0.0604 | 0.2528 | 0.0446 | 0.4752 | 0.1146 | 0.2472 | 0.0914 | 0.5791 | 0.1478 | 0.2678 | 0.1111 |
| | 0.70 | 0.5476 | 0.0692 | 0.2567 | 0.0571 | 0.5401 | 0.1264 | 0.2626 | 0.0954 | 0.5924 | 0.1729 | 0.2769 | 0.1196 |
| G-GD | 0.01 | 0.2246 | 0.0310 | 0.1360 | 0.0425 | 0.1629 | 0.0000 | 0.1286 | 0.0000 | 0.1759 | 0.0000 | 0.1251 | 0.0000 |
| | 0.10 | 0.2682 | 0.0530 | 0.1525 | 0.0411 | 0.2656 | 0.0000 | 0.1741 | 0.0000 | 0.2877 | 0.0000 | 0.1725 | 0.0000 |
| | 0.30 | 0.3690 | 0.0581 | 0.1792 | 0.0264 | 0.3223 | 0.0000 | 0.2015 | 0.0000 | 0.3541 | 0.0000 | 0.1953 | 0.0000 |
| | 0.50 | 0.4145 | 0.0585 | 0.2004 | 0.0259 | 0.3405 | 0.0000 | 0.2149 | 0.0000 | 0.3867 | 0.0005 | 0.2192 | 0.0001 |
| | 0.70 | 0.4427 | 0.0603 | 0.2076 | 0.0280 | 0.3714 | 0.0008 | 0.2148 | 0.0010 | 0.3714 | 0.0008 | 0.2034 | 0.0007 |
| T-GD | 0.01 | 0.2200 | 0.0311 | 0.1326 | 0.0425 | 0.1601 | 0.0001 | 0.1308 | 0.0001 | 0.1911 | 0.0000 | 0.1412 | 0.0000 |
| | 0.10 | 0.2693 | 0.0257 | 0.1540 | 0.0381 | 0.2988 | 0.0000 | 0.1824 | 0.0000 | 0.3169 | 0.0000 | 0.1933 | 0.0000 |
| | 0.30 | 0.3920 | 0.0507 | 0.1864 | 0.0282 | 0.3607 | 0.0000 | 0.2184 | 0.0000 | 0.4013 | 0.0000 | 0.2078 | 0.0000 |
| | 0.50 | 0.4499 | 0.0492 | 0.2117 | 0.0255 | 0.3924 | 0.0010 | 0.2256 | 0.0011 | 0.4089 | 0.0001 | 0.2173 | 0.0001 |
| | 0.70 | 0.4917 | 0.0535 | 0.2242 | 0.0301 | 0.4362 | 0.0111 | 0.2334 | 0.0129 | 0.4378 | 0.0000 | 0.2300 | 0.0000 |

*Table 13.* Impact of weight granularity on TOFU with LLaMA-2-7B (SimImp). ES score is reported. We consider Batch-wise (B), Instance-wise (I), Group-wise (G), and Token-wise (T) reweighting with 2 unlearning methods (GA, GD).

| Section 4.1 | | Forget-1% | | | | Forget-5% | | | | Forget-10% | | | |
|---|---|---|---|---|---|---|---|---|---|---|---|---|---|
| | | ES-exact | | ES-perturb | | ES-exact | | ES-perturb | | ES-exact | | ES-perturb | |
| Method | $\beta$ | retain ↑ | unlearn ↓ | retain ↑ | unlearn ↓ | retain ↑ | unlearn ↓ | retain ↑ | unlearn ↓ | retain ↑ | unlearn ↓ | retain ↑ | unlearn ↓ |
| B-GA | 0.01 | 0.3872 | 0.0565 | 0.2610 | 0.0507 | 0.0002 | 0.0031 | 0.0002 | 0.0031 | 0.0000 | 0.0000 | 0.0000 | 0.0000 |
| | 0.10 | 0.3855 | 0.0565 | 0.2572 | 0.0527 | 0.0267 | 0.0255 | 0.0267 | 0.0255 | 0.0267 | 0.0255 | 0.0267 | 0.0255 |
| | 0.30 | 0.3854 | 0.0969 | 0.2568 | 0.0677 | 0.0267 | 0.0221 | 0.0267 | 0.0225 | 0.0267 | 0.0252 | 0.0267 | 0.0252 |
| | 0.50 | 0.3755 | 0.0954 | 0.2499 | 0.0649 | 0.0267 | 0.0255 | 0.0267 | 0.0255 | 0.0000 | 0.0000 | 0.0000 | 0.0000 |
| | 0.70 | 0.3595 | 0.0989 | 0.2498 | 0.0944 | 0.0267 | 0.0255 | 0.0267 | 0.0255 | 0.0267 | 0.0252 | 0.0267 | 0.0252 |
| I-GA | 0.01 | 0.3876 | 0.0565 | 0.2628 | 0.0528 | 0.0000 | 0.0000 | 0.0000 | 0.0000 | 0.0000 | 0.0000 | 0.0000 | 0.0000 |
| | 0.10 | 0.3835 | 0.0610 | 0.2586 | 0.0533 | 0.0267 | 0.0224 | 0.0267 | 0.0255 | 0.0267 | 0.0255 | 0.0267 | 0.0255 |
| | 0.30 | 0.3767 | 0.0974 | 0.2574 | 0.0675 | 0.0267 | 0.0255 | 0.0267 | 0.0255 | 0.0267 | 0.0255 | 0.0267 | 0.0255 |
| | 0.50 | 0.3553 | 0.0901 | 0.2535 | 0.0887 | 0.0267 | 0.0255 | 0.0267 | 0.0255 | 0.0000 | 0.0000 | 0.0000 | 0.0000 |
| | 0.70 | 0.3465 | 0.0947 | 0.2532 | 0.0718 | 0.0267 | 0.0255 | 0.0267 | 0.0255 | 0.0267 | 0.0255 | 0.0267 | 0.0241 |
| G-GA | 0.01 | 0.3931 | 0.0570 | 0.2661 | 0.0537 | 0.0000 | 0.0000 | 0.0000 | 0.0000 | 0.0000 | 0.0000 | 0.0000 | 0.0000 |
| | 0.10 | 0.4060 | 0.0601 | 0.2786 | 0.0595 | 0.0267 | 0.0225 | 0.0267 | 0.0224 | 0.0267 | 0.0243 | 0.0267 | 0.0242 |
| | 0.30 | 0.4220 | 0.0991 | 0.2810 | 0.0806 | 0.0267 | 0.0255 | 0.0267 | 0.0255 | 0.0267 | 0.0252 | 0.0267 | 0.0252 |
| | 0.50 | 0.4386 | 0.1247 | 0.2988 | 0.0984 | 0.0000 | 0.0000 | 0.0000 | 0.0000 | 0.0267 | 0.0252 | 0.0267 | 0.0252 |
| | 0.70 | 0.4235 | 0.1402 | 0.2909 | 0.1235 | 0.0000 | 0.0000 | 0.0000 | 0.0000 | 0.0267 | 0.0252 | 0.0267 | 0.0252 |
| T-GA | 0.01 | 0.3955 | 0.0605 | 0.2639 | 0.0605 | 0.0267 | 0.0255 | 0.0267 | 0.0255 | 0.0000 | 0.0000 | 0.0000 | 0.0000 |
| | 0.10 | 0.3722 | 0.0966 | 0.2537 | 0.0734 | 0.0267 | 0.0255 | 0.0267 | 0.0255 | 0.0267 | 0.0252 | 0.0267 | 0.0252 |
| | 0.30 | 0.3474 | 0.1166 | 0.2385 | 0.0852 | 0.0267 | 0.0255 | 0.0267 | 0.0255 | 0.0267 | 0.0252 | 0.0267 | 0.0252 |
| | 0.50 | 0.3361 | 0.1426 | 0.2392 | 0.1379 | 0.0267 | 0.0255 | 0.0267 | 0.0255 | 0.0267 | 0.0252 | 0.0267 | 0.0252 |
| | 0.70 | 0.3249 | 0.1493 | 0.2353 | 0.1459 | 0.0267 | 0.0255 | 0.0267 | 0.0255 | 0.0267 | 0.0252 | 0.0267 | 0.0252 |
| B-GD | 0.01 | 0.3828 | 0.0565 | 0.2612 | 0.0507 | 0.0000 | 0.0004 | 0.0000 | 0.0000 | 0.0000 | 0.0000 | 0.0000 | 0.0000 |
| | 0.10 | 0.3871 | 0.0565 | 0.2579 | 0.0517 | 0.0267 | 0.0255 | 0.0267 | 0.0255 | 0.0267 | 0.0255 | 0.0267 | 0.0239 |
| | 0.30 | 0.3836 | 0.0587 | 0.2554 | 0.0528 | 0.0267 | 0.0255 | 0.0267 | 0.0255 | 0.0267 | 0.0255 | 0.0267 | 0.0252 |
| | 0.50 | 0.3765 | 0.0975 | 0.2522 | 0.0666 | 0.0267 | 0.0255 | 0.0267 | 0.0255 | 0.0252 | 0.0267 | 0.0252 | 0.0267 |
| | 0.70 | 0.3607 | 0.0994 | 0.2493 | 0.0729 | 0.0267 | 0.0255 | 0.0267 | 0.0255 | 0.0267 | 0.0252 | 0.0267 | 0.0252 |
| I-GD | 0.01 | 0.3883 | 0.0605 | 0.2634 | 0.0556 | 0.0000 | 0.0000 | 0.0000 | 0.0000 | 0.0000 | 0.0000 | 0.0000 | 0.0000 |
| | 0.10 | 0.3834 | 0.0605 | 0.2600 | 0.0557 | 0.0267 | 0.0255 | 0.0267 | 0.0255 | 0.0267 | 0.0255 | 0.0267 | 0.0252 |
| | 0.30 | 0.3793 | 0.0974 | 0.2584 | 0.0687 | 0.0267 | 0.0255 | 0.0267 | 0.0255 | 0.0267 | 0.0255 | 0.0267 | 0.0255 |
| | 0.50 | 0.3569 | 0.0935 | 0.2557 | 0.0908 | 0.0267 | 0.0255 | 0.0267 | 0.0255 | 0.0267 | 0.0255 | 0.0267 | 0.0255 |
| | 0.70 | 0.3593 | 0.1072 | 0.2562 | 0.0726 | 0.0267 | 0.0255 | 0.0267 | 0.0255 | 0.0267 | 0.0255 | 0.0267 | 0.0255 |
| G-GD | 0.01 | 0.4436 | 0.0624 | 0.2861 | 0.0589 | 0.2365 | 0.0000 | 0.1954 | 0.0000 | 0.1933 | 0.0000 | 0.1736 | 0.0000 |
| | 0.10 | 0.4710 | 0.0794 | 0.3061 | 0.0816 | 0.2914 | 0.0000 | 0.2283 | 0.0000 | 0.2342 | 0.0000 | 0.2066 | 0.0000 |
| | 0.30 | 0.5427 | 0.1520 | 0.3329 | 0.1312 | 0.3442 | 0.0000 | 0.2642 | 0.0000 | 0.2467 | 0.0000 | 0.2016 | 0.0000 |
| | 0.50 | 0.6174 | 0.1991 | 0.3776 | 0.1622 | 0.3405 | 0.0000 | 0.2550 | 0.0000 | 0.2765 | 0.0000 | 0.2337 | 0.0000 |
| | 0.70 | 0.7160 | 0.2640 | 0.4073 | 0.2070 | 0.3340 | 0.0000 | 0.2603 | 0.0000 | 0.2710 | 0.0000 | 0.2242 | 0.0000 |
| T-GD | 0.01 | 0.4401 | 0.0752 | 0.2843 | 0.0611 | 0.1723 | 0.0000 | 0.1528 | 0.0000 | 0.1775 | 0.0000 | 0.1478 | 0.0000 |
| | 0.10 | 0.4247 | 0.1112 | 0.2741 | 0.1132 | 0.2703 | 0.0000 | 0.2179 | 0.0000 | 0.2060 | 0.0000 | 0.1791 | 0.0000 |
| | 0.30 | 0.4295 | 0.1377 | 0.2908 | 0.1323 | 0.3257 | 0.0000 | 0.2423 | 0.0000 | 0.2248 | 0.0000 | 0.1929 | 0.0000 |
| | 0.50 | 0.4445 | 0.1676 | 0.2974 | 0.1444 | 0.3294 | 0.0000 | 0.2448 | 0.0000 | 0.1428 | 0.0000 | 0.1357 | 0.0000 |
| | 0.70 | 0.4671 | 0.1905 | 0.3159 | 0.1481 | 0.3592 | 0.0000 | 0.2671 | 0.0000 | 0.1926 | 0.0000 | 0.1735 | 0.0000 |

*Table 14.* Impact of weighting granularity on TOFU with Phi-1.5 (SimImp). ES score is reported. We consider Batch-wise (B), Instance-wise (I), Group-wise (G), and Token-wise (T) reweighting with 2 unlearning methods (GA, GD).

| Section 4.1 | | Forget-1% | | | | Forget-5% | | | | Forget-10% | | | |
|---|---|---|---|---|---|---|---|---|---|---|---|---|---|
| | | ES-exact | | ES-perturb | | ES-exact | | ES-perturb | | ES-exact | | ES-perturb | |
| Method | $\beta$ | retain ↑ | unlearn ↓ | retain ↑ | unlearn ↓ | retain ↑ | unlearn ↓ | retain ↑ | unlearn ↓ | retain ↑ | unlearn ↓ | retain ↑ | unlearn ↓ |
| B-GA | 0.01 | 0.1835 | 0.0364 | 0.1232 | 0.0567 | 0.0000 | 0.0000 | 0.0000 | 0.0000 | 0.0000 | 0.0000 | 0.0000 | 0.0000 |
| | 0.10 | 0.1865 | 0.0364 | 0.1240 | 0.0371 | 0.0000 | 0.0000 | 0.0000 | 0.0000 | 0.0000 | 0.0000 | 0.0000 | 0.0000 |
| | 0.30 | 0.1833 | 0.0364 | 0.1246 | 0.0447 | 0.0000 | 0.0000 | 0.0000 | 0.0000 | 0.0000 | 0.0000 | 0.0000 | 0.0000 |
| | 0.50 | 0.1848 | 0.0373 | 0.1229 | 0.0447 | 0.0000 | 0.0000 | 0.0000 | 0.0000 | 0.0000 | 0.0000 | 0.0000 | 0.0000 |
| | 0.70 | 0.1871 | 0.0373 | 0.1240 | 0.0475 | 0.0000 | 0.0000 | 0.0000 | 0.0000 | 0.0000 | 0.0000 | 0.0000 | 0.0000 |
| I-GA | 0.01 | 0.1848 | 0.0364 | 0.1241 | 0.0332 | 0.0000 | 0.0000 | 0.0000 | 0.0000 | 0.0000 | 0.0000 | 0.0000 | 0.0000 |
| | 0.10 | 0.1847 | 0.0364 | 0.1237 | 0.0477 | 0.0000 | 0.0000 | 0.0000 | 0.0000 | 0.0000 | 0.0000 | 0.0000 | 0.0000 |
| | 0.30 | 0.1834 | 0.0364 | 0.1222 | 0.0477 | 0.0000 | 0.0000 | 0.0000 | 0.0000 | 0.0000 | 0.0000 | 0.0000 | 0.0000 |
| | 0.50 | 0.1838 | 0.0373 | 0.1236 | 0.0477 | 0.0000 | 0.0000 | 0.0000 | 0.0000 | 0.0000 | 0.0000 | 0.0000 | 0.0000 |
| | 0.70 | 0.1866 | 0.0604 | 0.1256 | 0.0486 | 0.0000 | 0.0000 | 0.0000 | 0.0000 | 0.0000 | 0.0000 | 0.0000 | 0.0000 |
| G-GA | 0.01 | 0.1885 | 0.0369 | 0.1256 | 0.0577 | 0.0000 | 0.0000 | 0.0000 | 0.0000 | 0.0000 | 0.0000 | 0.0000 | 0.0000 |
| | 0.10 | 0.1908 | 0.0383 | 0.1286 | 0.0707 | 0.0000 | 0.0000 | 0.0000 | 0.0000 | 0.0000 | 0.0000 | 0.0000 | 0.0000 |
| | 0.30 | 0.1886 | 0.0383 | 0.1308 | 0.0707 | 0.0000 | 0.0000 | 0.0000 | 0.0000 | 0.0000 | 0.0000 | 0.0000 | 0.0000 |
| | 0.50 | 0.1850 | 0.0620 | 0.1337 | 0.0721 | 0.0000 | 0.0000 | 0.0000 | 0.0000 | 0.0000 | 0.0000 | 0.0000 | 0.0000 |
| | 0.70 | 0.1802 | 0.0634 | 0.1366 | 0.0824 | 0.0000 | 0.0000 | 0.0000 | 0.0000 | 0.0000 | 0.0000 | 0.0000 | 0.0000 |
| T-GA | 0.01 | 0.1835 | 0.0374 | 0.1245 | 0.0577 | 0.0000 | 0.0000 | 0.0000 | 0.0000 | 0.0000 | 0.0000 | 0.0000 | 0.0000 |
| | 0.10 | 0.1788 | 0.0374 | 0.1239 | 0.0692 | 0.0000 | 0.0000 | 0.0000 | 0.0000 | 0.0000 | 0.0000 | 0.0000 | 0.0000 |
| | 0.30 | 0.1831 | 0.0618 | 0.1301 | 0.0718 | 0.0000 | 0.0000 | 0.0000 | 0.0000 | 0.0000 | 0.0000 | 0.0000 | 0.0000 |
| | 0.50 | 0.1756 | 0.0850 | 0.1256 | 0.0746 | 0.0000 | 0.0000 | 0.0000 | 0.0000 | 0.0000 | 0.0000 | 0.0000 | 0.0000 |
| | 0.70 | 0.1666 | 0.0850 | 0.1220 | 0.0753 | 0.0000 | 0.0000 | 0.0000 | 0.0000 | 0.0329 | 0.0305 | 0.0329 | 0.0305 |
| B-GD | 0.01 | 0.1823 | 0.0364 | 0.1219 | 0.0567 | 0.0000 | 0.0000 | 0.0000 | 0.0000 | 0.0000 | 0.0000 | 0.0000 | 0.0000 |
| | 0.10 | 0.1836 | 0.0364 | 0.1231 | 0.0332 | 0.0000 | 0.0000 | 0.0000 | 0.0000 | 0.0000 | 0.0000 | 0.0000 | 0.0000 |
| | 0.30 | 0.1848 | 0.0373 | 0.1200 | 0.0447 | 0.0000 | 0.0000 | 0.0000 | 0.0000 | 0.0000 | 0.0000 | 0.0000 | 0.0000 |
| | 0.50 | 0.1811 | 0.0373 | 0.1218 | 0.0447 | 0.0000 | 0.0000 | 0.0000 | 0.0000 | 0.0000 | 0.0000 | 0.0000 | 0.0000 |
| | 0.70 | 0.1843 | 0.0364 | 0.1218 | 0.0682 | 0.0000 | 0.0000 | 0.0000 | 0.0000 | 0.0000 | 0.0000 | 0.0000 | 0.0000 |
| I-GD | 0.01 | 0.1850 | 0.0364 | 0.1201 | 0.0682 | 0.0000 | 0.0000 | 0.0000 | 0.0000 | 0.0000 | 0.0000 | 0.0000 | 0.0000 |
| | 0.10 | 0.1813 | 0.0364 | 0.1229 | 0.0477 | 0.0000 | 0.0000 | 0.0000 | 0.0000 | 0.0000 | 0.0000 | 0.0000 | 0.0000 |
| | 0.30 | 0.1802 | 0.0373 | 0.1237 | 0.0682 | 0.0000 | 0.0000 | 0.0000 | 0.0000 | 0.0000 | 0.0000 | 0.0000 | 0.0000 |
| | 0.50 | 0.1829 | 0.0373 | 0.1228 | 0.0447 | 0.0000 | 0.0000 | 0.0000 | 0.0000 | 0.0000 | 0.0000 | 0.0000 | 0.0000 |
| | 0.70 | 0.1811 | 0.0604 | 0.1231 | 0.0447 | 0.0000 | 0.0000 | 0.0000 | 0.0000 | 0.0000 | 0.0000 | 0.0000 | 0.0000 |
| G-GD | 0.01 | 0.2181 | 0.1349 | 0.0425 | 0.0317 | 0.1400 | 0.0001 | 0.1167 | 0.0001 | 0.1325 | 0.0000 | 0.1155 | 0.0000 |
| | 0.10 | 0.2248 | 0.1419 | 0.0441 | 0.0604 | 0.1130 | 0.0000 | 0.0900 | 0.0000 | 0.1542 | 0.0000 | 0.1211 | 0.0000 |
| | 0.30 | 0.2458 | 0.1529 | 0.0703 | 0.0930 | 0.1078 | 0.0000 | 0.0937 | 0.0000 | 0.1327 | 0.0000 | 0.1067 | 0.0000 |
| | 0.50 | 0.2655 | 0.1669 | 0.0790 | 0.1071 | 0.1166 | 0.0000 | 0.0918 | 0.0000 | 0.1466 | 0.0000 | 0.1148 | 0.0000 |
| | 0.70 | 0.2955 | 0.1746 | 0.0900 | 0.1117 | 0.1194 | 0.0000 | 0.0924 | 0.0000 | 0.1441 | 0.0000 | 0.1132 | 0.0000 |
| T-GD | 0.01 | 0.2148 | 0.0295 | 0.1348 | 0.0441 | 0.1341 | 0.0001 | 0.1120 | 0.0001 | 0.1381 | 0.0000 | 0.1146 | 0.0000 |
| | 0.10 | 0.2258 | 0.0621 | 0.1375 | 0.0448 | 0.1074 | 0.0000 | 0.0916 | 0.0000 | 0.1219 | 0.0000 | 0.1105 | 0.0000 |
| | 0.30 | 0.2354 | 0.0865 | 0.1451 | 0.0550 | 0.1000 | 0.0000 | 0.0895 | 0.0000 | 0.1432 | 0.0000 | 0.1246 | 0.0000 |
| | 0.50 | 0.2473 | 0.0933 | 0.1527 | 0.0685 | 0.1147 | 0.0000 | 0.0934 | 0.0000 | 0.1362 | 0.0000 | 0.1258 | 0.0000 |
| | 0.70 | 0.2568 | 0.1086 | 0.1635 | 0.0771 | 0.1108 | 0.0000 | 0.0835 | 0.0000 | 0.1175 | 0.0000 | 0.0984 | 0.0000 |

*Table 15.* Hard sampling on TOFU with LLaMA-2-7B (ES Score). We consider TopK, BottomK (BotK), and Random (Rand) sampling with 2 unlearning method (GA, GD).

| Section 4.1 | | Forget-1% | | | | Forget-5% | | | | Forget-10% | | | |
| --- | --- | --- | --- | --- | --- | --- | --- | --- | --- | --- | --- | --- | --- |
| | | ES-exact | | ES-perturb | | ES-exact | | ES-perturb | | ES-exact | | ES-perturb | |
| Method | $p$ | retain ↑ | unlearn ↓ | retain ↑ | unlearn ↓ | retain ↑ | unlearn ↓ | retain ↑ | unlearn ↓ | retain ↑ | unlearn ↓ | retain ↑ | unlearn ↓ |
| | 0.1 | 0.4968 | 0.0638 | 0.3137 | 0.0635 | 0.2627 | 0.0000 | 0.2165 | 0.0000 | 0.2194 | 0.0241 | 0.1906 | 0.0241 |
| | 0.2 | 0.5649 | 0.0571 | 0.3392 | 0.0555 | 0.0258 | 0.0000 | 0.0294 | 0.0000 | 0.2407 | 0.0000 | 0.2060 | 0.0000 |
| | 0.3 | 0.5676 | 0.0878 | 0.3318 | 0.0698 | 0.3693 | 0.0000 | 0.2666 | 0.0000 | 0.2437 | 0.0000 | 0.1976 | 0.0000 |
| | 0.4 | 0.5775 | 0.0923 | 0.3454 | 0.0694 | 0.3901 | 0.0000 | 0.2872 | 0.0000 | 0.2941 | 0.0000 | 0.2340 | 0.0000 |
| Rand-GD | 0.5 | 0.5766 | 0.0880 | 0.3424 | 0.0854 | 0.2844 | 0.0000 | 0.2270 | 0.0000 | 0.2748 | 0.0002 | 0.1992 | 0.0002 |
| | 0.6 | 0.6459 | 0.1510 | 0.3840 | 0.1145 | 0.3872 | 0.0000 | 0.2782 | 0.0000 | 0.3181 | 0.0000 | 0.2434 | 0.0000 |
| | 0.7 | 0.7004 | 0.1142 | 0.4111 | 0.1164 | 0.4059 | 0.0000 | 0.3073 | 0.0000 | 0.2929 | 0.0000 | 0.2259 | 0.0000 |
| | 0.8 | 0.7609 | 0.3002 | 0.4324 | 0.2412 | 0.3979 | 0.0000 | 0.2858 | 0.0000 | 0.3323 | 0.0068 | 0.2423 | 0.0068 |
| | 0.9 | 0.7544 | 0.3187 | 0.4379 | 0.2290 | 0.4567 | 0.1545 | 0.2796 | 0.1154 | 0.3842 | 0.0112 | 0.2632 | 0.0112 |
| | 0.1 | 0.5556 | 0.1728 | 0.3520 | 0.1389 | 0.4925 | 0.1200 | 0.3311 | 0.1008 | 0.3221 | 0.0000 | 0.2441 | 0.0000 |
| | 0.2 | 0.4691 | 0.0888 | 0.3153 | 0.0699 | 0.4411 | 0.0000 | 0.3119 | 0.0000 | 0.2408 | 0.0000 | 0.1950 | 0.0000 |
| | 0.3 | 0.4548 | 0.0718 | 0.3072 | 0.0646 | 0.3938 | 0.0006 | 0.2850 | 0.0010 | 0.2502 | 0.0000 | 0.2103 | 0.0000 |
| | 0.4 | 0.4476 | 0.0596 | 0.3018 | 0.0628 | 0.3672 | 0.0000 | 0.2672 | 0.0007 | 0.2732 | 0.0241 | 0.2237 | 0.0244 |
| TopK-GD | 0.5 | 0.4473 | 0.0615 | 0.2991 | 0.0640 | 0.3362 | 0.0000 | 0.2583 | 0.0000 | 0.2719 | 0.0244 | 0.2276 | 0.0244 |
| | 0.6 | 0.4413 | 0.0615 | 0.2992 | 0.0628 | 0.3009 | 0.0000 | 0.2413 | 0.0007 | 0.3094 | 0.0218 | 0.2346 | 0.0218 |
| | 0.7 | 0.4464 | 0.0615 | 0.3024 | 0.0594 | 0.3261 | 0.0004 | 0.2501 | 0.0004 | 0.2835 | 0.0229 | 0.2221 | 0.0229 |
| | 0.8 | 0.4455 | 0.0615 | 0.2972 | 0.0613 | 0.2922 | 0.0009 | 0.2257 | 0.0007 | 0.2641 | 0.0210 | 0.2111 | 0.0210 |
| | 0.9 | 0.4504 | 0.0615 | 0.3001 | 0.0594 | 0.3036 | 0.0000 | 0.2532 | 0.0000 | 0.2691 | 0.0198 | 0.2096 | 0.0195 |
| | 0.9 | 0.7603 | 0.1674 | 0.4256 | 0.1015 | 0.4933 | 0.0000 | 0.3443 | 0.0000 | 0.2163 | 0.0001 | 0.1609 | 0.0001 |
| | 0.8 | 0.8392 | 0.7243 | 0.4992 | 0.3871 | 0.5639 | 0.0197 | 0.3353 | 0.0099 | 0.1838 | 0.0000 | 0.1438 | 0.0000 |
| | 0.7 | 0.8416 | 0.8369 | 0.4926 | 0.4595 | 0.6855 | 0.4138 | 0.4202 | 0.2372 | 0.3038 | 0.0498 | 0.2117 | 0.0373 |
| | 0.6 | 0.8414 | 0.8711 | 0.4951 | 0.4704 | 0.7131 | 0.6923 | 0.4360 | 0.3834 | 0.6038 | 0.4175 | 0.3798 | 0.2349 |
| BotK-GD | 0.5 | 0.8407 | 0.8711 | 0.4933 | 0.4700 | 0.7191 | 0.7130 | 0.4368 | 0.3823 | 0.6297 | 0.6389 | 0.3839 | 0.3459 |
| | 0.4 | 0.8425 | 0.8817 | 0.4929 | 0.4704 | 0.7312 | 0.7245 | 0.4361 | 0.4003 | 0.6180 | 0.6383 | 0.3864 | 0.3572 |
| | 0.3 | 0.8423 | 0.8711 | 0.4938 | 0.4700 | 0.7252 | 0.7284 | 0.4417 | 0.3969 | 0.6196 | 0.6558 | 0.3903 | 0.3541 |
| | 0.2 | 0.8414 | 0.8711 | 0.4949 | 0.4704 | 0.7158 | 0.7265 | 0.4347 | 0.3963 | 0.6363 | 0.6542 | 0.3907 | 0.3492 |
| | 0.1 | 0.8389 | 0.8711 | 0.4946 | 0.4704 | 0.7196 | 0.7301 | 0.4346 | 0.4009 | 0.6265 | 0.6490 | 0.3848 | 0.3458 |
| | 0.1 | 0.4098 | 0.1484 | 0.2703 | 0.1214 | 0.0267 | 0.0255 | 0.0267 | 0.0255 | 0.0267 | 0.0252 | 0.0267 | 0.0252 |
| | 0.2 | 0.4197 | 0.0811 | 0.2789 | 0.0678 | 0.0267 | 0.0255 | 0.0267 | 0.0255 | 0.0267 | 0.0252 | 0.0267 | 0.0252 |
| | 0.3 | 0.3919 | 0.0636 | 0.2613 | 0.0631 | 0.0267 | 0.0255 | 0.0267 | 0.0255 | 0.0267 | 0.0252 | 0.0267 | 0.0252 |
| | 0.4 | 0.3958 | 0.0534 | 0.2630 | 0.0566 | 0.0267 | 0.0255 | 0.0267 | 0.0255 | 0.0267 | 0.0252 | 0.0267 | 0.0252 |
| TopK-GA | 0.5 | 0.3918 | 0.0526 | 0.2632 | 0.0577 | 0.0267 | 0.0240 | 0.0267 | 0.0240 | 0.0267 | 0.0252 | 0.0267 | 0.0252 |
| | 0.6 | 0.3943 | 0.0559 | 0.2660 | 0.0527 | 0.0267 | 0.0248 | 0.0267 | 0.0248 | 0.0267 | 0.0252 | 0.0267 | 0.0252 |
| | 0.7 | 0.3841 | 0.0565 | 0.2578 | 0.0523 | 0.0267 | 0.0255 | 0.0267 | 0.0255 | 0.0000 | 0.0000 | 0.0000 | 0.0000 |
| | 0.8 | 0.3911 | 0.0573 | 0.2667 | 0.0561 | 0.0267 | 0.0255 | 0.0267 | 0.0255 | 0.0000 | 0.0000 | 0.0000 | 0.0000 |
| | 0.9 | 0.3901 | 0.0539 | 0.2595 | 0.0531 | 0.0267 | 0.0233 | 0.0267 | 0.0233 | 0.0000 | 0.0000 | 0.0000 | 0.0000 |
| | 0.9 | 0.4950 | 0.0209 | 0.3132 | 0.0167 | 0.0000 | 0.0000 | 0.0000 | 0.0000 | 0.0000 | 0.0000 | 0.0000 | 0.0000 |
| | 0.8 | 0.4083 | 0.0255 | 0.2717 | 0.0269 | 0.0000 | 0.0000 | 0.0000 | 0.0000 | 0.0000 | 0.0000 | 0.0000 | 0.0000 |
| | 0.7 | 0.4328 | 0.0174 | 0.2804 | 0.0191 | 0.0000 | 0.0000 | 0.0000 | 0.0000 | 0.0000 | 0.0000 | 0.0000 | 0.0000 |
| | 0.6 | 0.4532 | 0.0233 | 0.2836 | 0.0303 | 0.0000 | 0.0000 | 0.0000 | 0.0000 | 0.0000 | 0.0000 | 0.0000 | 0.0000 |
| BotK-GA | 0.5 | 0.5860 | 0.0168 | 0.3554 | 0.0184 | 0.0000 | 0.0000 | 0.0000 | 0.0000 | 0.0000 | 0.0000 | 0.0000 | 0.0000 |
| | 0.4 | 0.5943 | 0.0436 | 0.3768 | 0.0501 | 0.0000 | 0.0000 | 0.0000 | 0.0000 | 0.0000 | 0.0000 | 0.0000 | 0.0000 |
| | 0.3 | 0.6552 | 0.0804 | 0.4064 | 0.0826 | 0.0000 | 0.0000 | 0.0000 | 0.0000 | 0.0000 | 0.0000 | 0.0000 | 0.0000 |
| | 0.2 | 0.6131 | 0.0576 | 0.3771 | 0.0624 | 0.0000 | 0.0000 | 0.0000 | 0.0000 | 0.0000 | 0.0000 | 0.0000 | 0.0000 |
| | 0.1 | 0.8307 | 0.4028 | 0.5005 | 0.2959 | 0.0000 | 0.0000 | 0.0000 | 0.0000 | 0.0000 | 0.0000 | 0.0000 | 0.0000 |
| | 0.1 | 0.4022 | 0.0591 | 0.2660 | 0.0553 | 0.0000 | 0.0000 | 0.0000 | 0.0000 | 0.0000 | 0.0000 | 0.0000 | 0.0000 |
| | 0.2 | 0.4144 | 0.0589 | 0.2762 | 0.0587 | 0.0000 | 0.0000 | 0.0000 | 0.0000 | 0.0000 | 0.0000 | 0.0000 | 0.0000 |
| | 0.3 | 0.4144 | 0.0685 | 0.2804 | 0.0616 | 0.0000 | 0.0000 | 0.0000 | 0.0000 | 0.0000 | 0.0000 | 0.0000 | 0.0000 |
| | 0.4 | 0.4259 | 0.1053 | 0.2873 | 0.0825 | 0.0000 | 0.0000 | 0.0000 | 0.0000 | 0.0000 | 0.0000 | 0.0000 | 0.0000 |
| Rand-GA | 0.5 | 0.4092 | 0.1002 | 0.2884 | 0.0859 | 0.0000 | 0.0000 | 0.0000 | 0.0000 | 0.0000 | 0.0000 | 0.0000 | 0.0000 |
| | 0.6 | 0.5994 | 0.1167 | 0.3519 | 0.0688 | 0.0000 | 0.0000 | 0.0000 | 0.0000 | 0.0000 | 0.0000 | 0.0000 | 0.0000 |
| | 0.7 | 0.5762 | 0.1317 | 0.3687 | 0.1064 | 0.0000 | 0.0000 | 0.0000 | 0.0000 | 0.0000 | 0.0000 | 0.0000 | 0.0000 |
| | 0.8 | 0.6568 | 0.1752 | 0.4089 | 0.1373 | 0.0000 | 0.0000 | 0.0000 | 0.0000 | 0.0000 | 0.0000 | 0.0000 | 0.0000 |
| | 0.9 | 0.6938 | 0.2624 | 0.4317 | 0.2108 | 0.0000 | 0.0000 | 0.0000 | 0.0000 | 0.0000 | 0.0000 | 0.0000 | 0.0000 |

*Table 16.* Hard sampling on TOFU with Phi-1.5 (ES Score). We consider TopK, BottomK (BotK), and Random (Rand) sampling with 2 unlearning method (GA, GD).

| Section 4.1 | | Forget-1% | | | | Forget-5% | | | | Forget-10% | | | |
| --- | --- | --- | --- | --- | --- | --- | --- | --- | --- | --- | --- | --- | --- |
| | | ES-exact | | ES-perturb | | ES-exact | | ES-perturb | | ES-exact | | ES-perturb | |
| Method | p | retain ↑ | unlearn ↓ | retain ↑ | unlearn ↓ | retain ↑ | unlearn ↓ | retain ↑ | unlearn ↓ | retain ↑ | unlearn ↓ | retain ↑ | unlearn ↓ |
| | 0.1 | 0.2254 | 0.0624 | 0.1490 | 0.0737 | 0.1418 | 0.0001 | 0.1170 | 0.0001 | 0.1244 | 0.0000 | 0.0959 | 0.0000 |
| | 0.2 | 0.2431 | 0.0626 | 0.1557 | 0.0714 | 0.1387 | 0.0000 | 0.1218 | 0.0000 | 0.1204 | 0.0000 | 0.1010 | 0.0000 |
| | 0.3 | 0.2631 | 0.0660 | 0.1639 | 0.0763 | 0.1557 | 0.0000 | 0.1287 | 0.0000 | 0.1400 | 0.0000 | 0.1108 | 0.0000 |
| | 0.4 | 0.2922 | 0.0778 | 0.1786 | 0.0817 | 0.0979 | 0.0000 | 0.0680 | 0.0000 | 0.1400 | 0.0000 | 0.1071 | 0.0000 |
| Rand-GD | 0.5 | 0.3450 | 0.0819 | 0.1920 | 0.0769 | 0.1290 | 0.0000 | 0.0934 | 0.0000 | 0.1517 | 0.0032 | 0.1175 | 0.0034 |
| | 0.6 | 0.3567 | 0.0947 | 0.2014 | 0.0888 | 0.2151 | 0.0000 | 0.1405 | 0.0000 | 0.1821 | 0.0000 | 0.1364 | 0.0000 |
| | 0.7 | 0.4487 | 0.1340 | 0.2468 | 0.1280 | 0.1960 | 0.0163 | 0.1187 | 0.0161 | 0.1906 | 0.0000 | 0.1421 | 0.0000 |
| | 0.8 | 0.5201 | 0.1307 | 0.2537 | 0.1493 | 0.3832 | 0.1074 | 0.1899 | 0.0745 | 0.1904 | 0.0089 | 0.1303 | 0.0081 |
| | 0.9 | 0.5877 | 0.2520 | 0.2811 | 0.1816 | 0.4632 | 0.2184 | 0.2460 | 0.1358 | 0.4341 | 0.1727 | 0.2088 | 0.1267 |
| | 0.1 | 0.3129 | 0.1068 | 0.1878 | 0.1074 | 0.3004 | 0.0824 | 0.1947 | 0.0756 | 0.2151 | 0.0000 | 0.1495 | 0.0000 |
| | 0.2 | 0.2745 | 0.0902 | 0.1836 | 0.0958 | 0.1642 | 0.0312 | 0.1180 | 0.0312 | 0.1955 | 0.0000 | 0.1385 | 0.0000 |
| | 0.3 | 0.2437 | 0.0647 | 0.1506 | 0.0779 | 0.1619 | 0.0000 | 0.1137 | 0.0000 | 0.1649 | 0.0000 | 0.1358 | 0.0000 |
| | 0.4 | 0.2339 | 0.0598 | 0.1494 | 0.0665 | 0.1383 | 0.0001 | 0.1154 | 0.0001 | 0.1667 | 0.0305 | 0.1372 | 0.0305 |
| TopK-GD | 0.5 | 0.2287 | 0.0568 | 0.1454 | 0.0643 | 0.1397 | 0.0001 | 0.1160 | 0.0001 | 0.1823 | 0.0000 | 0.1396 | 0.0000 |
| | 0.6 | 0.2289 | 0.0568 | 0.1442 | 0.0521 | 0.1411 | 0.0001 | 0.1179 | 0.0001 | 0.1470 | 0.0000 | 0.1208 | 0.0000 |
| | 0.7 | 0.2301 | 0.0539 | 0.1456 | 0.0636 | 0.1430 | 0.0001 | 0.1192 | 0.0001 | 0.1466 | 0.0000 | 0.1234 | 0.0000 |
| | 0.8 | 0.2328 | 0.0568 | 0.1459 | 0.0521 | 0.1375 | 0.0001 | 0.1177 | 0.0001 | 0.1524 | 0.0305 | 0.1277 | 0.0305 |
| | 0.9 | 0.2303 | 0.0568 | 0.1451 | 0.0636 | 0.1265 | 0.0001 | 0.1114 | 0.0001 | 0.1344 | 0.0305 | 0.1153 | 0.0305 |
| | 0.9 | 0.2918 | 0.0377 | 0.1563 | 0.0427 | 0.1419 | 0.1357 | 0.1711 | 0.4148 | 0.0950 | 0.1239 | 0.0042 | 0.1211 |
| | 0.8 | 0.5429 | 0.1356 | 0.2321 | 0.0936 | 0.1357 | 0.4148 | 0.4226 | 0.5398 | 0.1239 | 0.0972 | 0.0027 | 0.0991 |
| | 0.7 | 0.5804 | 0.3823 | 0.2710 | 0.1628 | 0.4148 | 0.5398 | 0.5072 | 0.5337 | 0.1211 | 0.0991 | 0.3437 | 0.4818 |
| | 0.6 | 0.5937 | 0.4875 | 0.2708 | 0.1779 | 0.5398 | 0.5337 | 0.5355 | 0.5381 | 0.4818 | 0.2455 | 0.3437 | 0.4818 |
| BotK-GD | 0.5 | 0.5869 | 0.5245 | 0.2732 | 0.1726 | 0.5337 | 0.5381 | 0.5355 | 0.5381 | 0.4838 | 0.2491 | 0.4825 | 0.4838 |
| | 0.4 | 0.5859 | 0.5263 | 0.2731 | 0.1846 | 0.5381 | 0.5381 | 0.5332 | 0.5343 | 0.4826 | 0.2434 | 0.4872 | 0.4826 |
| | 0.3 | 0.5881 | 0.5378 | 0.2699 | 0.1846 | 0.5343 | 0.5340 | 0.5332 | 0.5343 | 0.4790 | 0.2482 | 0.4890 | 0.4790 |
| | 0.2 | 0.5892 | 0.5490 | 0.2749 | 0.1846 | 0.5340 | 0.5343 | 0.5287 | 0.5340 | 0.4744 | 0.2493 | 0.5038 | 0.4744 |
| | 0.1 | 0.5850 | 0.5691 | 0.2707 | 0.1810 | 0.5372 | 0.5372 | 0.5339 | 0.5372 | 0.4793 | 0.2460 | 0.4933 | 0.4793 |
| | 0.1 | 0.1860 | 0.0827 | 0.1362 | 0.0827 | 0.0000 | 0.0000 | 0.0000 | 0.0000 | 0.0000 | 0.0000 | 0.0000 | 0.0000 |
| | 0.2 | 0.1788 | 0.0549 | 0.1392 | 0.0758 | 0.0000 | 0.0000 | 0.0000 | 0.0000 | 0.0000 | 0.0000 | 0.0000 | 0.0000 |
| | 0.3 | 0.1867 | 0.0413 | 0.1346 | 0.0737 | 0.0000 | 0.0000 | 0.0000 | 0.0000 | 0.0000 | 0.0000 | 0.0000 | 0.0000 |
| | 0.4 | 0.1932 | 0.0382 | 0.1294 | 0.0715 | 0.0000 | 0.0000 | 0.0000 | 0.0000 | 0.0000 | 0.0000 | 0.0000 | 0.0000 |
| TopK-GA | 0.5 | 0.1865 | 0.0364 | 0.1267 | 0.0567 | 0.0000 | 0.0000 | 0.0000 | 0.0000 | 0.0000 | 0.0000 | 0.0000 | 0.0000 |
| | 0.6 | 0.1878 | 0.0364 | 0.1266 | 0.0567 | 0.0000 | 0.0000 | 0.0000 | 0.0000 | 0.0000 | 0.0000 | 0.0000 | 0.0000 |
| | 0.7 | 0.1893 | 0.0359 | 0.1250 | 0.0567 | 0.0000 | 0.0000 | 0.0000 | 0.0000 | 0.0000 | 0.0000 | 0.0000 | 0.0000 |
| | 0.8 | 0.1875 | 0.0364 | 0.1258 | 0.0567 | 0.0000 | 0.0000 | 0.0000 | 0.0000 | 0.0000 | 0.0000 | 0.0000 | 0.0000 |
| | 0.9 | 0.1896 | 0.0364 | 0.1248 | 0.0567 | 0.0000 | 0.0000 | 0.0000 | 0.0000 | 0.0000 | 0.0000 | 0.0000 | 0.0000 |
| | 0.9 | 0.0571 | 0.0014 | 0.0502 | 0.0014 | 0.0000 | 0.0000 | 0.0000 | 0.0000 | 0.0000 | 0.0000 | 0.0000 | 0.0000 |
| | 0.8 | 0.0569 | 0.0011 | 0.0397 | 0.0000 | 0.0000 | 0.0000 | 0.0000 | 0.0000 | 0.0000 | 0.0000 | 0.0000 | 0.0000 |
| | 0.7 | 0.1084 | 0.0101 | 0.0788 | 0.0060 | 0.0000 | 0.0000 | 0.0000 | 0.0000 | 0.0000 | 0.0000 | 0.0000 | 0.0000 |
| | 0.6 | 0.0751 | 0.0048 | 0.0568 | 0.0013 | 0.0000 | 0.0000 | 0.0000 | 0.0000 | 0.0000 | 0.0000 | 0.0000 | 0.0000 |
| BotK-GA | 0.5 | 0.1461 | 0.0243 | 0.1021 | 0.0119 | 0.0000 | 0.0000 | 0.0000 | 0.0000 | 0.0000 | 0.0000 | 0.0000 | 0.0000 |
| | 0.4 | 0.2134 | 0.0298 | 0.1281 | 0.0183 | 0.0000 | 0.0000 | 0.0000 | 0.0000 | 0.0000 | 0.0000 | 0.0000 | 0.0000 |
| | 0.3 | 0.2514 | 0.0642 | 0.1493 | 0.0304 | 0.0000 | 0.0000 | 0.0000 | 0.0000 | 0.0000 | 0.0000 | 0.0000 | 0.0000 |
| | 0.2 | 0.4632 | 0.0902 | 0.2199 | 0.0495 | 0.0000 | 0.0000 | 0.0000 | 0.0000 | 0.0000 | 0.0000 | 0.0000 | 0.0000 |
| | 0.1 | 0.4936 | 0.0998 | 0.2278 | 0.0868 | 0.0000 | 0.0000 | 0.0000 | 0.0000 | 0.0000 | 0.0000 | 0.0000 | 0.0000 |
| | 0.1 | 0.1886 | 0.0404 | 0.1282 | 0.0368 | 0.0000 | 0.0000 | 0.0000 | 0.0000 | 0.0000 | 0.0000 | 0.0000 | 0.0000 |
| | 0.2 | 0.1721 | 0.0267 | 0.1189 | 0.0339 | 0.0000 | 0.0000 | 0.0000 | 0.0000 | 0.0000 | 0.0000 | 0.0000 | 0.0000 |
| | 0.3 | 0.1818 | 0.0437 | 0.1207 | 0.0273 | 0.0000 | 0.0000 | 0.0000 | 0.0000 | 0.0000 | 0.0000 | 0.0000 | 0.0000 |
| | 0.4 | 0.1869 | 0.0479 | 0.1294 | 0.0367 | 0.0000 | 0.0000 | 0.0000 | 0.0000 | 0.0000 | 0.0000 | 0.0000 | 0.0000 |
| Rand-GA | 0.5 | 0.1952 | 0.0567 | 0.1370 | 0.0433 | 0.0000 | 0.0000 | 0.0000 | 0.0000 | 0.0000 | 0.0000 | 0.0000 | 0.0000 |
| | 0.6 | 0.2182 | 0.0954 | 0.1454 | 0.0633 | 0.0000 | 0.0000 | 0.0000 | 0.0000 | 0.0000 | 0.0000 | 0.0000 | 0.0000 |
| | 0.7 | 0.2294 | 0.0744 | 0.1518 | 0.0465 | 0.0000 | 0.0000 | 0.0000 | 0.0000 | 0.0000 | 0.0000 | 0.0000 | 0.0000 |
| | 0.8 | 0.2749 | 0.0852 | 0.1704 | 0.0825 | 0.0000 | 0.0000 | 0.0000 | 0.0000 | 0.0000 | 0.0000 | 0.0000 | 0.0000 |
| | 0.9 | 0.3179 | 0.0696 | 0.1576 | 0.0373 | 0.0000 | 0.0000 | 0.0000 | 0.0000 | 0.0000 | 0.0000 | 0.0000 | 0.0000 |

