# OpenReview forum: "Exploring Criteria of Loss Reweighting to Enhance LLM Unlearning"
_ICML.cc/2025/Conference — ICML 2025 poster_

### Official Review · Reviewer_3V34 · 2025-03-10

**Overall Recommendation:** 3

**Summary:**

The paper studies the effect of token-wise reweighting on gradient ascent type unlearning algorithms. They consider two kinds of reweighing: importance-based and saturation-based reweighing. They find that saturation is often more effective than importance-based reweighing and assigns lower weights to data with lower likelihoods while importance-based reweighting does the opposite. Based on their observation, the authors propose SatImp, which uses the product of importance weight and saturation weight in reweighing. They demonstrate the effectiveness of their method through extensive experiments on ToFU, WMDP and MUSE benchmarks.

**Claims And Evidence:**

The claims made in the submission are supported by clear and extensive experimental results.

**Essential References Not Discussed:**

As far as I am aware, the authors have not omitted essential references in their submission.

**Experimental Designs Or Analyses:**

The experiments look valid based on my understanding. However, the experiments are done without repetitions (i.e., the authors do not report the standard deviation (or error bar) of the performance of each method). For example, in Table 2, it is unclear whether SatImp outperforms existing methods as the improvement seems to be marginal.

**Methods And Evaluation Criteria:**

The authors use three popular unlearning benchmarks (TOFU, WMDP, MUSE) to evaluate the method. The choice of datasets are suitable for the problem at hand.

**Other Comments Or Suggestions:**

N/A

**Other Strengths And Weaknesses:**

The paper is well-written and extensive experiments are done to support the authors' findings. However, the method proposed in this work appears to be a straightforward combination of existing importance-based and saturation-based reweighting methods and does not seem to offer significant novelty. Moreover, as shown in Table 2,3, the improvement of SatImp over existing methods seems marginal and could be due to the randomness in evaluation/unlearning.

**Questions For Authors:**

In addition to the previous comments, I find the intuition "importance-based methods put more weights on tokens with less likelihood" a bit confusing. If my understanding were correct, in importance-based methods, a token is assigned a large weight if and only if it contains important information in the text.

**Relation To Broader Scientific Literature:**

N/A

**Theoretical Claims:**

N/A

---

> ### Author Rebuttal · Authors · 2025-04-01
>
> Many thanks for your constructive comments and suggestions! Please see our responses below.
>
> **Q1 The experiments are done without repetitions, and it is unclear whether SatImp has a marginal improvement.**
>
> Many thanks for your suggestions. Here we provide the standard deviations for part of Table 2 (due to the 5000 chars limitation), more results will be updated if our paper could be accepted. This is the results of LLaMA-2-7B.
> | Method| 1\% ES Re. | 1\% ES Un. | 1 \%  FQ  | 1\% MU | 5\% ES Re. | 5\% ES Un. | 5 \%  FQ  | 5\% MU | 10\% ES Re. | 10\% ES Un. | 10\%  FQ  | 10\% MU |
> | --------- | ------ | ------ | ------ | ------ | ------ | ------- | ------ | ------ | ------ | ------- | ------ | ------ |
> | GA      | 0.4747 ± 0.0158 | 0.1024 ± 0.0147 | 0.4127 ± 0.0082 | 0.5736 ± 0.0144 | 0.2212 ± 0.0148 | 0.0000 ± 0.0000 | 3.2e-11 ± 3.3e-12 | 0.5461 ± 0.0114 | 0.2216 ± 0.0118 | 0.0000 ± 0.0000 | 2.8e-30 ± 4.5e-31 | 0.5208 ± 0.0125 |
> | PO      | 0.7110 ± 0.0166 | 0.5031 ± 0.0136 | 0.1748 ± 0.0188 | 0.5310 ± 0.0070 | 0.6735 ± 0.0148 | 0.6054 ± 0.0181 | 7.3e-12 ± 5.5e-13 | 0.5564 ± 0.0134 | 0.6314 ± 0.0142 | 0.6124 ± 0.0106 | 6.3e-17 ± 3.2e-18 | 0.5174 ± 0.0121 |
> | DPO     | 0.6228 ± 0.0134 | 0.4000 ± 0.0211 | 0.0940 ± 0.0175 | 0.6053 ± 0.0145 | 0.5044 ± 0.0171 | 0.3846 ± 0.0130 | 4.4e-10 ± 6.2e-11 | 0.5759 ± 0.0108 | 0.6625 ± 0.0151 | 0.6174 ± 0.0126 | 7.1e-15 ± 4.9e-16 | 0.6158 ± 0.0188 |
> | NPO     | 0.6709 ± 0.0110 | 0.1106 ± 0.0150 | 0.5881 ± 0.0151 | 0.6341 ± 0.0122 | 0.5888 ± 0.0165 | 0.1150 ± 0.0209 | 0.0653 ± 0.0142 | 0.6197 ± 0.0185 | 0.6567 ± 0.0089 | 0.0941 ± 0.0141 | 0.0140 ± 0.0071 | 0.6267 ± 0.0147 |
> | SimNPO | 0.7933 ± 0.0069 | 0.2993 ± 0.0140 | 0.0027 ± 0.0012 | 0.6246 ± 0.0164 | 0.6742 ± 0.0165 | 0.2096 ± 0.0053 | 0.0002 ± 0.0001 | 0.6144 ± 0.0084 | 0.6175 ± 0.0145 | 0.1862 ± 0.0120 | 4.9e-6 ± 3.3e-7 | 0.5970 ± 0.0061 |
> | WGA     | 0.7539 ± 0.0074 | 0.0631 ± 0.0206 | 0.2708 ± 0.0123 | 0.6218 ± 0.0095 | 0.6700 ± 0.0156 | 0.0001 ± 0.0001 | 7.8e-8 ± 6.0e-9 | 0.6291 ± 0.0074 | 0.6453 ± 0.0150 | 0.0029 ± 0.0008 | 5.3e-17 ± 2.7e-18 | 0.6324 ± 0.0148 |
> | SatImp  |0.8091 ± 0.0208 | 0.0552 ± 0.0140 | 0.2583 ± 0.0103 | 0.6375 ± 0.0154 | 0.7000 ± 0.0106 | 0.0019 ± 0.0010 | 1.9e-17 ± 4.3e-18 | 0.6415 ± 0.0092 | 0.6706 ± 0.0175 | 0.0153 ± 0.0028 | 2.9e-18 ± 5.1e-19 | 0.6419 ± 0.0115 |
>
>
> Considering the marginal improvement problem, other reviewers are also concerned. Please note that we have mentioned the selection of hyper-parameter setting are fixed after the analysis on TOFU (In the Appendix Line 1152). The best preformance setting for ES score is typically not the best for other metrics.
> Thus, the performance gaps may be marginal on FQ\&MU of TOFU, Acc. on WMDP, and metrics on MUSE with such a setting.
> After adjust this setting, new results on WMDP and TOFU are shown in **Reviewer LUgf Q2** and **Reviewer 8Bwg Q2**.
>
> **Q2 SatImp is a simple combination of existing method thus this paper does not seem to offer significant novelty.**
>
> We sincerely appreciate your comments, but we respectfully disagree with your assessment regarding the lack of novelty of our work.
>
> As far as we know, we are the first that proposing the importance-based (IB) reweighting mechanism (i).
> There is no IB annotations in all benchmarks before this paper.
> We are the first to make such annotations and make it public (in the supplementary material, will be released once this paper being accepted).
>
> This paper is also the first that summarize existing methods with a universal saturation-based (SB) framework (ii). While mounts of literature have explore the reweighting in LLM unlearning, no one has made such a summary to figure out why they are effective from the reweighting perspective.
>
> Besides, we simplify IB and SB with a simple mapping -- SimImp (iii) and SimSat (iv).
> Finally, we propose SatImp, which further enhances the performance via above analysis.
> Among these five findings, only (iv) is encountered a contemporary work (please refer to our response in **Reviewer LUgf Q1**).
>
> Thus, we sincerely hope you can re-evaluate the contribution and novelty of our paper.
>
> **Q3 Confusion about the importance-based setting**
>
> We apologize for any confusion that may arise. Your understanding are correct that a token is assigned a large weight if and only if it contains important information. In fact, we have done this in Page 3 Line 160.
>
> As mentioned above, we are the first to make such annotations. However, considering that there are numerous data points without importance labels. It would be labor-intensive for human to annotate all of samples in existing datasets. Thus, we try to find an approximate representation of importance labels. We notice that it is high-correlated between weights and likelihoods (Figure 2(e), (g)). Thus, we simplify the importance-based and saturation-based methods as put more weights on tokens with lower and higher likelihood, respectively.

---

### Official Review · Reviewer_8Bwg · 2025-03-12

**Overall Recommendation:** 3

**Summary:**

This paper studies the loss reweighting mechanism for gradient difference-based LLM unlearning methods. It investigates the effect of importance-based reweighting and saturation-based reweighting on TOFU-1% setting. Based on the observations, a new reweight mechanism, SatImp, is proposed, and the results of the TOFU and WMDP experiments show some advantages.

**Claims And Evidence:**

**Claim 1:**

Importance-based reweighting is better at unlearning while saturation-based reweighting is better at knowledge retention.



*Evidence:* Experiments in Section 4.1 aim to answer this claim.



*Comment:* This experiment does not fully support the claim as it operates on a limited setting. Authors may include the performance for Imp only and Sat only in the main experiments shown in Table 2/3 to fully support this claim.



**Claim 2:**

Smoothness of the loss reweighting has a big impact on performance.



*Evidence:* experiments in section 4.2 aim to answer this claim.



*Comment:* Same as claim 1, I don't think that an experiment on a limited setting fully supports the claim.



**Claim 3:**

More fine-grained token-level reweighting is better than span/instance/group level reweighting.



*Evidence:* experiments in Section 4.2 aim to answer this claim.



*Comment:* Same as Claim 1/2.



**Claim 4:**

Soft reweighting is better than hard sampling reweighting



*Evidence:* Experiments in section 4.2 aim to answer this claim.



*Comment:* I don't think this is closely related to the paper. Besides, intuitively, TopK/BottomK hard-sampling is similar to some extreme cases for the shape of weight distribution, it can be covered by a study on the effect of varying shaped weight distribution (shown in Figure 5) on unlearning performance.



**Claim 5:**

SatImp is better than baseline methods like NPO/DPO/SimNPO in LLM unlearning.



*Evidence:* Experiments in section 5 aim to answer this claim.



*Comment:* I have two concerns for this claim.

​	1. The performance for baselines does not match numbers reported in other papers. For example, NPO/SimNPO are reported to achieve near 1 FQ and close to original MU (>0.5) on Forget-1%, and other higher numbers on Forget-5, 10% compared to numbers in Table 2.

​	2. SatImp is reported to achieve only 0.2491 MMLU performance for WMDP, which is close to random guessing. This indicates a heavy loss on the base LLM knowledge. More discussion is needed for this experiment.





**Other major concern:**

​	1. The hyper-parameter choosing. The hyper-parameter for Equation 8/9/11/12 does not have a consistent pattern. It seems very tricky to choose a good parameter for the proposed reweighting.

​	2. Sat reweighting in Equation 9 and 11 has two completely different forms. Is there any particular reason for this change?

**Essential References Not Discussed:**

* A work involving loss reweighting only on forget data

​	LLM Unlearning via Loss Adjustment with Only Forget Data

* And several works about logit reweighting:

​	Reversing the Forget-Retain Objectives: An Efficient LLM Unlearning Framework from Logit Difference

​	UNDIAL: Self-Distillation with Adjusted Logits for Robust Unlearning in Large Language Models

​	Offset Unlearning for Large Language Models

**Experimental Designs Or Analyses:**

Please see claims and evidence section for my comment on experiments.

**Methods And Evaluation Criteria:**

The evaluation is conducted on popular LLM unlearning datasets, including TOFU/WMDP/MUSE, and the evaluation criteria are standard. I think it is reliable.

**Other Comments Or Suggestions:**

* Algorithm 1 seems not too important as it does not clearly correlate to the proposed algorithm and can be moved to the appendix.

* Incorporating unlearning-retain performance trajectory along training would better illustrate the performance difference between different methods. And it is also a very popular illustration in previous LLM unlearning works.

* Incorporating the explanation about ES scores in main text instead of only in appendix is better for readers to digest the paper.

**Other Strengths And Weaknesses:**

Minor weakness:

1. Some notations are misleading, for example, Equation 8 use $p$ as the hyper-parameter, which may cause confusion because most of the cases $p$ in this paper refers to the token probability. And the variable names for the hyper-parameter used in Equations 8/9/10 are inconsistent, which may cause confusion. I would suggest using a consistent name like $\beta$, but differentiate with subscripts.
2. Some information missing for the figures. For example, what's the baseline in Figure 3 a-d?

**Questions For Authors:**

1. The proposed reweighting assigns a lower than 1 weight for the token losses, which downscale the GD loss. Is there any intuition on why this is useful?

**Relation To Broader Scientific Literature:**

The idea in this paper is related to other loss reweighting methods for LLM unlearning. The experiments aim to understand the mechnism of several reweighting methods and combine them to achieve a better balance between unlearning and knowledge retention.

**Theoretical Claims:**

No theoretical claims are made in this paper.

---

> ### Author Rebuttal · Authors · 2025-04-01
>
> Many thanks for your constructive comments and suggestions! Please see our responses below.
>
> **Q1 More experiment about Claims 1-4**
>
> Thanks for the detailed review.
>
> For Claim 1, we contain the relevant results in the Appendix Figure 13,14 (m)-(x). We will include the performance for SimImp in all tables.
>
> For Claim 2 and 3, more results are shown in the Appendix Figure 13-16.
>
> For Claim 4, the motivation that we include the discussion about hard-sampling and soft-weighting is we do not want to overlook any element in our quest to achieve better LLM unlearning performance.
> As mentioned in Line 324, there have been some concerns that hard sampling may outperforms soft weighting. Thus, it is necessary to figure it out. Only through such a thorough investigation can we derive reliable empirical conclusions. Our analysis can help subsequent work avoid detours.
>
> **Q2 Claim 5: Baseline is not match the numbers in other papers and the results on the WMDP should be discussed.**
>
> Regrading the WMDP results first, the response is shown in **Reviewer LUgf Q2**.
>
> Regarding the inconsistent results about FQ and MU, we indicate that the performance on ES score and FQ\&MU are not always aligning, which means when models obtain best performance on ES score, they may have mean performances on FQ\&MU. It is because the objectives of ES and FQ\&MU are different. Especially the FQ and ES Un., FQ requires the unlearning model towards a standard model yet ES Un. requires the model forget the unlearn data as much as possible.
>
> There are some literature that have mentioned this phenomenon. First, we find that WGA also mentions this discrimination.
> Second, in the paper 'LLM Unlearning via Loss Adjustment with Only Forget Data' that you mentioned as a essential reference, the FQ and MU are similiar as results in Table 2.
>
> In this paper, we have made comprehensive analysis on TOFU with ES score.
> Therefore, in Table 2, we report the results with ES score as the primary metric. All methods were selected with the goal of optimizing the ES score‌.
> Due to limited rebuttal length, here we report part of results which set the FQ and MU as the primary metrics. Results with retain regularization on LLaMA-2-7B are shown as follows:
>
> | Method| 1\% ES Re. | 1\% ES Un. | 1 \%  FQ  | 1\% MU |
> | --------- | ------ | ------ | ------ | ------ |
> | GA | 0.4086 | 0.0732 | 0.9188 | 0.5228|
> | NPO | 0.4586 | 0.0744 | 0.9188 | 0.5379 |
> | SimNPO | 0.4243 | 0.0822 | 0.9188 | 0.5332 |
> | WGA | 0.6037 | 0.0498 | 0.9900 | 0.5926 |
> | SatImp | 0.7242 | 0.0405 |0.9900 | 0.6350 |
>
>
> **Q3 Confusion about Eq. 8,9,11,12**
>
> We sincerely apologize to the confusion about the exploration process of our paper.
> First, Eq. 8 and 9 are our intuition about importance-based and saturation-based unlearning.
> Eq. 8. indicates that a token should be noticed if and only if it contains important information.
> Eq. 9 represents an objective, which summarizes all existing reweighting methods.
> For instance, Eq. 9 is equivalent to SimNPO when $\tau = \frac{|y|(P+1)}{2} - P$‌.
>
> However, considering that there are numerous training samples that without importance-based labels, it is difficult for researchers to labeling them all. Thus, we try to find an approximate representation for both unlearning branches.
> We notice it is high-correlated between weights and likelihoods (Figure 2(e)-(h)). Thus, we simplify two branches as put more weights on tokens with lower or higher likelihood, respectively.
> An intuition for this simplification is probability-based mapping, thus we propose SimImp and SimSat (Eq.11). We make a comprehensive investigation and then combine these two solutions (Eq. 12).
>
> **Q4 References**
>
> Thanks for providing several LLM unlearning methods, we will cite them and compare the performance. According to the results in these papers, none outperforms SatImp.
>
> **Q5 Typo and organization of the paper**
>
> Many thanks for your wonderful suggestions!
>
> 1. We will modify the $p$ in Eq. 8. $\beta$ has been used as the smoothness index in Eq. 9, 11, 12.
>
> 2. The baseline in Figure 3a-d is the vanilla GD, which is represented by the dashed line.
>
> 3. We will discuss and re-organize the paper about the position of Algorithm 1 and the explanation about ES scores.
>
> 4. Actually, we have made a lot figures that recording the retain-unlearn performance at different training steps. However, considering the length of our paper, these figures are not shown in the submission. We will include part of them if our paper is accepted.
>
> **Q6 Intuition about re-weighting a lower than 1 weight for the token losses**
> When we observe the training process of the vanilla GD, we notice that the unlearning process of the model is completed rapidly within just few steps after warm-up, even reaching an over-unlearning state. Such a swift change is detrimental to finding the optimal model‌.
> Smaller weights can prolong the unlearning, helping the model stop training at a more appropriate step.

---

> > ### Comment · Reviewer_8Bwg · 2025-04-02
> >
> > Thanks for the additional results. It's great to see that these performance improvements after changing the checkpoint selection criteria. And since this is a core issue, I would suggest it is necessary to include the forget-retain performance trade-off. trajectory  results in the paper.

---

> > > ### Author Response · Authors · 2025-04-04
> > >
> > > We sincerely appreciate your timely reply!
> > >
> > > The retain-unlearn trade-off is a crucial issue. As is well known, reweighting-based LLM unlearning methods are sensitive to hyper-parameters, which has already been mentioned in our paper (Appendix Line 1175).
> > >
> > > In fact, we have conducted relevant experiments on this trade-off (mentioned in the Appendix Line 816). The ES score-related content has been included in our supplementary materials (results.xlsx, under the SatImp table). Here, we provide additional results on FQ and MU, hoping that these findings will meet your new requirements for our work. Due to space constraints, we first present FQ and MU experiments on the Forget 1\% setting.
> > >
> > > First, we present the results with forget-only regularizations. FQ/MU denotes the reporting format.
> > >
> > > | $(\beta_1, \beta_2)$| LLaMA-2-7B| Phi-1.5 |
> > > | :--: | :--: | :--: |
> > > |  (1,0.05) | 0.7659/0.5890 | 0.0143/0.3886
> > > |  (1,0.1) | 0.7659/0.5861 | 0.0541/0.3918
> > > |  (1,0.2) | 0.9188/0.5833 | 0.1650/0.3891
> > > |  (1,0.5) | 0.9188/0.5920 | 0.5786/0.4140
> > > |  (1,1.0) | 0.4046/0.5863 | 0.4046/0.4022
> > > |  (2,0.05) |0.9188/0.5881 | 0.0286/0.3791
> > > |  (2,0.1) |0.9188/0.5847 | 0.0068/0.3824
> > > |  (2,0.2) |0.9188/0.5851 | 0.0541/0.3974
> > > |  (2,0.5) |0.7659/0.5773 | 0.2657/0.4110
> > > |  (2,1.0) |0.9188/0.5920 | 0.9188/0.4243
> > > |  (3,1.0) |0.4046/0.5864 | 0.7659/0.4181
> > > |  (3,2.0) |0.1650/0.5927 | 0.1650/0.4312
> > > |  (3,3.0) |0.0286/0.6135 | 0.4046/0.4378
> > > |  (5,1.0) |0.7659/0.5912 | 0.5786/0.4207
> > > |  (5,2.0) |0.4046/0.5801 | 0.7659/0.4391
> > > |  (5,3.0) |0.0286/0.6232 | 0.0971/0.4490
> > >
> > > Second, we present the results with retain regularizations. FQ/MU denotes the reporting format.
> > > | $(\beta_1, \beta_2, \lambda)$| LLaMA-2-7B| Phi-1.5 |
> > > | :--: | :--: | :--: |
> > > |  (1,0.05,0.1) | 0.7659/0.6027 | 0.0971/0.4793
> > > |  (1,0.05,0.2) |0.7659/0.6136 | 0.0971/0.4859
> > > |  (1,0.05,0.5) |0.7659/0.6177 | 0.9900/0.5170
> > > |  (1,0.1,0.1) | 0.7659/0.6014 | 0.0541/0.4789
> > > |  (1,0.1,0.2) | 0.9188/0.6107 | 0.2657/0.4864
> > > |  (1,0.1,0.5) | 0.7659/0.6160 | 0.7659/0.5144
> > > |  (5,0.05,0.1) |0.9188/0.6371 | 0.4046/0.5096
> > > |  (5,0.05,0.2) |0.9188/0.6323 | 0.9900/0.5187
> > > |  (5,0.05,0.5) |0.2657/0.6215 | 0.1650/0.5216
> > > |  (5,0.1,0.1) |0.9900/0.6428 | 0.7659/0.5122
> > > |  (5,0.1,0.2) |0.9900/0.6350 | 0.9900/0.5195
> > > |  (5,0.1,0.5) |0.2657/0.6277 | 0.0971/0.5248
> > >
> > > This represents only a portion of the hyperparameter experiments we have completed, we hope these results provide insights into addressing the core issue you've newly identified.
> > >
> > > Besides, we sincerely appreciate your meticulous, responsible, and highly professional feedback throughout the review process.
> > > At the same time, we sincerely hope our responses will merit your further support of this work.
> > > Should any additional clarification be needed, we stand ready to address it promptly.

---

### Official Review · Reviewer_LUgf · 2025-03-14

**Overall Recommendation:** 2

**Summary:**

This paper studies LLM unlearning. The authors investigate the loss reweighting mechanism for LLM unlearning, where each token in the forget set is assigned a different weight in loss calculation. Specifically, the authors propose two ideas for loss reweighting: saturation, which suggests that tokens that are sufficiently unlearned should be assigned smaller weights; and importance, which suggests that tokens that convey important information should have larger weights. The paper studies the influence of two reweighting on unlearned model performance and proposes a method that combines the two ideas into a single weight for unlearning. Experiments are conducted on three benchmarks to demonstrate the performance of the method.

**Claims And Evidence:**

Yes, the claims are supported by experiments.

**Essential References Not Discussed:**

No.

**Experimental Designs Or Analyses:**

Yes, I checked the experiment section.

**Methods And Evaluation Criteria:**

Most of the evaluation makes sense, except that the paper introduces the unlearning setting where the goal is to forget a subset of knowledge from the web-scale training corpora (i.e., pre-training data), but most of the experiments are conducted on TOFU, which is a synthetic dataset where knowledge to forget is learned in special fine-tuning instead of pre-training. Therefore, it's not clear whether the analyses can generalize to the more practical setting where pre-training knowledge needs to be unlearned.

**Other Comments Or Suggestions:**

N/A

**Other Strengths And Weaknesses:**

Weaknesses:

1. The proposed method does not seem to have a significant improvement over the baselines (TOFU 5% and 10% setting, and Table 3 on WMDP). Especially on retain performance, the unlearned model's MMLU performance drops to ~25%, which is random guessing. This indicates that unlearning completely destroyed the general knowledge in LLMs. In Table 5, the utility also drops to 0 on MUSE. In Table 6, the method is no better than RMU.

2. The proposed method seems to be a marginal improvement over baseline WGA.

**Questions For Authors:**

N/A

**Relation To Broader Scientific Literature:**

As mentioned in the paper, the proposed method is very similar to WGA. Although the paper contains detailed analyses of the impacts of the two reweighting terms, the final method seems to be an incremental change on WGA, making the contribution limited.

**Theoretical Claims:**

The paper does not involve theoretical claims.

---

> ### Author Rebuttal · Authors · 2025-04-01
>
> Many thanks for your constructive comments and suggestions! Please see our responses below.
>
> ‌**Q1: Is SatImp an incremental version of WGA?‌**
>
> We sincerely appreciate your comments, but we respectfully disagree with your assessment that SatImp is an incremental version. Before reading the response we prepare for you, please first refer to the response in **Reviewer 3V34 (Q2 and Q3)**.
>
> We apologize for any confusion that may arise. To begin with, we would like to highlight the difference between the WGA paper and ours.
>
> Specifically, the motivation of the WGA paper is presenting a toolkit of the gradient effect that qualifies the impacts of unlearning objectives on model performance. The literature focus on analyzing the details of diverse unlearning objective from a gradient perspective. WGA is an additional discovery in that paper, which could be an auxiliary strategy to enhance unlearning performance.
>
> Differently, the motivation of our submission is identifying the best reweighting strategy.
> Our submission contributes annotations of important tokens, summary of saturation-based methods, correlation between weights and losses in different paradigms, and combinations between different paradigms.
> Overall, the motivation and contribution are clearly distinct between these two papers.
>
>
> **Q2 Unsatisfied results and marginal improvements**
>
> Thanks for your comments about the experiments. However, we present more results and analysis that may help you understand this ‘marginal’ comprehensively.
> First, regarding the limited improvement, as demonstrated in **Reviewer 8Bwg Q2**, we have mentioned in the Appendix (Line 1152) that our hyper-parameter settings followed the exploration method based on the ES score in previous sections.
> We selected the hyper-param that achieved the best results on the ES score metric and fixed these params across all benchmarks.
> However, this setting is likely not optimal for other metrics (FQ, MU) or benchmarks (WMDP).
> As shown in **Reviewer 8Bwg Q2**, if FQ and MU are prioritized in the experiments, the ES results would decline, but the performance gap between different methods would widen‌.
>
> Regarding the WMDP results, the 25% performance on MMLU is caused by 125-step training process, which is too long for forget-only regularization in Table2. Except for NPO, the other methods are all leading to an over-unlearning situation. Thus, we utilize the early-stopping at the 30-th, 60-th, 90-th steps. The MMLU results are shown as follows:
> | Methods | 30-th | 60-th | 90-th |
> | ---- | ---- | ---- | ---- |
> |GA| 0.2865 | 0.2524 | 0.2454 |
> |NPO| 0.5607 | 0.5329 | 0.5239 |
> |SimNPO| 0.3418 | 0.2616 | 0.2493 |
> |RMU| 0.3193 | 0.2709 | 0.2697 |
> |WGA| 0.2963 | 0.2457 | 0.2463 |
> |SatImp| 0.3197 | 0.2687 | 0.2512 |
>
> Furthermore, we report the results at 30-th step.
> | Methods | WMDP-Bio | WMDP-Cyber | MMLU |
> | ---- | ---- | ---- | ---- |
> |GA| 0.2474 | 0.2431 | 0.2465 |
> |NPO| 0.5260 | 0.4616 | 0.5607 |
> |SimNPO| 0.3519 | 0.3562 | 0.3418 |
> |RMU| 0.2479 | 0.2963 | 0.3193 |
> |WGA| 0.2467 | 0.2617 | 0.2963 |
> |SatImp| 0.2474 | 0.2431 | 0.3197 |
>
> Additionally, unlearning with retain regularizations on WMDP (Table 6) are all under-unlearned, which are caused by too high $\lambda$ (the ratio between forget and retain objectives in **Eq. (2)**). We adjust $\lambda$ to accomplish more effective unlearning. Results are shown as follows:
> | Methods | WMDP-Bio | WMDP-Cyber | MMLU |
> | ---- | ---- | ---- | ---- |
> | GA| 0.2739 | 0.2657 | 0.4265 |
> | NPO| 0.2647 | 0.3067 | 0.4434 |
> | SimNPO| 0.2617 | 0.3163 | 0.4453 |
> | RMU| 0.3493 | 0.3578 | 0.5523 |
> | WGA| 0.2490 | 0.2989 | 0.4806 |
> | SatImp| 0.2598 | 0.2815 | 0.5391 |
>
> Regarding the MUSE results, we sincerely apologize that there is a typo in SatImp with retain regularization on MUSE-Books dataset.
> We wrongly recorded a forget-only regularization result here.
> The correct results should be:
> | Methods | VerbMem | KnowMem | PrivLeak | UtilPres |
> | ---- | ---- | ---- | ---- | ---- |
> | SatImp| 0.0 | 0.0 | -21.1 | 37.9|
>
> Regarding the MUSE results with forget-only regularizations, we have mentioned that we choose the checkpoint based on the Privacy Leak metric, which measures the degree of unlearning (whether the model is over-unlearning or under-unlearning).
> Extensive results indicate that model would have 0 utility when PrivLeak near to 0 under the forget-only regularization setting.
> NPO is different because it is significantly under-unlearned, which obtains a very low negative value on the PrivLeak metric.
> Thus, it is normal that almost all methods have 0 utility under the forget-only regularization setting.

---

### Decision · Program_Chairs · 2025-05-01

**Decision:**

Accept (poster)

**Comment:**

The paper studies the effect of reweighting on LLM unlearning through an empirical study. The problem is important and timely. The reviewers were mostly concerned about some details in the empirical results, particularly that certain cases show limited performance improvement. In the rebuttal, the authors provided more results to address the concerns. Two reviewers confirmed that they are satisfied with the response and additional results. One reviewer didn't provide feedback after the response, but based on my own reading, it appears that the concerns have been addressed too.